# Learning where to learn: Training data distribution optimization for scientific machine learning

## Abstract

In scientific machine learning, models are routinely deployed with parameter values or boundary conditions far from those used in training. This paper studies the learning-where-to-learn problem of designing a training data distribution that minimizes average prediction error across a family of deployment regimes. A theoretical analysis shows how the training distribution shapes deployment accuracy. This motivates two adaptive algorithms based on bilevel or alternating optimization in the space of probability measures. Discretized implementations using parametric distribution classes or nonparametric particle-based gradient flows deliver optimized training distributions that outperform nonadaptive designs. Once trained, the resulting models exhibit improved sample complexity and robustness to distribution shift. This framework unlocks the potential of principled data acquisition for learning functions and solution operators of partial differential equations.

## 1 Introduction

In scientific settings, machine learning models encounter data at deployment time that can differ substantially from the data seen during training. Standard empirical risk minimization (ERM), which minimizes an average loss over the fixed training distribution, may yield models that perform well in-sample yet degrade under distribution shift. This challenge is acute in operator learning for partial differential equations (PDEs) and infinite-dimensional inverse problems. For example, in data-driven electrical impedance tomography (EIT), neural operators are trained to map electrical patterns on the boundary to internal conductivities (Fan & Ying, 2020; Molinaro et al., 2023; Park et al., 2021). The input is a collection of voltage and corresponding current pairs that represent Dirichlet and Neumann boundary conditions, respectively. The probability distribution $\nu$ that boundary voltages are sampled from is a user choice. Crucially, the choice of $\nu$ shapes how well the trained model generalizes to new boundary measurement distributions. Fig. 1 illustrates this effect with a variant of the Neural Inverse Operator (NIO) architecture (Molinaro et al., 2023); see **SM** E–F for details.

These observations motivate the central premise of the present work: *learning where to learn* by designing an optimal training distribution $\nu$ for a prescribed family of deployment regimes. Guided by theoretical and practical considerations, this paper develops a principled and computationally feasible framework for optimizing $\nu$ in function regression and operator learning tasks. It establishes intelligent data acquisition as a crucial component of scientific machine learning (SciML) workflows.

**Contributions.** Our main contributions are as follows.

(C1) *Lipschitz-based distribution shift bounds.* We develop quantitative bounds that connect out-of-distribution (OOD) error to the training error, Lipschitz continuity of the model and target map, and the Wasserstein mismatch between training and test distributions.

(C2) *Algorithms for training distribution optimization.* Motivated by the theory, we develop two adaptive algorithms that actively optimize the training distribution for general model classes: (1) a bilevel procedure that directly minimizes OOD error and (2) a computationally efficient alternating scheme that minimizes an upper bound on the OOD error. We implement (1) and (2) over the space of probability measures by considering (a) parametric training distribution families (e.g., Gaussian processes) and (b) nonparametric particle representations of the training distribution (e.g., that are updated with Wasserstein gradient flows).

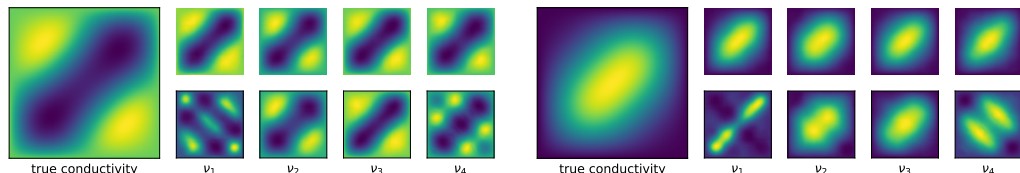

true conductivity    $\nu_1$    $\nu_2$    $\nu_3$    $\nu_4$      true conductivity    $\nu_1$    $\nu_2$    $\nu_3$    $\nu_4$

Figure 1: Two conductivity samples in EIT. The true conductivity is on the left of each panel, followed by predictions from NIO models trained on four Dirichlet boundary condition distributions $\{\nu_i\}_{i=1}^4$. Top row: in-distribution predictions; bottom row: out-of-distribution predictions. See **SM** E–F for details.

(C3) *Empirical performance in SciML tasks.* We perform numerical experiments involving function approximation as well as forward and inverse operator learning for a range of elliptic, parabolic, and hyperbolic PDEs (EIT, Darcy flow, viscous Burgers', and radiative transport). Consistent with our theory, the results show that the training data distribution strongly governs OOD deployment error. In the experiments, the two proposed adaptive algorithms efficiently optimize the training distribution, greatly reduce OOD error, and often outperform adaptive and nonadaptive baselines.

**Related work.** The careful selection of sampling points for regression appears frequently in numerical analysis (Barron et al., 2008; Krieg et al., 2022). Weighted least-squares regression with Christoffel sampling is one prominent example (Cohen & Migliorati, 2017; Adcock et al., 2022). One of the main findings of these works is that training on samples from the target test distribution is a suboptimal strategy in general (cf. Canatar et al. (2021)). Instead, a change of measure is derived by minimizing an upper bound on average error. These ideas have their origin in numerical integration and Monte Carlo methods (Caflisch, 1998). Some other related design strategies include greedy Bayesian herding (Huszár & Duvenaud, 2012) and kernel quadrature (Epperly & Moreno, 2023). However, quadrature-based sampling does not necessarily deliver optimal performance for regression.

In SciML for PDEs, several adaptive sampling and reweighting strategies have emerged, particularly for unsupervised methods such as physics-informed neural networks (PINNs) (Wu et al., 2023). Common approaches either reweight a fixed pool of training data samples (Chen et al., 2025a; McClenny & Braga-Neto, 2023) or sample new data points from a density proportional to the PDE residual (Gao et al., 2023; Tang et al., 2023). Similar methods based on importance sampling, reweighting, or coreset selection have been applied to neural operators (Wang et al., 2022b; Toscano et al., 2025). Others use residual information to move samples directly (Ouyang et al., 2025) or iteratively expand training datasets (Lye et al., 2021). The current work builds on these advances by operating at the level of probability distributions. This enables a wide range of parametrizations and discretizations beyond samples only and provides extra freedom to explore the design space.

For a discussion of broader connections to other areas of machine learning and statistics, see **SM** A.

**Outline.** This paper is organized as follows. Sec. 2 provides the necessary background. Sec. 3 establishes motivating theoretical results (C1). Sec. 4 proposes two implementable data distribution selection algorithms (C2). Sec. 5 performs numerical studies (C3) and Sec. 6 gives concluding remarks. Proof and implementation details are deferred to the supplementary material (**SM**).

## 2   PRELIMINARIES

This section sets the notation, provides essential background on key mathematical concepts, and reviews the framework of operator learning. **SM** B contains additional reference material.

**Notation.** This paper works with an abstract input space $(\mathcal{U}, \langle \cdot, \cdot \rangle_{\mathcal{U}}, \|\cdot\|_{\mathcal{U}})$ and output space $(\mathcal{Y}, \langle \cdot, \cdot \rangle_{\mathcal{Y}}, \|\cdot\|_{\mathcal{Y}})$, both assumed to be real separable Hilbert spaces. We often drop the subscripts on norms and inner products when the meaning is clear from the context. Unless otherwise stated, $\mathcal{U}$ or $\mathcal{Y}$ could be infinite-dimensional. We denote the Lipschitz constant of a map $F: \mathcal{U} \to \mathcal{Y}$ by $\mathrm{Lip}(F) \coloneqq \sup_{u \neq u'} \|F(u) - F(u')\|_{\mathcal{Y}} / \|u - u'\|_{\mathcal{U}}$. For any element $\zeta$ of a metric space, the Dirac probability measure assigning unit mass to $\zeta$ is written as $\delta_\zeta$. For $p \in [1, \infty]$, (Bochner) $L^p_\mu(\mathcal{U}; \mathcal{Y})$ spaces are defined in the usual way. We say that a probability measure $\mu \in \mathscr{P}(\mathcal{U})$ belongs to $\mathscr{P}_p(\mathcal{U})$ if $\mathsf{m}_p(\mu) < \infty$, where $\mathsf{m}_p: \mathscr{P}(\mathcal{U}) \to \mathbb{R}_{\geq 0} \cup \{\infty\}$ denotes the uncentered $p$-th moment defined by

$\mu \mapsto \mathsf{m}_p(\mu) \coloneqq \mathbb{E}_{u \sim \mu} \|u\|_{\mathcal{U}}^p$. We often work with Gaussian measures $\mathcal{N}(m, \mathcal{C}) \in \mathscr{P}_2(\mathcal{U})$, where $m \in \mathcal{U}$ is the mean and $\mathcal{C} \colon \mathcal{U} \to \mathcal{U}$ is the symmetric trace-class positive semidefinite covariance operator; see Dashti & Stuart (2017, App. A.3) for details in the infinite-dimensional setting.

**Background.**  We review several key concepts that frequently appear in this work.

The *pushforward operator* maps a measure from one measurable space to another via a measurable map. Specifically, given measure spaces $(X, \mathcal{B}_X, \mu)$ and $(Y, \mathcal{B}_Y, \nu)$ and a measurable map $T \colon X \to Y$, the pushforward operator $T_\#$ assigns to $\mu$ a measure $T_\# \mu$ on $Y$ defined by $T_\# \mu(B) = \mu(T^{-1}(B))$ for all $B \in \mathcal{B}_Y$, where $T^{-1}(B) = \{x \in X \,|\, T(x) \in B\}$ is the pre-image of $B$. Intuitively, $T_\#$ redistributes the measure $\mu$ on $X$ to a measure $T_\# \mu$ on $Y$ following the structure of the map $T$.

***Definition* 2.1** (*p*-Wasserstein distance)**.** Consider probability measures $\mu$ and $\nu$ on a metric space $(X, \mathsf{d})$ with finite $p$-th moments, where $p \in [1, \infty)$. Define the set of all couplings $\Gamma(\mu, \nu) \coloneqq \left\{ \gamma \in \mathscr{P}(X \times X) \,\middle|\, \pi_\#^1 \gamma = \mu, \ \pi_\#^2 \gamma = \nu \right\}$, where $\pi^1(x, y) = x$ and $\pi^2(x, y) = y$. The *p-Wasserstein distance* between $\mu$ and $\nu$ is $\mathsf{W}_p(\mu, \nu) \coloneqq (\inf_{\gamma \in \Gamma(\mu, \nu)} \int_{X \times X} \mathsf{d}(x, y)^p \, \gamma(dx, dy))^{1/p}$.

The cases of $p = 1$ and $p = 2$ are often considered, corresponding to the $\mathsf{W}_1$ and $\mathsf{W}_2$ metrics (Villani, 2021). The case when $\mu$ and $\nu$ are both Gaussian and $X$ is a Hilbert space is also used frequently.

We will use the notion of random measure (Kallenberg, 2017) to quantify OOD error in Secs. 3–4.

***Definition* 2.2** (random measure)**.** Let $(\Omega, \mathcal{F}, \mathbb{P})$ be a probability space, and let $(M, \mathcal{B})$ be a measurable space. A *random measure* on $M$ is a map $\Xi \colon \Omega \to \mathscr{P}(M)$, $\Xi(\omega) \coloneqq \mu_\omega$, such that for every $A \in \mathcal{B}$, the function $\omega \mapsto \mu_\omega(A)$ is a random variable, i.e., measurable from $(\Omega, \mathcal{F})$ to $(\mathbb{R}_{\geq 0}, \mathcal{B}(\mathbb{R}))$.

**Operator learning.**  Operator learning provides a powerful framework for approximating mappings $\mathcal{G}^\star \colon \mathcal{U} \to \mathcal{Y}$ between function spaces. It plays an increasingly pivotal role for accelerating forward and inverse problems in the computational sciences (Boullé & Townsend, 2024; Kovachki et al., 2023; 2024b; Morel et al., 2025; Nelsen & Stuart, 2024; Subedi & Tewari, 2025). As in traditional regression, the learned operator $\mathcal{G}_{\widehat{\theta}}$ is parametrized over a set $\Theta$ and minimizes an *empirical risk*

$$\widehat{\theta} \in \arg\min_{\theta \in \Theta} \frac{1}{N} \sum_{n=1}^{N} \big\| \mathcal{G}^\star(u_n) - \mathcal{G}_\theta(u_n) \big\|_{\mathcal{Y}}^2 . \tag{2.1}$$

The Hilbert norm in (2.1) and discretization-invariant architectures (e.g., DeepONet (Lu et al., 2021), Fourier Neural Operator (FNO) (Li et al., 2021), NIO (Molinaro et al., 2023)) are points of departure from classical function approximation. In (2.1), $\{u_n\} \sim \nu^{\otimes N}$, where $\nu$ is the training distribution over input functions. The choice and properties of $\nu$ are crucial. An inadequately selected $\nu$ may result in $\mathcal{G}_{\widehat{\theta}}(u)$ being a poor approximation to $\mathcal{G}^\star(u)$ for $u \sim \nu'$, where $\nu' \neq \nu$ (Adcock et al., 2022; Boullé et al., 2023; de Hoop et al., 2023; Li et al., 2024; Musekamp et al., 2025; Pickering et al., 2022; Subedi & Tewari, 2024). The present work aims to improve accuracy under distribution shift in both function approximation and operator learning settings.

## 3 LIPSCHITZ THEORY FOR OUT-OF-DISTRIBUTION ERROR BOUNDS

This section studies how loss functions behave under distribution shifts and the implications of such shifts on average-case OOD accuracy. The assumed Lipschitz continuity of the target map and the elements in the approximation class play a central role in the analysis. Although our theoretical contributions build upon established tools, we are not aware of any existing work that presents these results in the precise form developed here.

**Distribution shift inequalities.**  In the infinite data limit, the functional in (2.1) naturally converges to the so-called *expected risk*, which serves as a notion of model accuracy in this work. The following result forms the foundation for our forthcoming data selection algorithms in Sec. 4.

**Proposition 3.1** (distribution shift error)**.** *Let $\mathcal{G}_1$ and $\mathcal{G}_2$ both be Lipschitz continuous maps from Hilbert space $\mathcal{U}$ to Hilbert space $\mathcal{Y}$. For any $\nu \in \mathscr{P}_1(\mathcal{U})$ and $\nu' \in \mathscr{P}_1(\mathcal{U})$, it holds that*

$$\mathbb{E}_{u' \sim \nu'} \|\mathcal{G}_1(u') - \mathcal{G}_2(u')\|_{\mathcal{Y}} \leq \mathbb{E}_{u \sim \nu} \|\mathcal{G}_1(u) - \mathcal{G}_2(u)\|_{\mathcal{Y}} + \big(\mathrm{Lip}(\mathcal{G}_1) + \mathrm{Lip}(\mathcal{G}_2)\big) \mathsf{W}_1(\nu, \nu'). \tag{3.1}$$

*Moreover, for any $\mu \in \mathscr{P}_2(\mathcal{U})$ and $\mu' \in \mathscr{P}_2(\mathcal{U})$, it holds that*

$$\mathbb{E}_{u' \sim \mu'} \|\mathcal{G}_1(u') - \mathcal{G}_2(u')\|_{\mathcal{Y}}^2 \leq \mathbb{E}_{u \sim \mu} \|\mathcal{G}_1(u) - \mathcal{G}_2(u)\|_{\mathcal{Y}}^2 + c(\mathcal{G}_1, \mathcal{G}_2, \mu, \mu') \mathsf{W}_2(\mu, \mu'), \tag{3.2}$$

*where $c(\mathcal{G}_1, \mathcal{G}_2, \mu, \mu')$ equals*

$$\left(\mathrm{Lip}(\mathcal{G}_1) + \mathrm{Lip}(\mathcal{G}_2)\right)\sqrt{4\left(\mathrm{Lip}(\mathcal{G}_1) + \mathrm{Lip}(\mathcal{G}_2)\right)^2\left[\mathsf{m}_2(\mu) + \mathsf{m}_2(\mu')\right] + 16\left[\|\mathcal{G}_1(0)\|_{\mathcal{Y}}^2 + \|\mathcal{G}_2(0)\|_{\mathcal{Y}}^2\right]}.$$

See **SM** C for the proof. Provided that the maps $\mathcal{G}_1$ and $\mathcal{G}_2$ are Lipschitz, Prop. 3.1 shows that the functional $\mu \mapsto \mathbb{E}_{u\sim\mu}\|\mathcal{G}_1(u) - \mathcal{G}_2(u)\|$ is globally Lipschitz continuous in the $\mathsf{W}_1$ metric. Similarly, the functional $\mu \mapsto \mathbb{E}_{u\sim\mu}\|\mathcal{G}_1(u) - \mathcal{G}_2(u)\|^2$ is *locally* Lipschitz continuous in the $\mathsf{W}_2$ metric; in particular, it is Lipschitz on sets with uniformly bounded second moments. We remark that the Lipschitz map hypotheses in Prop. 3.1 are invoked for simplicity. Hölder continuity is also compatible with Wasserstein distances, and the preceding result can be adapted to hold under such regularity.

**Average-case performance.** OOD performance may be quantified in several ways. This paper chooses *average-case accuracy* with respect to the law $\mathbb{Q} \in \mathscr{P}(\mathscr{P}(\mathcal{U}))$ of a random probability measure $\nu' \sim \mathbb{Q}$. For a ground truth map $\mathcal{G}^\star$ and any hypothesis $\mathcal{G}$, we strive for

$$\mathscr{E}_{\mathbb{Q}}(\mathcal{G}) \coloneqq \mathbb{E}_{\nu'\sim\mathbb{Q}}\,\mathbb{E}_{u'\sim\nu'}\big\|\mathcal{G}^\star(u') - \mathcal{G}(u')\big\|_{\mathcal{Y}}^2 \tag{3.3}$$

to be small. An equivalent viewpoint in (D.2) from **SM** D.1 rewrites (3.3) as a single expectation against a mixture. Other approaches to quantify OOD error are possible, such as worst-case analysis. We choose average-case accuracy because it is more natural for learning from finite samples and because the linearity of (3.3) with respect to $\mathbb{Q}$ is useful for algorithm development.

The following corollary is useful. It is a consequence of Prop. 3.1 and the Cauchy–Schwarz inequality.

**Corollary 3.2** (basic inequality). *For any $\mu \in \mathscr{P}_2(\mathcal{U})$ and maps $\mathcal{G}^\star$ and $\mathcal{G}$, it holds that*

$$\mathscr{E}_{\mathbb{Q}}(\mathcal{G}) \leq \mathbb{E}_{u\sim\mu}\big\|\mathcal{G}^\star(u) - \mathcal{G}(u)\big\|_{\mathcal{Y}}^2 + \sqrt{\mathbb{E}_{\nu'\sim\mathbb{Q}}\,c^2(\mathcal{G}^\star, \mathcal{G}, \mu, \nu')}\sqrt{\mathbb{E}_{\nu'\sim\mathbb{Q}}\,\mathsf{W}_2^2(\mu, \nu')}. \tag{3.4}$$

The upper bound (3.4) on the average-case OOD error $\mathscr{E}_{\mathbb{Q}}$ is intuitive. We view the arbitrary probability measure $\mu$ as the training data distribution that we are free to select. The first term on the right-hand side of (3.4) then represents the in-distribution generalization error of $\mathcal{G}$. On the other hand, the second term can be considered a regularizer that penalizes how far $\mu$ is from $\mathbb{Q}$ on average with respect to the $\mathsf{W}_2$ metric. The factor involving the quantity $c$ further penalizes the size of moments of $\mu$ and $\mathbb{Q}$ as well as the size of the Lipschitz constants of $\mathcal{G}^\star$ and $\mathcal{G}$.

We instantiate Cor. 3.2 in the concrete context of Gaussian mixtures in **SM** C, Ex. C.1.

Although the preceding OOD results hold in a quite general setting, they do not distinguish between an arbitrary model and a trained model obtained by ERM (2.1). Sharpening the bounds for a trained model would require probabilistic estimates on the model's Lipschitz constant and in-distribution error. These are challenging theoretical tasks. Instead, we turn our attention to implementable algorithms inspired by Cor. 3.2 that are designed to reduce OOD error in practice.

## 4 PRACTICAL ALGORITHMS FOR TRAINING DISTRIBUTION DESIGN

Let $\mathcal{G}^\star\colon \mathcal{U} \to \mathcal{Y}$ be the target mapping and $\mathbb{Q}$ be a probability measure over $\mathscr{P}_2(\mathcal{U})$. We assume query access to an oracle that returns the label $\mathcal{G}^\star(u) \in \mathcal{Y}$ when presented with an input $u \in \mathcal{U}$. The idealized infinite data training distribution selection problem that we wish to address is

$$\inf\left\{\mathscr{E}_{\mathbb{Q}}\big(\widehat{\mathcal{G}}^{(\nu)}\big) \,\middle|\, \nu \in \mathscr{P}_2(\mathcal{U}) \quad \text{and} \quad \widehat{\mathcal{G}}^{(\nu)} \in \underset{\mathcal{G}\in\mathcal{H}}{\arg\min}\,\mathbb{E}_{u\sim\nu}\big\|\mathcal{G}^\star(u) - \mathcal{G}(u)\big\|_{\mathcal{Y}}^2\right\}, \tag{4.1}$$

where $\mathscr{E}_{\mathbb{Q}}$ is the average-case OOD accuracy functional (3.3) and $\mathcal{H}$ is the model hypothesis class.

Eqn. (4.1) is a bilevel optimization problem. The inner minimization problem seeks the best-fit model $\widehat{\mathcal{G}}^{(\nu)}$ for a fixed training distribution $\nu$. The outer problem minimizes $\mathscr{E}_{\mathbb{Q}}\big(\widehat{\mathcal{G}}^{(\nu)}\big)$ over $\nu$ in the space of probability measures. The map $\nu \mapsto \widehat{\mathcal{G}}^{(\nu)}$ can be highly nonlinear, which poses a challenge to solve (4.1). To devise practical solution methods, this section considers the bilevel optimization problem (4.1) in Sec. 4.1 and a computationally tractable relaxation of (4.1) in Sec. 4.2.

***Remark* 4.1** (finite data). It is necessary to use finite data to estimate the optimal training distribution (**SM** D.1). We thus assume sample access to the chosen feasible set of training distributions, e.g., i.i.d. sampling or interacting particles. Moreover, we only require sample access to samples from $\mathbb{Q}$.

## 4.1 Exact bilevel formulation

This subsection focuses on directly solving the bilevel optimization problem (4.1) to identify the optimal training distribution. Bilevel optimization has been extensively studied in the literature; see, for example, the work of Colson et al. (2007); Golub & Pereyra (2003); Van Leeuwen & Aravkin (2021). For the infinite-dimensional problem (4.1), our approach begins by specifying a Hilbert space $\mathcal{H}$. Using its linear structure, we calculate the gradient of the Lagrangian associated with (4.1) with respect to the variable $\nu$. This gradient computation leverages the *adjoint state method* and the envelope theorem (Afriat, 1971). In what follows, we specialize this general framework to the case where $\mathcal{H}$ is a reproducing kernel Hilbert space (RKHS) with scalar output space $\mathcal{Y} = \mathbb{R}$. The resulting algorithm can be efficiently implemented using representer theorems for kernel methods (Micchelli & Pontil, 2005). Additionally, other linear classes of functions can be incorporated into this framework with minimal effort. Extending the framework to nonlinear, vector-valued function spaces $\mathcal{H}$, such as neural network or neural operator spaces, is an exciting direction for future research.

**RKHS.** Let $\mathcal{H} := (\mathcal{H}_\kappa, \langle \cdot, \cdot \rangle_\kappa, \| \cdot \|_\kappa)$ be a separable RKHS of real-valued functions on $\mathcal{U}$ with bounded continuous reproducing kernel $\kappa \colon \mathcal{U} \times \mathcal{U} \to \mathbb{R}$. For fixed $\nu \in \mathscr{P}_2(\mathcal{U})$, let $\iota_\nu \colon \mathcal{H} \hookrightarrow L_\nu^2$ be the canonical inclusion map (which is compact under the assumptions on $\kappa$). The adjoint of the inclusion map is the kernel integral operator $\iota_\nu^* \colon L_\nu^2 \to \mathcal{H}$ defined by $\iota_\nu^* h = \int_{\mathcal{U}} \kappa(\,\cdot\,, u) h(u) \nu(du)$. We also define the self-adjoint operator $\mathcal{K}_\nu := \iota_\nu^* \iota_\nu \colon \mathcal{H} \to \mathcal{H}$, which has the same action as $\iota_\nu^*$.

**Method of adjoints.** Assume that $\mathcal{G}^\star \in L_\nu^2 \equiv L_\nu^2(\mathcal{U}; \mathbb{R})$ for every $\nu \in \mathscr{P}_2(\mathcal{U})$ under consideration; this is true, for example, if $\mathcal{G}^\star$ is uniformly bounded on $\mathcal{U}$. Define the data misfit $\Psi \colon \mathcal{H} \to \mathbb{R}$ by $\Psi(\mathcal{G}) := \frac{1}{2} \| \iota_\nu \mathcal{G} - \mathcal{G}^\star \|_{L_\nu^2}^2$; this is an abstract way to write the convex inner objective in (4.1) (rescaled by $1/2$). By Lem. D.1, the inner solution $\widehat{\mathcal{G}}^{(\nu)}$ solves the equation $\mathcal{K}_\nu \widehat{\mathcal{G}}^{(\nu)} = \iota_\nu^* \mathcal{G}^\star$ in $\mathcal{H}$. This fact allows us to define the Lagrangian $\mathcal{L} \colon \mathscr{P}_2(H) \times \mathcal{H} \times \mathcal{H} \to \mathbb{R}$ for (4.1) by

$$\mathcal{L}(\nu, \mathcal{G}, \lambda) = \frac{1}{2} \mathbb{E}_{\nu' \sim \mathbb{Q}} \| \iota_{\nu'} \mathcal{G} - \mathcal{G}^\star \|_{L_{\nu'}^2}^2 + \big\langle (\mathcal{K}_\nu \mathcal{G} - \iota_\nu^* \mathcal{G}^\star), \lambda \big\rangle_\kappa . \tag{4.2}$$

Differentiating $\mathcal{L}$ with respect to $\mathcal{G} \in \mathcal{H}$ leads to the adjoint equation.

**Lemma 4.2** (adjoint state equation). *At a critical point of the map $\mathcal{G} \mapsto \mathcal{L}(\nu, \mathcal{G}, \lambda)$ for fixed $(\nu, \mathcal{G})$, the adjoint state $\lambda = \lambda^{(\nu)}(\mathcal{G}) \in \mathcal{H}$ satisfies the compact operator equation*

$$\mathcal{K}_\nu \lambda^{(\nu)}(\mathcal{G}) = \mathbb{E}_{\nu' \sim \mathbb{Q}} \mathbb{E}_{u' \sim \nu'} \big[ \kappa(\,\cdot\,, u') \big( \mathcal{G}^\star(u') - \mathcal{G}(u') \big) \big] . \tag{4.3}$$

The preceding results enable us to differentiate $\mathcal{L}$ with respect to $\nu$ along the constraints of (4.1).

**Proposition 4.3** (derivative: infinite-dimensional case). *Define $\mathcal{J} \colon \mathscr{P}_2(\mathcal{U}) \to \mathbb{R}$ by $\mathcal{J}(\nu) := \mathcal{L}(\nu, \widehat{\mathcal{G}}^{(\nu)}, \lambda^{(\nu)}(\widehat{\mathcal{G}}^{(\nu)}))$. Then, in the sense of duality, $(D\mathcal{J})(\nu) = (\widehat{\mathcal{G}}^{(\nu)} - \mathcal{G}^\star) \lambda^{(\nu)}(\widehat{\mathcal{G}}^{(\nu)})$ for each $\nu$.*

**SM** D.2 provides the proofs of Lem. 4.2 and Prop. 4.3 and implementation details.

*Remark* 4.4 (Wasserstein gradient flow). A nonparametric approach to solve the bilevel problem (4.1) is to derive the Wasserstein gradient flow and its Lagrangian equivalence $\dot{u} = -\nabla(D\mathcal{J})(\mathrm{Law}(u))(u)$, where $D\mathcal{J}$ is as in Prop. 4.3. This leads to a curve of measures $t \mapsto \mathrm{Law}(u(t))$ converging to a steady-state probability distribution. It is important to note that this dynamic requires gradients of the true map $\mathcal{G}^\star$, which may not be available or could be too expensive to compute even if available. To implement this mean-field evolution, we can discretize it with interacting particles; see **SM** F.5.

**Parametrized training distributions.** Although Prop. 4.3 provides the desired derivative of the bilevel problem (4.1), it does not specify how to use the derivative to compute the optimal $\nu$. This issue is nontrivial because $\nu \in \mathscr{P}_2(\mathcal{U})$ belongs to a nonlinear metric space. One approach assumes a parametrization $\nu = \nu_\vartheta$, where $\vartheta \in \mathcal{V}$ for some separable Hilbert parameter space $\mathcal{V}$. Typically $\mathcal{V} \subseteq \mathbb{R}^P$ for some $P \in \mathbb{N}$. We then use the chain rule to apply Euclidean gradient-based optimization tools to solve (4.1). To this end, define $\mathsf{J} \colon \mathcal{V} \to \mathbb{R}$ by $\vartheta \mapsto \mathsf{J}(\vartheta) := \mathcal{J}(\nu_\vartheta)$, where $\mathcal{J}$ is as in Prop. 4.3. The following theorem computes the gradient $\nabla \mathsf{J} \colon \mathcal{V} \to \mathcal{V}$.

**Theorem 4.5** (gradient: parametric case). *Suppose there exists a dominating measure $\mu_0$ such that for every $\vartheta \in \mathcal{V}$, $\nu_\vartheta$ has density $p_\vartheta$ with respect to $\mu_0$. If $(u, \vartheta) \mapsto p_\vartheta(u)$ is sufficiently regular, then*

$$\nabla \mathsf{J}(\vartheta) = \int_{\mathcal{U}} \big( \widehat{\mathcal{G}}^{(\nu_\vartheta)}(u) - \mathcal{G}^\star(u) \big) \big( \lambda^{(\nu_\vartheta)}(\widehat{\mathcal{G}}^{(\nu_\vartheta)}) \big)(u) \big( \nabla_\vartheta \log p_\vartheta(u) \big) \nu_\vartheta(du) . \tag{4.4}$$

---

**Algorithm 1** Training Data Design via Gradient Descent on a Parametrized Bilevel Objective

---

1: **Initialize:** Parameter $\vartheta^{(0)}$, step sizes $\{t_k\}$
2: **for** $k = 0, 1, 2, \ldots$ **do**
3:     **Gradient step:** With $\nabla \mathsf{J}$ as in (4.4), update the training distribution's parameters via
$$\vartheta^{(k+1)} = \vartheta^{(k)} - t_k \nabla \mathsf{J}(\vartheta^{(k)})$$
4:     **if** stopping criterion is met **then**
5:         Return $\vartheta^{(k+1)}$ then **break**

---

In Thm. 4.5, $\mu_0$ is typically the Lebesgue measure if $\mathcal{U} \subseteq \mathbb{R}^d$. **SM** D.2 contains the proof.

***Example* 4.6** (Gaussian parametrization)**.** Consider the Gaussian model $\nu_\vartheta := \mathcal{N}(m_\vartheta, \mathcal{C}_\vartheta)$ with $\mu_0$ being the Lebesgue measure on $\mathcal{U} = \mathbb{R}^d$. Explicit calculations with Gaussian densities show that

$$\nabla_\vartheta \log p_\vartheta(u) = -\nabla_\vartheta \Phi(u, \vartheta), \text{ where } \Phi(u, \vartheta) := \tfrac{1}{2} \big\| \mathcal{C}_\vartheta^{-1/2}(u - m_\vartheta) \big\|_2^2 + \tfrac{1}{2} \log \det(\mathcal{C}_\vartheta). \quad (4.5)$$

The parameters $\vartheta$ of $\nu_\vartheta$ can now be updated using any gradient-based optimization method. For instance, a simple gradient flow in Hilbert space is given by $\dot{\vartheta} = -\nabla \mathsf{J}(\vartheta)$.

**Implementation.** A gradient descent scheme for the bilevel optimization problem (4.1) is summarized in Alg. 1. Alternative gradient-based optimization methods can also be explored. We emphasize that the derived bilevel gradients are exact; in implementation, the only source of error arises from discretization. Discretization details for the gradient (4.4) are provided in **SM** D.2; in particular, we "empiricalize" all expectations.

### 4.2 Upper-bound minimization

Although Alg. 1 is implementable assuming query access to the ground truth map $u \mapsto \mathcal{G}^\star(u)$, it is so far restricted to linear vector spaces $\mathcal{H}$. This subsection relaxes problem (4.1) by instead minimizing an upper bound on the objective, which is no longer a bilevel optimization problem. We then design an alternating descent algorithm that applies to any hypothesis class $\mathcal{H}$ (e.g., neural networks and neural operators). Empirically, the alternating algorithm converges fast, often in a few iterations.

**An alternating scheme.** Recall the upper bound (3.4) from Cor. 3.2, which holds for any $\mu \in \mathscr{P}_2(\mathcal{U})$. The particular choice $\mu := \nu$, where $\nu$ is a candidate training data distribution, delivers a valid upper bound for the objective functional $\mathscr{E}_\mathbb{Q}(\widehat{\mathcal{G}}^{(\nu)})$ in (4.1). That is,

$$\mathscr{E}_\mathbb{Q}(\widehat{\mathcal{G}}^{(\nu)}) \leq \mathbb{E}_{u \sim \nu} \big\| \mathcal{G}^\star(u) - \widehat{\mathcal{G}}^{(\nu)}(u) \big\|_\mathcal{Y}^2 + \sqrt{\mathbb{E}_{\nu' \sim \mathbb{Q}} \, c^2\big(\mathcal{G}^\star, \widehat{\mathcal{G}}^{(\nu)}, \nu, \nu'\big)} \sqrt{\mathbb{E}_{\nu' \sim \mathbb{Q}} \, \mathsf{W}_2^2(\nu, \nu')}. \quad (4.6)$$

It is thus natural to approximate the solution to (4.1) by the minimizing distribution of the right-hand side of (4.6). The dependence of the factor $c$ in Prop. 3.1 on the Lipschitz constant of $\widehat{\mathcal{G}}^{(\nu)}$ is hard to capture in practice. One way to circumvent this difficulty is to further assume an upper bound $\mathrm{Lip}(\widehat{\mathcal{G}}^{(\nu)}) \leq R$ uniformly in $\nu$ for some constant $R > 0$. We define $c_R(\mathcal{G}^\star, \nu, \nu')$ to be $c(\mathcal{G}^\star, \widehat{\mathcal{G}}^{(\nu)}, \nu, \nu')$ with all instances of $\mathrm{Lip}(\widehat{\mathcal{G}}^{(\nu)})$ replaced by $R$. The assumption of a uniform bound on the Lipschitz constant is valid for certain chosen model classes $\mathcal{H}$, e.g., kernels with Lipschitz feature maps or Lipschitz-constrained neural networks (Gouk et al., 2021). Although such Lipschitz control in deep neural networks is challenging, there has been substantial recent progress on architectures with explicit Lipschitz constraints, e.g., spectral normalization, orthogonal/Householder layers, 1-Lipschitz activations (Miyato et al., 2018; Anil et al., 2019; Wang et al., 2020; Murari et al., 2025). Thus, expressive neural models with controlled Lipschitz behavior can be made practical.

We denote the candidate set of training distributions by $\mathsf{P} \subseteq \mathscr{P}_2(\mathcal{U})$. Finally, the upper-bound minimization problem becomes

$$\inf_{(\nu, \mathcal{G}) \in \mathsf{P} \times \mathcal{H}} \left\{ \mathbb{E}_{u \sim \nu} \big\| \mathcal{G}^\star(u) - \mathcal{G}(u) \big\|_\mathcal{Y}^2 + \sqrt{\mathbb{E}_{\nu' \sim \mathbb{Q}} \, c_R^2\big(\mathcal{G}^\star, \nu, \nu'\big)} \sqrt{\mathbb{E}_{\nu' \sim \mathbb{Q}} \, \mathsf{W}_2^2(\nu, \nu')} \right\}. \quad (4.7)$$

Note that since only the first term of (4.7) depends on the variable $\mathcal{G}$, the minimizing $\mathcal{G}$ is given by $\widehat{\mathcal{G}}^{(\nu)}$, as defined previously. This observation motivates the alternating scheme outlined in Alg. 2.

---

**Algorithm 2** Training Data Design via Alternating Model Fitting and Distribution Update Steps

---

1: **Initialize:** Training data distribution $\nu^{(0)}$, estimate of the constant $R > 0$
2: **for** $k = 0, 1, 2, \ldots$ **do**
3:     **Model training:** To obtain the trained model $\widehat{\mathcal{G}}^{(k)}$, solve the optimization problem

$$\widehat{\mathcal{G}}^{(k)} \leftarrow \arg\min_{\mathcal{G} \in \mathcal{H}} \mathbb{E}_{u \sim \nu^{(k)}} \left\| \mathcal{G}^{\star}(u) - \mathcal{G}(u) \right\|_{\mathcal{Y}}^2$$

4:     **Distribution update:** Update the training distribution by solving

$$\nu^{(k+1)} \leftarrow \arg\min_{\nu \in \mathsf{P}} \left\{ \mathbb{E}_{u \sim \nu} \left\| \mathcal{G}^{\star}(u) - \widehat{\mathcal{G}}^{(k)}(u) \right\|_{\mathcal{Y}}^2 + \sqrt{\mathbb{E}_{\nu' \sim \mathbb{Q}} \, c_R^2(\mathcal{G}^{\star}, \nu, \nu')} \sqrt{\mathbb{E}_{\nu' \sim \mathbb{Q}} \, \mathsf{W}_2^2(\nu, \nu')} \right\}$$

5:     **if** stopping criterion is met **then**
6:         Return $(\widehat{\mathcal{G}}^{(k)}, \nu^{(k+1)})$ then **break**

---

It is important to note that not all alternating algorithms have convergence guarantees, which requires a detailed understanding of the associated fixed-point operator. In contrast, the proposed Alg. 2 comes with the following convergence guarantee.

**Proposition 4.7** (monotonicity)**.** *Under Alg. 2, the objective in* (4.7) *is always non-increasing.*

*Proof.* Line 3 ensures that the first term of the objective (4.7) does not increase, while keeping the second term constant. Next, Line 4 minimizes both terms. Iterating this argument, we deduce that the algorithm maintains a non-increasing objective value for all $k$. $\qquad\square$

***Remark* 4.8** (connection to Wasserstein barycenters)**.** If the set $\mathsf{P}$ contains distributions with uniformly bounded second moments and the first term on the right-hand side of (4.6) is uniformly bounded over $\mathsf{P}$, then the optimal value of (4.7) is bounded above by the value of another minimization problem whose solution in $\mathscr{P}_2(\mathcal{U})$ is the 2-Wasserstein barycenter of $\mathbb{Q} \in \mathscr{P}_2(\mathscr{P}_2(\mathcal{U}))$ (Backhoff-Veraguas et al., 2022, Thm 3.7, p. 444). An analogous bound is even easier to verify for the $\mathsf{W}_1$ result (3.1). Thus, the $\mathcal{G}^{\star}$-dependent solution of the proposed problem (4.7) over $\mathsf{P}$ (and hence also the solution of Eqn. (4.1)) is always *at least as good* as the barycenter, which only depends on $\mathsf{P}$ and $\mathbb{Q}$.

**Implementation.** To implement Alg. 2, we simply replace expectations with finite averages over empirical samples. The number of samples per iteration is a hyperparameter. Line 3 in Alg. 2 is the usual model training step, i.e., Eqn. (2.1), while the distribution update step in Line 4 can be achieved in several ways. One way involves interacting particle systems derived from (Wasserstein) gradient flows over $\mathsf{P} \subseteq \mathscr{P}_2(\mathcal{U})$. Alternatively, if the distribution is parametrized, we can optimize over the parametrization, e.g., transport maps or Gaussian means and covariances. In this case, one can apply global optimization methods or Euclidean gradient-based solvers.

To apply gradient-based optimization algorithms, we need the derivative of the objective in (4.7) with respect to $\nu$. Up to constants, the objective takes the form $F \colon \mathsf{P} \subseteq \mathscr{P}_2(\mathcal{U}) \to \mathbb{R}$, where

$$\nu \mapsto F(\nu) \coloneqq \mathbb{E}_{u \sim \nu} f(u) + \sqrt{1 + \mathsf{m}_2(\nu)} \sqrt{\mathbb{E}_{\nu' \sim \mathbb{Q}} \, \mathsf{W}_2^2(\nu, \nu')} \tag{4.8}$$

and $f \colon \mathcal{U} \to \mathbb{R}$. The next result differentiates $F$; the proof is in **SM** D.3.

**Proposition 4.9** (derivative: infinite-dimensional case)**.** *Let $F$ be as in* (4.8) *with $f$ and $\mathsf{P}$ sufficiently regular. Then for every $\nu \in \mathsf{P}$ and $u \in \mathcal{U}$, it holds that*

$$\big(DF(\nu)\big)(u) = f(u) + \frac{1}{2} \sqrt{\frac{\mathbb{E}_{\nu' \sim \mathbb{Q}} \, \mathsf{W}_2^2(\nu, \nu')}{1 + \mathsf{m}_2(\nu)}} \|u\|_{\mathcal{U}}^2 + \sqrt{\frac{1 + \mathsf{m}_2(\nu)}{\mathbb{E}_{\nu' \sim \mathbb{Q}} \, \mathsf{W}_2^2(\nu, \nu')}} \, \mathbb{E}_{\nu' \sim \mathbb{Q}} \, \phi_{\nu, \nu'}(u) \tag{4.9}$$

*in the sense of duality. The potential $\phi_{\nu, \nu'} \colon \mathcal{U} \to \mathbb{R}$ satisfies $\phi_{\nu, \nu'} = \frac{1}{2} \| \cdot \|_{\mathcal{U}}^2 - \varphi_{\nu, \nu'}$, where $\nabla \varphi_{\nu, \nu'} = T_{\nu \to \nu'}$ and $T_{\nu \to \nu'}$ is the $\mathsf{W}_2$-optimal transport map from $\nu$ to $\nu'$.*

Taking the gradient of $u \mapsto DF(\nu)(u)$ in (4.9) yields the Wasserstein gradient, which can be utilized in Wasserstein gradient flow discretizations; see Rmk. 4.4. In the case of parametric distributions, combining Prop. 4.9 with the chain rule yields gradient descent schemes similar to those introduced in Sec. 4.1. Again, the derived gradients are exact and require only discretization for their numerical evaluation. For further details, see Cor. D.7 and Lem. D.8 and its special case (Rmk. D.9 in **SM** D.3).

Table 1: Function approximation with bilevel Alg. 1. Reported values are mean Err for kernel regressors trained with 1024 labeled data samples from the optimized $\nu_\vartheta$ (ours) from Alg. 1 with 1000 iterations vs. other adaptive (aCoreSet) and nonadaptive (Normal, Barycenter, Mixture, Uniform, nCoreSet) sampling distributions. Two standard deviations from the mean Err over 10 independent runs is also reported. Also see Fig. 10.

|           | $g_1$ $(d=2)$ | $g_2$ $(d=5)$ | $g_2$ $(d=8)$ | $g_3$ $(d=4)$ | $g_3$ $(d=5)$ | $g_4$ $(d=10)$ |
|-----------|---------------|---------------|---------------|---------------|---------------|----------------|
| *Ours*    | **0.040** $\pm$ 0.010 | **0.169** $\pm$ 0.007 | **0.556** $\pm$ 0.007 | **0.276** $\pm$ 0.019 | 0.558 $\pm$ 0.009 | 0.025 $\pm$ 0.000 |
| Normal    | 0.808 $\pm$ 0.015 | 0.811 $\pm$ 0.010 | 0.949 $\pm$ 0.003 | 0.937 $\pm$ 0.005 | 0.947 $\pm$ 0.003 | 0.029 $\pm$ 0.001 |
| Barycenter | 0.546 $\pm$ 0.025 | 0.514 $\pm$ 0.012 | 0.782 $\pm$ 0.005 | 0.771 $\pm$ 0.008 | 0.784 $\pm$ 0.010 | 0.021 $\pm$ 0.001 |
| Mixture   | 0.308 $\pm$ 0.011 | 0.372 $\pm$ 0.015 | 0.712 $\pm$ 0.011 | 0.471 $\pm$ 0.037 | 0.658 $\pm$ 0.011 | 0.022 $\pm$ 0.001 |
| Uniform   | 0.983 $\pm$ 0.001 | 0.971 $\pm$ 0.002 | 0.993 $\pm$ 0.000 | 0.976 $\pm$ 0.000 | 1.000 $\pm$ 0.000 | 0.121 $\pm$ 0.001 |
| nCoreSet  | 0.185 $\pm$ 0.000 | 0.247 $\pm$ 0.002 | 0.622 $\pm$ 0.002 | 0.295 $\pm$ 0.002 | **0.494** $\pm$ 0.002 | 0.098 $\pm$ 0.012 |
| aCoreSet  | 0.501 $\pm$ 0.060 | 0.538 $\pm$ 0.028 | 0.782 $\pm$ 0.018 | 0.793 $\pm$ 0.048 | 0.848 $\pm$ 0.034 | **0.013** $\pm$ 0.000 |

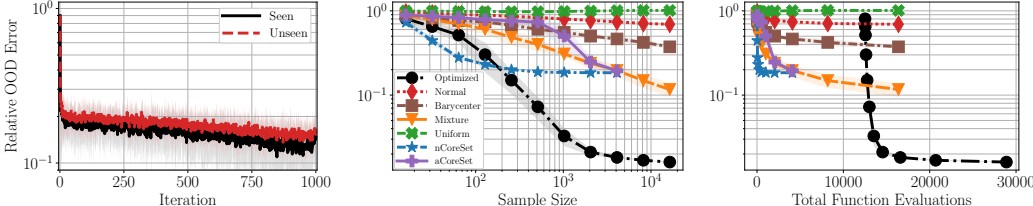

Figure 2: Alg. 1 applied to ground truth $g_1 \colon \mathbb{R}^2 \to \mathbb{R}$. (Left) Evolution of Err over 1000 iterations ($N = 250$ samples per step) of gradient descent. (Center) Err of model trained on $N$ samples from optimized $\nu_\vartheta$ (ours) vs. initial normal $\mathcal{N}(m_0, I_2)$, empirical $\mathbb{Q}$ $W_2$-barycenter, empirical $\mathbb{Q}$ mixture, $\mathrm{Unif}([0,1]^2)$, and two pool-based coresets. (Right) Same as center, except incorporating the additional function evaluation cost incurred from Alg. 1. Shading represents two standard deviations away from the mean Err over 10 independent runs.

## 5 NUMERICAL RESULTS

This section implements Alg. 1 (the bilevel gradient descent algorithm) for function approximation using kernel methods and Alg. 2 (the alternating minimization algorithm) for several operator learning tasks using DeepONets. Comprehensive experiment details and settings are collected in **SM** F.

**Bilevel gradient descent algorithm.** We instantiate a discretization of Alg. 1 on several function approximation benchmarks. The test functions $\mathcal{G}^\star \in \{g_i\}_{i=1}^4$ each map $\mathbb{R}^d$ to $\mathbb{R}$ and are comprised of the Sobol G ($g_1$) and Friedmann functions ($g_2, g_3$), and a kernel expansion ($g_4$). We use the kernel framework from Sec. 4.1 with $\mathcal{H}$ the RKHS of the squared exponential kernel with a fixed scalar bandwidth hyperparameter. Also, $\mathbb{Q}$ is an empirical measure with 10 atoms centered at fixed Gaussian measures realized by standard normally distributed means and Wishart distributed covariance matrices. A cosine annealing scheduler determines the gradient descent step sizes. The training distribution $\nu_\vartheta := \mathcal{N}(m, \mathcal{C})$ is parametrized by the pair $\vartheta = (m, \mathcal{C})$, which we optimize using Alg. 1.

Tab. 1 shows that models trained on the final iterate $\nu_\vartheta$ of Alg. 1 nearly always outperforms those trained on adaptive or nonadaptive baseline distributions in terms of the root relative average OOD squared error $\mathrm{Err} := (\mathscr{E}_{\mathbb{Q}}(\widehat{\mathcal{G}}^{(\nu)})/\mathscr{E}_{\mathbb{Q}}(0))^{1/2}$. Nonadaptive (nCoreSet) and adaptive (aCoreSet) pool-based coresets from active learning (Musekamp et al., 2025) slightly outperform $\nu_\vartheta$ on the $g_3$ ($d = 5$) and $g_4$ ($d = 10$) test problems, respectively. Here, the fixed pool of size 5000 consists of the empirical samples of samples from $\mathbb{Q}$ available to Alg. 1. Overall, the performance gap between our bilevel method and the baselines can close depending on the complexity of $\mathbb{Q}$ or $\mathcal{G}^\star$. Fig. 2 specializes to the $\mathcal{G}^\star = g_1$ case. Although fitting the model on data sampled from the precomputed optimal $\nu_\vartheta$ uniformly outperforms baseline distribution sampling for large enough $N$, when corrected for the "offline" cost of finding the optimal $\nu_\vartheta$, the gains are more modest (Fig. 2, right). Moreover, Fig. 2 (center) shows that the nCoreSet method based on raw kernel feature maps performs better than our $\nu_\vartheta$ and empirical $\mathbb{Q}$ mixture for very small sample sizes, but eventually saturates to the mixture error level as $N$ approaches the pool size. In contrast, the error of $\nu_\vartheta$ continues to decrease.

**Alternating minimization algorithm.** We apply the alternating scheme Alg. 2 to several important operator learning problems from the physical sciences. Let $\Omega \subset \mathbb{R}^2$ denote a bounded domain

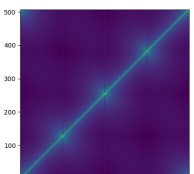 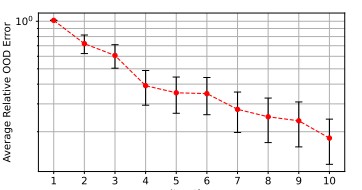 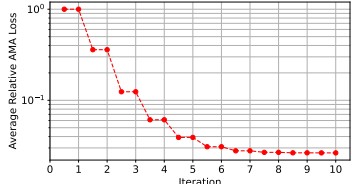

Figure 3: (Left) Matrix approximation of NtD map. (Center) After 80 independent runs, decay of the average relative OOD error of the model when trained on the optimal distribution identified at each iteration of Alg. 2; a 95% confidence interval of the true relative OOD error is provided at each iteration. (Right) Decay of average AMA loss defined in (4.7) vs. iteration relative to the same loss at initialization.

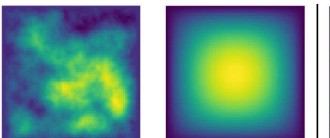 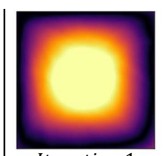 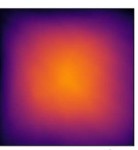 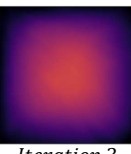 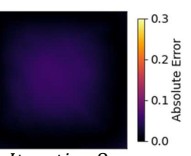

$a$  $\rightarrow$  $u$  *Iteration* 1  *Iteration* 2  *Iteration* 3  *Iteration* 8

Figure 4: The first two images show the test conductivity $a$ and true PDE solution $u$. The next four images show the absolute error $|u_{\text{pred}} - u|$ of the model trained on the training distribution from iterations 1, 2, 3, and 8.

with boundary $\partial\Omega$. Consider the elliptic PDE $\nabla \cdot (a(x)\nabla u(x)) = 0$, where $x \in \Omega$, $a = a(x) > 0$ is the conductivity, and $u = u(x)$ is the PDE solution. Given a mean-zero Neumann boundary condition (BC) $g = a\frac{\partial u}{\partial \mathbf{n}}|_{\partial\Omega}$, one can obtain a unique PDE solution $u$ and the resulting Dirichlet data $f := u|_{\partial\Omega}$. For a fixed conductivity $a$, we define a Neumann-to-Dirichlet (NtD) map $\Lambda_a^{-1} \colon g \mapsto f$, which is also used in the EIT problem (Dunlop & Stuart, 2016); see also Sec. 1. On the other hand, the map $\mathcal{G} \colon a \mapsto u$ with a fixed Neumann or Dirichlet BC (and nonzero source) is the Darcy flow parameter-to-solution operator, which plays a crucial role in geothermal energy extraction and carbon storage applications. Another important operator maps the scattering coefficient to the spatial-domain density in the radiative transport equation, which has key applications in optical tomography and atmospheric science. Last, we consider the viscous Burgers' equation with small viscosity; it models steep shock-like phenomena in fluids. Learning surrogate operators for these mappings can enable rapid simulations for efficient inference, design, and control.

*NtD map learning.* In this example, we aim to find the optimal training distribution for learning the NtD map using the alternating minimization algorithm (AMA). The domain $\Omega = (0,1)^2$ and the conductivity $a$ is fixed. The meta test distribution is a mixture of $K$ components $\mathbb{Q} = \frac{1}{K}\sum_{k=1}^{K} \delta_{\nu'_k}$, where each $\nu'_k$ corresponds to a distribution over $g$, the Neumann BC. Fig. 3 illustrates the reduction in both average relative OOD error (3.3) and relative AMA loss (4.7) over 10 iterations of Alg. 2.

*Darcy flow forward map learning.* We use AMA to identify the optimal training distribution for learning the Darcy flow parameter-to-solution map. Here, $\mathbb{Q}$ is as before except now each $\nu'_k$ represents a test distribution over the conductivity. Fig. 4 illustrates improved predictions of the neural operator trained on distributions obtained from Alg. 2 across various iterations.

*Radiative transport operator learning.* We apply a nonparametric particle-based version of AMA to find the optimal training distribution for learning the solution operator of the radiative transport equation. We vary the Knudsen number which corresponds to different physical regimes. Fig. 5 shows how the model trained on the optimal empirical measure compares to the initial model and a benchmark model trained on the empirical $\mathbb{Q}$ mixture. When the sample size is $N = 120$ particles, particle-based AMA reduces the average relative OOD error by 88% compared to its initial value for a Knudsen number of 8.

*Burgers' equation operator learning.* We apply AMA to learn the operator that maps an initial condition $u(\,\cdot\,, 0)$ to the corresponding solution $u(\,\cdot\,, 1)$ of the viscous Burgers' equation. The left and center panels of Fig. 6 illustrate an example input-output pair for this operator. We compare AMA against two pool-based active learning methods: Query-by-Committee (QbC) and CoreSet (Musekamp et al.,

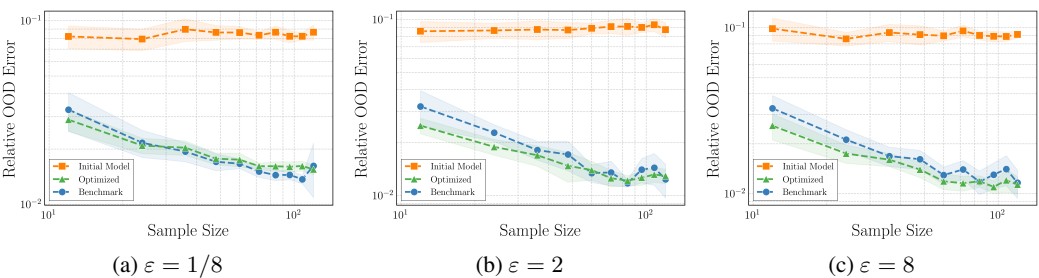

(a) $\varepsilon = 1/8$           (b) $\varepsilon = 2$           (c) $\varepsilon = 8$

Figure 5: Relative OOD error vs. sample size $N$ for learning the radiative transport solution operator. Results are shown for Knudsen numbers $\varepsilon \in \{1/8, 2, 8\}$. For each $N$, each panel displays 95% confidence intervals over 10 trials for three DeepONet models trained for 5000 epochs: the initial model, the model after particle-based AMA, and a benchmark trained the test distribution mixture $\nu_{\mathbb{Q}} := \frac{1}{3} \sum_{k=1}^{3} \nu_k'$.

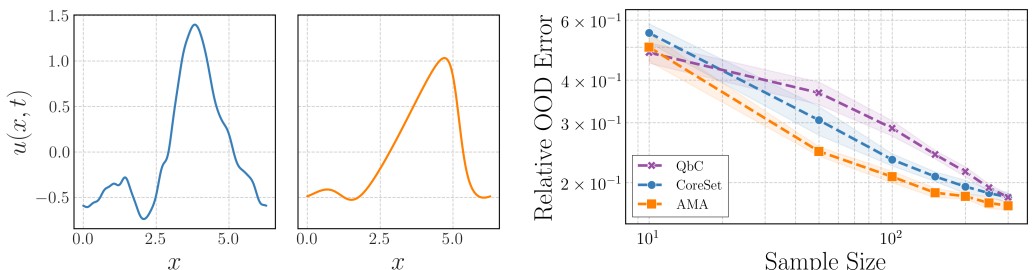

Figure 6: (Left) Example of a random initial condition $u(\,\cdot\,, 0)$ for the Burgers' equation. (Center) Corresponding solution $u(\,\cdot\,, 1)$ of the Burgers' equation at time 1 for the given initial condition. (Right) Relative OOD error vs. sample size $N$ for learning the mapping from the initial condition $u(\,\cdot\,, 0)$ to the solution $u(\,\cdot\,, 1)$. After computing the optimal training distribution using a fixed data pool, the model labeled "AMA" is trained on $N$ samples drawn from this distribution. This model is compared with two models trained using pool-based active learning methods, QbC and CoreSet. All models share the same DeepONet architecture and differ only in the training data received. The experiment is repeated 10 times, and the plot shows the 95% confidence interval for the expected relative OOD error.

2025). Using a fixed pool, we first compute the optimal training distribution from AMA and then draw $N$ samples from it to train the DeepONet model. The same pool is used to obtain models trained with each active learning strategy. The right panel of Fig. 6 reports the relative OOD performance as $N$ increases. The AMA-trained model performs comparably to the active learning baselines and, in the mid-range of sample sizes, exceeds their performance. When the sample size is small, all methods begin with randomly selected points, leading to similar performance. For large sample sizes, the training sets seem to overlap, causing the methods to converge. In the intermediate regime, however, our model consistently outperforms the other two approaches. Moreover, AMA provides the additional advantage of generating arbitrarily many high-quality training samples beyond the fixed pool, yielding not only a stronger learned operator but also a robust training distribution.

## 6 CONCLUSION

This paper shows how optimizing training distributions using theoretically founded bilevel and alternating optimization algorithms produces more accurate and robust models for function approximation and PDE operator learning. Numerical experiments instate the methods with parametric distribution families and nonparametric particle-based discretizations of Wasserstein gradient flows. Though promising, the proposed approach is limited by distribution family expressivity, global optimization, and line search steps. Future work will address these limitations by incorporating transport maps and importance reweighting techniques. Understanding cost versus accuracy trade-offs for sample allocation and iteration complexity of the methodology is another important future challenge.

REPRODUCIBILITY STATEMENT

We have taken several steps to ensure the reproducibility of our results. All proposed algorithms are described in detail in the main text, with pseudocode and additional implementation and discretization details provided in **SM** D, E, and F. Complete proofs of all theoretical results from the main text, together with explicit statements of assumptions, interpretations, and discussion, are included in **SM** C and D. The function approximation and PDE benchmarks are introduced in the main text, with full specifications of parameter ranges, boundary conditions, and discretizations in **SM** E and F. An anonymous numerical implementation, including code for training distribution optimization, model training, and evaluation, is provided in the supplementary material as a zip file. Hyperparameters and experimental settings are listed in **SM** F and in configuration files within the code repository. Together, these resources should enable full replication and extension of our results.

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

**Supplementary Material for:**
Learning where to learn:
Training data distribution optimization for scientific machine learning

## A  EXPANDED OVERVIEW OF RELATED WORK

Beyond the SciML literature reviewed in Sec. 1, our work also sits alongside research on learning under distribution shift (Mallinar et al., 2024). This includes meta-learning, domain generalization and adaptation, and active learning and optimal experimental design (OED). We now explain these connections.

Meta-learning aims to learn models or priors that adapt rapidly across tasks drawn from a task distribution (Hospedales et al., 2021; Mouli et al., 2023). While related in spirit, meta-learning typically treats the data-generating distribution per task as given; in contrast, we *optimize* the training distribution itself to improve average performance across deployment regimes. Both frameworks involve a two-level structure: an inner optimization that optimizes the model over a training dataset, and an outer optimization that adjusts a higher-level quantity beyond model parameters, guided by generalization performance. In optimization-based meta-learning methods, the outer loop adjusts the model initialization or update rules, allowing for rapid adaptation across new tasks. On the other hand, the outer loop in our work optimizes the training data distribution.

Domain generalization seeks representations that transfer across multiple source domains without access to target data (Wang et al., 2022a; Deng et al., 2024; Gagnon-Audet et al., 2023; Eastwood et al., 2022; Singh et al., 2024; Wei et al., 2022). Instead, domain adaptation leverages (often unlabeled) target data to reduce source-target discrepancy (Fang et al., 2023). Both paradigms assume a fixed training set and focus on achieving robustness during training. Common benchmarks for unsupervised domain adaptation only assume access to samples from a single target domain ($\nu'$ in $\mathbb{Q} = \delta_{\nu'}$) and do not have the ability to generate new labels (Setinek et al., 2025); this setting is orthogonal to our work. Robust domain generalization methods, such as distributionally robust optimization and invariant risk minimization (Lin et al., 2022a; Rahimian & Mehrotra, 2022; Arjovsky et al., 2019; Kamath et al., 2021; Lin et al., 2022b; Zhou et al., 2022), modify objectives or constraints to ensure stability across domains given the training data. In contrast, we *design* the training distribution *before* learning: we retain standard training objectives but *actively shape* where data are drawn for improved sample complexity and robustness to distribution shift in regression problems. We do so with bilevel (or alternating) optimization in the space of probability measures, guided by theoretical bounds. In particular, the minimizing training data distribution depends not just on the chosen model architecture and the accuracy metric, but also on the target map itself.

The perspective presented in our paper naturally connects to active learning and OED. Active learning selects informative samples from a given data pool to improve label efficiency (Deng et al., 2023; Jain et al., 2025; Musekamp et al., 2025; Settles, 2009; Yang et al., 2023); however, it typically operates *reactively* over a fixed unlabeled set during training. As a result, the support of the underlying distribution remains fixed as well. We instead study the *a priori* optimization of the sampling distribution itself. Like our work, there are active learning algorithms that are adaptive to labels (Zhu et al., 2019). However, this approach is pool-based, involves joint optimization instead of bilevel optimization, applies only to classification instead of regression, and adopts a worst-case instead of average-case perspective. On the other hand, OED explicitly reasons about where to make measurements, often through A- or D-optimality or Bayesian and sequential criteria (Atkinson et al., 2007; Fedorov, 2010; Foster et al., 2019; Huan et al., 2024). The classical OED problem emphasizes parameter inference and uncertainty reduction. In contrast, we target distributions that minimize the average OOD prediction error for function and operator learning, a design principle not commonly prioritized in OED. Nevertheless, our work shares similarities with modern measure-centric OED (Hellmuth et al., 2025; Jin et al., 2024a), in which the design variable is a probability measure over the design space instead of a fixed number of sample points. In the nonlinear case, measure-based OED requires the formulation of a bilevel optimization problem (Jin et al., 2024b). The difference is that our inner loop involves the supervised training of a machine learning model, while for OED it typically involves solving a PDE-constrained inverse problem.

Other related work includes ridge leverage score and randomly pivoted Cholesky sampling for kernel regression (Rudi et al., 2018; Chen et al., 2025b) as well as Bayesian optimization for sequential

point-by-point dataset updates (Shahriari et al., 2015). These techniques provide principled but typically label-independent sampling rules that do not depend on the target map to be learned. Instead, the present paper proposes adaptive, target-dependent, and supervised distribution design methods.

## B  ADDITIONAL MATHEMATICAL BACKGROUND

This appendix provides more mathematical background for the present work.

We begin with two comments about Gaussian distributions on Hilbert spaces. First, we remark that the 2-Wasserstein metric between Gaussian measures has a closed-form expression.

**Example** B.1 (2-Wasserstein metric between Gaussians). Let $\mu = \mathcal{N}(m_\mu, \mathcal{C}_\mu)$ and $\nu = \mathcal{N}(m_\nu, \mathcal{C}_\nu)$ be Gaussian measures on a real separable Hilbert space $(H, \langle \cdot, \cdot \rangle, \| \cdot \|)$. Then Gelbrich (1990) gives

$$\mathsf{W}_2^2(\mu, \nu) = \|m_\mu - m_\nu\|^2 + \mathrm{tr}\Big(\mathcal{C}_\mu + \mathcal{C}_\nu - 2\big(\mathcal{C}_\mu^{1/2}\mathcal{C}_\nu\mathcal{C}_\mu^{1/2}\big)^{1/2}\Big). \tag{B.1}$$

The square root in (B.1) is the unique self-adjoint positive semidefinite (PSD) square root of a self-adjoint PSD linear operator. The trace of an operator $A \colon H \to H$ is given by $\mathrm{tr}(A) = \sum_{j \in \mathbb{N}} \langle e_j, A e_j \rangle$ for any orthonormal basis $\{e_j\}_{j \in \mathbb{N}}$ of $H$. If $A$ is PSD, then we can also write $\mathrm{tr}(A) = \|A^{1/2}\|_{\mathrm{HS}}^2$, where $B \mapsto \|B\|_{\mathrm{HS}}^2 := \sum_{j \in \mathbb{N}} \|B e_j\|^2$ is the squared Hilbert–Schmidt norm.

We frequently invoke (B.1) in this paper. When $H$ is infinite-dimensional, the *Karhunen–Loève expansion* (KLE) is often useful in calculations (Stuart, 2010, Thm. 6.19, p. 533).

**Lemma B.2** (KLE of Gaussian measure). *Let $m \in (H, \langle \cdot, \cdot \rangle, \| \cdot \|)$ and $\mathcal{C} \colon H \to H$ be self-adjoint, PSD, and trace-class. Let $\{\lambda_j\}_{j \in \mathbb{N}} \subseteq \mathbb{R}_{\geq 0}$ be a non-increasing rearrangement of the eigenvalues of $\mathcal{C}$ and $\{\phi_j\}_{j \in \mathbb{N}} \subseteq H$ be the corresponding orthonormal set of eigenvectors. If $\{\xi_j\}_{j \in \mathbb{N}}$ is an independent and identically distributed sequence with $\xi_1 \sim \mathcal{N}(0, 1)$, then*

$$\mathrm{Law}\bigg(m + \sum_{j \in \mathbb{N}} \sqrt{\lambda_j}\xi_j\phi_j\bigg) = \mathcal{N}(m, \mathcal{C}). \tag{B.2}$$

Generalizing Lem. B.2, there exist Karhunen–Loève expansions of non-Gaussian measures. In this case, the $\{\xi_j\}_{j \in \mathbb{N}}$ are non-Gaussian, have mean zero and variance one, and are only pairwise uncorrelated instead of independent and identically distributed (i.i.d.); still, the KLE "diagonalizes" the probability measure.

Next, we turn our attention to general probability measures on abstract spaces. This allows us to define probability measures *over* the space of probability measures.

**Definition B.3** (spaces of probability measures). Let $(X, \mathcal{B})$ be a measurable space. The space of probability measures on $X$ is defined by

$$\mathscr{P}(X) := \{\mu \colon \mathcal{B} \to [0, 1] \,|\, \mu \text{ is a probability measure}\}.$$

It is equipped with the $\sigma$-algebra $\mathcal{B}(\mathscr{P}(X))$ generated by the evaluation maps $e_A \colon \mathscr{P}(X) \to [0, 1]$ given by $e_A(\mu) := \mu(A)$ for all $A \in \mathcal{B}$. Similarly, the space of probability measures on $\mathscr{P}(X)$ is

$$\mathscr{P}\big(\mathscr{P}(X)\big) := \big\{\mathbb{Q} \colon \mathcal{B}\big(\mathscr{P}(X)\big) \to [0, 1] \,\big|\, \mathbb{Q} \text{ is a probability measure}\big\}.$$

Now suppose that $(\mathscr{P}(X), \mathsf{d})$ is a metric space. For $p \in [1, \infty)$, we say that $\mathbb{Q} \in \mathscr{P}_p(\mathscr{P}(X))$ if $\mathbb{Q} \in \mathscr{P}(\mathscr{P}(X))$ and $\mathbb{E}_{\mu \sim \mathbb{Q}}[\mathsf{d}^p(\mu, \mu_0)] < \infty$ for some (and hence for all) $\mu_0 \in \mathscr{P}(X)$.

Def. B.3 can be related to the notion of random measure from Def. 2.2 as follows. Let $(\Omega, \mathcal{F}, \mathbb{P})$ be a probability space. Recall that a random probability measure on $X$ is a measurable map

$$\Xi \colon (\Omega, \mathcal{F}) \to \big(\mathscr{P}(X), \mathcal{B}(\mathscr{P}(X))\big).$$

Then the *law* (or *distribution*) of $\Xi$ is the pushforward measure $\mathrm{Law}(\Xi) \in \mathscr{P}(\mathscr{P}(X))$ defined by

$$\mathrm{Law}(\Xi)(E) := \mathbb{P}\big(\{\omega \in \Omega \,|\, \Xi(\omega) \in E\}\big) \quad \text{for all} \quad E \in \mathcal{B}\big(\mathscr{P}(X)\big).$$

In this paper, we typically do not distinguish between a random measure and its law.[1]

We now provide three examples of random measures.

---

[1]Both concepts are closely related to *Markov kernels* (Kallenberg, 2017).

**Example B.4** (empirical measure). Given i.i.d. random variables $X_1, \ldots, X_N$ taking values in measurable space $(M, \mathcal{B})$ with $X_1 \sim \mu$, the *empirical measure* is the random probability measure $\mu^{(N)} := \frac{1}{N} \sum_{n=1}^{N} \delta_{X_n}$, where $\delta_x \in \mathscr{P}(M)$ is the Dirac measure at $x \in M$. For each measurable set $A \in \mathcal{B}$, the number $\mu^{(N)}(A) \in [0, 1]$ counts the fraction of samples that fall in $A$. The measure $\mu^{(N)}$ is random because the samples $\{X_n\}_{n=1}^{N}$ are random.

**Example B.5** (Dirichlet process). Let $\alpha$ be a finite nonnegative measure on $(M, \mathcal{B})$. A *Dirichlet process* $\mu \sim \mathrm{DP}(\alpha)$ is a random probability measure such that for every $k$ and every measurable partition $\{A_1, \ldots, A_k\}$ of $M$, it holds that

$$\big(\mu(A_1), \ldots, \mu(A_k)\big) \sim \mathsf{Dirichlet}\big(\alpha(A_1), \ldots, \alpha(A_k)\big).$$

**Example B.6** (random Gaussian measures). In the infinite-dimensional state space setting, a canonical example is that of a random Gaussian measure defined through the KLE (B.2) of its mean and covariance. Let $(H, \langle \cdot, \cdot \rangle, \|\cdot\|)$ be infinite-dimensional, $\{\phi_j\}$ be an orthonormal basis of $H$, and $\{\lambda_j\} \subset \mathbb{R}_{\geq 0}$ and $\{\sigma_{ij}\} \subset \mathbb{R}_{\geq 0}$ be summable. Define the random measure $\mathcal{N}(m, \mathcal{C})$ by

$$m \overset{\mathrm{Law}}{:=} \sum_{j \in \mathbb{N}} \sqrt{\lambda_j} \xi_j \phi_j \quad \text{and} \quad \mathcal{C} := \mathcal{R}^* \mathcal{R}, \quad \text{where} \quad \mathcal{R} \overset{\mathrm{Law}}{:=} \sum_{i \in \mathbb{N}} \sum_{j \in \mathbb{N}} \sqrt{\sigma_{ij}} \zeta_{ij} \phi_i \otimes \phi_j. \tag{B.3}$$

The $\{\xi_j\}$ and $\{\zeta_{ij}\}$ are zero mean, unit variance, pairwise-uncorrelated real random variables. The outer product in (B.3) is defined by $(a \otimes b)c := \langle b, c \rangle a$. Since $\mathcal{R}$ is a KLE in the space of Hilbert–Schmidt operators on $H$, it follows that $\mathcal{C} = \mathcal{R}^* \mathcal{R}$ is a valid self-adjoint PSD trace-class covariance operator with probability one. Since $\{\phi_j\}$ is fixed, the degrees of freedom in $\mathbb{Q} := \mathrm{Law}(\mathcal{N}(m, \mathcal{C}))$ are the choice of eigenvalues $\{\lambda_j\}$ and $\{\sigma_{ij}\}$ and the distribution of the coefficients $\{\xi_j\}$ and $\{\zeta_{ij}\}$. Typically the coefficients are chosen i.i.d. from some common distribution such as the standard normal distribution $\mathcal{N}(0, 1)$ or the uniform distribution on the interval $[-\sqrt{3}, \sqrt{3}]$. The preceding construction of $\mathcal{N}(m, \mathcal{C})$ immediately extends to mixtures of Gaussian measures.

To conclude **SM** B, we connect back to the barycenters described in Rmk. 4.8.

**Definition B.7** (Wasserstein barycenter). Fix $p \in [1, \infty)$. Given a probability measure $\mathbb{Q} \in \mathscr{P}_p(\mathscr{P}_p(\mathcal{U}))$ and a set $\mathsf{P} \subseteq \mathscr{P}_p(\mathcal{U})$, a $(p, \mathsf{P})$-Wasserstein barycenter of $\mathbb{Q}$ is any solution of

$$\inf_{\mu \in \mathsf{P}} \mathbb{E}_{\nu \sim \mathbb{Q}} \mathsf{W}_p^p(\mu, \nu). \tag{B.4}$$

If $\mathsf{P} = \mathscr{P}_p(\mathcal{U})$, then we simply write $p$-Wasserstein barycenter for brevity.

When $\mathbb{Q}$ is supported on Gaussian measures, the 2-Wasserstein barycenter of $\mathbb{Q}$ can be characterized.

**Example B.8** (Gaussian barycenter). Let $\mathbb{Q} := \mathrm{Law}(\mathcal{N}(m, \mathcal{C})) \in \mathscr{P}_2(\mathscr{P}_2(\mathcal{U}))$, where $m \in \mathcal{U}$ and $\mathcal{C} : \mathcal{U} \to \mathcal{U}$ are the random mean vector and random covariance operator of a random Gaussian measure. Then the solution $\overline{\mu}$ to $\inf_{\mu \in \mathscr{P}_2(\mathcal{U})} \mathbb{E}[\mathsf{W}_2^2(\mu, \mathcal{N}(m, \mathcal{C}))]$ is the Gaussian $\overline{\mu} = \mathcal{N}(\mathbb{E}[m], \overline{\mathcal{C}})$ (Álvarez-Esteban et al., 2016, Thm. 2.4), where $\overline{\mathcal{C}}$ is a non-zero PSD solution to the nonlinear equation

$$\overline{\mathcal{C}} = \mathbb{E}\left[\left(\overline{\mathcal{C}}^{1/2} \mathcal{C} \overline{\mathcal{C}}^{1/2}\right)^{1/2}\right]. \tag{B.5}$$

One can replace the expectation with an empirical sum and employ fixed-point iterations to approximately solve (B.5) for $\overline{\mathcal{C}}$.

# C DETAILS FOR SECTION 3: LIPSCHITZ THEORY FOR OUT-OF-DISTRIBUTION ERROR BOUNDS

This appendix provides the proof of Prop. 3.1 and expands on Cor. 3.2 with Ex. C.1. We begin with the proof.

*Proof of Prop. 3.1.* We begin by proving the first assertion, which is similar to the result and proof of Benitez et al. (2024, Lem. 7.2, p. 23). Let $\phi(u) := \|\mathcal{G}_1(u) - \mathcal{G}_2(u)\|$. For any $u$ and $u'$, the triangle inequality yields

$$\begin{aligned}
|\phi(u) - \phi(u')| &\leq \|\mathcal{G}_1(u) - \mathcal{G}_2(u) - (\mathcal{G}_1(u') - \mathcal{G}_2(u'))\| \\
&\leq \|\mathcal{G}_2(u) - \mathcal{G}_2(u')\| + \|\mathcal{G}_1(u) - \mathcal{G}_1(u'))\| \\
&\leq \big(\mathrm{Lip}(\mathcal{G}_1) + \mathrm{Lip}(\mathcal{G}_2)\big) \|u - u'\|.
\end{aligned}$$

Thus, $\mathrm{Lip}(\phi) \le \mathrm{Lip}(\mathcal{G}_1) + \mathrm{Lip}(\mathcal{G}_2) =: C$. By Kantorovich–Rubenstein duality, $\mathbb{E}_{u' \sim \nu'} \phi(u') - \mathbb{E}_{u \sim \nu} \phi(u) \le C \mathsf{W}_1(\nu, \nu')$ as asserted.

For the $\mathsf{W}_2$ result, note that $\|\mathcal{G}_i(u)\| \le \mathrm{Lip}(\mathcal{G}_i)\|u\| + \|\mathcal{G}_i(0)\|$. This and the previous estimate gives

$$
\begin{aligned}
\phi^2(u') - \phi^2(u) &= \big(\phi(u') + \phi(u)\big)\big(\phi(u') - \phi(u)\big) \\
&\le \big(\|\mathcal{G}_1(u)\| + \|\mathcal{G}_1(u')\| + \|\mathcal{G}_2(u)\| + \|\mathcal{G}_2(u')\|\big) C \|u - u'\| \\
&\le \big(C\|u\| + C\|u'\| + 2\|\mathcal{G}_1(0)\| + 2\|\mathcal{G}_2(0)\|\big) C \|u - u'\| \,.
\end{aligned}
$$

Let $\pi$ be the $\mathsf{W}_2$-optimal coupling of $\mu$ and $\mu'$. By the Cauchy–Schwarz inequality, the expectation $\mathbb{E}_{u' \sim \mu'} \phi^2(u') - \mathbb{E}_{u \sim \mu} \phi^2(u) = \mathbb{E}_{(u,u') \sim \pi}[\phi^2(u') - \phi^2(u)]$ is bounded above by

$$
\begin{aligned}
C \sqrt{\mathbb{E}_{(u,u') \sim \pi}\big(C\|u\| + C\|u'\| + 2\|\mathcal{G}_1(0)\| + 2\|\mathcal{G}_2(0)\|\big)^2} &\sqrt{\mathbb{E}_{(u,u') \sim \pi}\|u - u'\|^2} \\
&\le C \sqrt{4C^2\big(\mathsf{m}_2(\mu) + \mathsf{m}_2(\mu')\big) + 16\|\mathcal{G}_1(0)\|^2 + 16\|\mathcal{G}_2(0)\|^2}\, \mathsf{W}_2(\mu, \mu') \,.
\end{aligned}
$$

The last line is due to $(a + b + c + d)^2 \le 4(a^2 + b^2 + c^2 + d^2)$. The second assertion is proved. $\square$

Next, we apply Cor. 3.2 in the context of Gaussian mixtures.

***Example* C.1** (Gaussian mixtures). Consider the *random* Gaussian measure $\nu' = \mathcal{N}(m, \mathcal{C}) \sim \mathbb{Q} \in \mathscr{P}_2(\mathscr{P}_2(\mathcal{U}))$, where $(m, \mathcal{C}) \sim \Pi$ and $\Pi$ is the mixing distribution over the mean (with marginal $\Pi_1$) and covariance (with marginal $\Pi_2$). Define the Gaussian mixture $\overline{\nu} := \mathbb{E}_{(m,\mathcal{C}) \sim \Pi} \mathcal{N}(m, \mathcal{C}) \in \mathscr{P}_2(\mathcal{U})$ via duality. Let $\overline{\mu} := \mathbb{E}_{(m',\mathcal{C}') \sim \Pi'} \mathcal{N}(m', \mathcal{C}')$ be another Gaussian mixture with mixing distribution $\Pi'$ (with marginals denoted by $\Pi_1'$ and $\Pi_2'$). By linearity of expectation and Fubini's theorem, $\mathscr{E}_{\mathbb{Q}}(\mathcal{G}) = \mathscr{E}_{\delta_{\overline{\nu}}}(\mathcal{G})$; that is, average-case accuracy with respect to $\mathbb{Q}$ is equivalent to accuracy with respect to the single test distribution $\overline{\nu}$; also see (D.2). This fact and Cor. 3.2 yield $\mathscr{E}_{\mathbb{Q}}(\mathcal{G}) \le \mathbb{E}_{u \sim \overline{\mu}}\|\mathcal{G}^\star(u) - \mathcal{G}(u)\|^2 + c(\mathcal{G}^\star, \mathcal{G}, \overline{\mu}, \overline{\nu})\mathsf{W}_2(\overline{\mu}, \overline{\nu})$. The factor $c(\mathcal{G}^\star, \mathcal{G}, \overline{\mu}, \overline{\nu})$ can be further simplified using the closed-form expression for second moments of Gaussian measures. For the $\mathsf{W}_2$ factor, Lem. C.2 and coupling arguments imply that

$$
\mathscr{E}_{\mathbb{Q}}(\mathcal{G}) \le \mathbb{E}_{u \sim \overline{\mu}}\big\|\mathcal{G}^\star(u) - \mathcal{G}(u)\big\|_{\mathcal{Y}}^2 + c\big(\mathcal{G}^\star, \mathcal{G}, \overline{\mu}, \overline{\nu}\big)\sqrt{\mathsf{W}_2^2(\Pi_1, \Pi_1') + \mathsf{W}_2^2(\Pi_2, \Pi_2')}\,. \tag{C.1}
$$

The two $\mathsf{W}_2$ distances between the marginals may be further bounded with (B.1) in **SM** B under additional structural assumptions on the mixing distributions; see Rmk. C.3. We often invoke this example in numerical studies with $\mathbb{Q} = \sum_{k=1}^K w_k \delta_{\nu_k'}$ (where $K \in \mathbb{N}$, i.e., a finite mixture), weights $\sum_{k=1}^K w_k = 1$, Gaussian test measures $\{\nu_k'\}$, and degenerate training mixture $\overline{\mu} := \mathcal{N}(m_0, \mathcal{C}_0)$.

The following lemma is used in Ex. C.1; it is an independently interesting result.

**Lemma C.2** (Lipschitz stability of mixtures in Wasserstein metric). *Let $p \in [1, \infty)$. If $\mathbb{Q}$ and $\mathbb{Q}'$ belong to $\mathscr{P}_p(\mathscr{P}_p(\mathcal{U}))$, then $\mathsf{W}_p(\mathbb{E}_{\mu \sim \mathbb{Q}}\, \mu, \mathbb{E}_{\nu \sim \mathbb{Q}'}\, \nu) \le \mathsf{W}_p(\mathbb{Q}, \mathbb{Q}')$.*

*Proof.* Let $\Pi \in \Gamma(\mathbb{Q}, \mathbb{Q}') \subseteq \mathscr{P}(\mathscr{P}_p(\mathcal{U}) \times \mathscr{P}_p(\mathcal{U}))$ be a $\mathsf{W}_p$-optimal coupling with respect to the ground space $(\mathscr{P}_p(\mathcal{U}), \mathsf{W}_p)$. By the joint convexity of the functional $\mathsf{W}_p^p$ and Jensen's inequality,

$$
\mathsf{W}_p^p(\mathbb{E}_{\mu \sim \mathbb{Q}}\, \mu, \mathbb{E}_{\nu \sim \mathbb{Q}'}\, \nu) = \mathsf{W}_p^p(\mathbb{E}_{(\mu,\nu) \sim \Pi}\, \mu, \mathbb{E}_{(\mu,\nu) \sim \Pi}\, \nu) \le \mathbb{E}_{(\mu,\nu) \sim \Pi}\, \mathsf{W}_p^p(\mu, \nu) = \mathsf{W}_p^p(\mathbb{Q}, \mathbb{Q}')\,.
$$

Taking $p$-th roots completes the proof. $\square$

In Lem. C.2, the Wasserstein distance $\mathsf{W}_p(\mathbb{Q}, \mathbb{Q}')$ uses $(\mathscr{P}_p(\mathcal{U}), \mathsf{W}_p)$ as the ground metric space in Def. 2.1. Thus, it is a Wasserstein metric over the Wasserstein space.

Last, we remark on structural assumptions that allow us to further bound (C.1) in Ex. C.1.

***Remark* C.3** (KLE mixing distributions). Instate the setting and notation of Ex. C.1. Furthermore, let $m \sim \Pi_1$ and $m' \sim \Pi_1'$ have centered (possibly non-Gaussian) KLEs

$$
m \overset{\mathrm{Law}}{=} \sum_{j \in \mathbb{N}} \sqrt{\lambda_j}\, \xi_j \phi_j \quad \text{and} \quad m' \overset{\mathrm{Law}}{=} \sum_{j \in \mathbb{N}} \sqrt{\lambda_j'}\, \xi_j' \phi_j\,, \quad \text{where} \quad \xi_j \overset{\mathrm{i.i.d.}}{\sim} \eta \quad \text{and} \quad \xi_j' \overset{\mathrm{i.i.d.}}{\sim} \eta'\,.
$$

In the preceding display, we further use the notation from (B.3) in Ex. B.6; thus, $\{\xi_j\}$ and $\{\xi_j'\}$ have zero mean and unit variance, with laws in $\mathscr{P}_2(\mathbb{R})$. Let $\rho \in \Gamma(\eta, \eta')$ be the $\mathsf{W}_2$-optimal coupling. Similar to Garbuno-Inigo et al. (2023, Lem. 4.1, pp. 13–14), let $\pi \in \Gamma(\Pi_1, \Pi_1')$ be a coupling with the property that $\pi = \mathrm{Law}(h, h')$, where $h = \sum_{j \in \mathbb{N}} \sqrt{\lambda_j} z_j \phi_j$ and $h' = \sum_{j \in \mathbb{N}} \sqrt{\lambda_j'} z_j' \phi_j$ in law and $(z_j, z_j') \overset{\text{i.i.d.}}{\sim} \rho$. Then by (B.1) and the triangle inequality,

$$
\begin{aligned}
\mathsf{W}_2^2(\Pi_1, \Pi_1') &\leq \mathbb{E}_{(m,m')\sim\pi} \|m - m'\|^2 \\
&\leq 2\,\mathbb{E}_\rho \sum_{j\in\mathbb{N}} \lambda_j |\xi_j - \xi_j'|^2 + 2\,\mathbb{E}_{\eta'} \sum_{j\in\mathbb{N}} |\xi_j'|^2 \left|\sqrt{\lambda_j} - \sqrt{\lambda_j'}\right|^2 \\
&= 2\mathsf{W}_2^2(\eta, \eta') \sum_{j\in\mathbb{N}} \lambda_j + 2\sum_{j\in\mathbb{N}} \left|\sqrt{\lambda_j} - \sqrt{\lambda_j'}\right|^2 .
\end{aligned}
$$

This estimate shows that $\Pi_1$ and $\Pi_1'$ are close in $\mathsf{W}_2$ if $\eta$ and $\eta'$ are close in $\mathsf{W}_2$ and $\{\sqrt{\lambda_j}\}$ and $\{\sqrt{\lambda_j'}\}$ are close in $\ell^2(\mathbb{N}; \mathbb{R})$. We conclude this remark by upper bounding the term $\mathsf{W}_2^2(\Pi_2, \Pi_2')$ appearing (C.1). Following (B.3) from Ex. B.6, suppose that $\mathcal{C} = \mathcal{R}^*\mathcal{R}$ and $\mathcal{C}' = (\mathcal{R}')^*\mathcal{R}'$, where

$$
\mathcal{R} \overset{\text{Law}}{=} \sum_{i\in\mathbb{N}}\sum_{j\in\mathbb{N}} \sqrt{\sigma_{ij}} \zeta_{ij} \phi_i \otimes \phi_j, \quad \mathcal{R}' \overset{\text{Law}}{=} \sum_{i\in\mathbb{N}}\sum_{j\in\mathbb{N}} \sqrt{\sigma_{ij}'} \zeta_{ij}' \phi_i \otimes \phi_j, \quad \zeta_{ij} \overset{\text{i.i.d.}}{\sim} \varrho, \text{ and } \zeta_{ij}' \overset{\text{i.i.d.}}{\sim} \varrho'
$$

are KLE expansions in the Hilbert space $\mathrm{HS}(\mathcal{U})$ of Hilbert–Schmidt operators from $\mathcal{U}$ into itself. Using (B.1) and Cockayne & Duncan (2021, Lem. SM1.2, p. SM2), it holds that

$$
\begin{aligned}
\mathsf{W}_2^2(\Pi_2, \Pi_2') &= \inf_{\gamma\in\Gamma(\Pi_2,\Pi_2')} \mathbb{E}_{(\mathcal{C},\mathcal{C}')\sim\gamma} \mathsf{W}_2^2\big(\mathcal{N}(0,\mathcal{C}), \mathcal{N}(0,\mathcal{C}')\big) \\
&\leq \inf_{\gamma\in\Gamma(\Pi_2,\Pi_2')} \mathbb{E}_\gamma \big\|\mathcal{R} - \mathcal{R}'\big\|_{\mathrm{HS}}^2 .
\end{aligned}
$$

Then a nearly identical coupling argument to the one used for $\mathsf{W}_2(\Pi_1, \Pi_1')$ delivers the bound

$$
\mathsf{W}_2^2(\Pi_2, \Pi_2') \leq 2\mathsf{W}_2^2(\varrho, \varrho') \sum_{i\in\mathbb{N}}\sum_{j\in\mathbb{N}} \sigma_{ij} + 2\sum_{i\in\mathbb{N}}\sum_{j\in\mathbb{N}} \left|\sqrt{\sigma_{ij}} - \sqrt{\sigma_{ij}'}\right|^2 .
$$

# D DETAILS FOR SECTION 4: PRACTICAL ALGORITHMS FOR TRAINING DISTRIBUTION DESIGN

This appendix expands on the content of Sec. 4 and provides the deferred proofs of the theoretical results from that section. It begins in **SM** D.1 by identifying a degeneracy of the bilevel problem formulation when infinite data are available. **SM** D.2 concerns the bilevel minimization algorithm from Sec. 4.1 and **SM** D.3 pertains to the alternating minimization algorithm from Sec. 4.2.

## D.1 NECESSITY OF FINITE DATA

Although Secs. 3 and 4 are formulated at the infinite data level, i.e., with full access to $\mathbb{Q}$ and the candidate training distribution $\nu$, the proposed formulation is only advantageous when $\nu$ is accessible from a finite number of samples. Of course, this is the case in practice (and moreover $\mathbb{Q}$ will often only be accessible from samples as well); recall Rem. 4.1. Our *optimize-then-discretize* approach of Secs. 3 and 4 simplifies the exposition and notation, while allowing us to derive continuum algorithms that still work at the finite data level. Regardless, it is instructive to see how the necessity of finite data emerges from a theoretical viewpoint.

To this end, recall the functional $\mathcal{E}_\mathbb{Q}$ from (3.3). Define the infinite mixture probability measure

$$
\nu_\mathbb{Q} := \mathbb{E}_{\nu'\sim\mathbb{Q}}[\nu'] \tag{D.1}
$$

via duality with bounded continuous test functions. By linearity, one can check that

$$
\mathcal{E}_\mathbb{Q}(\mathcal{G}) = \mathbb{E}_{u'\sim\nu_\mathbb{Q}} \|\mathcal{G}^\star(u') - \mathcal{G}(u')\|_\mathcal{Y}^2 . \tag{D.2}
$$

This display shows that the average OOD accuracy with respect to $\mathbb{Q}$ is equal to the standard test error with respect to the test mixture distribution $\nu_{\mathbb{Q}}$.

Now define the optimal value $V_{\mathbb{Q}}^{\star} := \inf_{\mathcal{G} \in \mathcal{H}} \mathcal{E}_{\mathbb{Q}}(\mathcal{G})$. By the definition of the trained model in (4.1), it holds that $V_{\mathbb{Q}}^{\star} = \mathcal{E}_{\mathbb{Q}}(\widehat{\mathcal{G}}^{(\nu_{\mathbb{Q}})})$. The inequality $V_{\mathbb{Q}}^{\star} \leq \inf_{\nu} \mathcal{E}_{\mathbb{Q}}(\widehat{\mathcal{G}}^{(\nu)})$ holds because $V_{\mathbb{Q}}^{\star}$ is the optimal value over all elements from $\mathcal{H}$. Moreover, $\inf_{\nu} \mathcal{E}_{\mathbb{Q}}(\widehat{\mathcal{G}}^{(\nu)}) \leq \mathcal{E}_{\mathbb{Q}}(\widehat{\mathcal{G}}^{(\nu_{\mathbb{Q}})}) = V_{\mathbb{Q}}^{\star}$ due to optimality over training distributions. We deduce that

$$\inf_{\nu} \mathcal{E}_{\mathbb{Q}}(\widehat{\mathcal{G}}^{(\nu)}) = \inf_{\mathcal{G} \in \mathcal{H}} \mathcal{E}_{\mathbb{Q}}(\mathcal{G}) . \tag{D.3}$$

That is, the optimal value of the objective over distributions versus the objective over models coincide. Write $\widehat{\nu}$ for a solution of $\inf_{\nu} \mathcal{E}_{\mathbb{Q}}(\widehat{\mathcal{G}}^{(\nu)})$ from (4.1). The preceding display then says

$$\mathcal{E}_{\mathbb{Q}}(\widehat{\mathcal{G}}^{(\widehat{\nu})}) = \mathcal{E}_{\mathbb{Q}}(\widehat{\mathcal{G}}^{(\nu_{\mathbb{Q}})}) . \tag{D.4}$$

This does not imply that $\widehat{\nu}$ equals $\nu_{\mathbb{Q}}$. Indeed, these infinite-dimensional optimization problems over possibly noncompact sets could admit multiple minima. What (D.4) does suggest is that the mixture $\nu_{\mathbb{Q}}$ (which only depends on $\mathbb{Q}$) is an equally good solution for making the OOD accuracy small compared to the adaptive measure $\widehat{\nu}$ (which depends on $\mathcal{G}^{\star}$, $\mathcal{H}$, and $\mathbb{Q}$), assuming full knowledge of $\mathbb{Q}$ and infinite training data. In this case, there is no reason to construct $\widehat{\nu}$ if $\nu_{\mathbb{Q}}$ is known.

However, for each $N$, writing $\nu^{N} = \frac{1}{N} \sum_{n=1}^{N} \delta_{u_n}$ for the empirical measure corresponding to i.i.d. samples $u_n \sim \nu$, we can instead consider

$$\widehat{\nu}_N \in \arg\min \left\{ \mathscr{E}_{\mathbb{Q}}(\widehat{\mathcal{G}}_N^{(\nu)}) \,\middle|\, \nu \in \mathsf{P} \subseteq \mathscr{P}_2(\mathcal{U}) \quad \text{and} \quad \widehat{\mathcal{G}}_N^{(\nu)} \in \arg\min_{\mathcal{G} \in \mathcal{H}} \mathbb{E}_{u \sim \nu^N} \big\| \mathcal{G}^{\star}(u) - \mathcal{G}(u) \big\|_{\mathcal{Y}}^2 \right\} \tag{D.5}$$

for some search space $\mathsf{P}$. The preceding argument leading to the degenerate behavior shown in (D.4) no longer holds for the empirical bilevel problem (D.5) due to the finite data constraint. It is now possible that models trained on samples from $\widehat{\nu}_N$ can outperform models trained on samples from the mixture $\nu_{\mathbb{Q}}$. Indeed, we observe the superiority of $\widehat{\nu}_N$ over $\nu_{\mathbb{Q}}$ even for simple Gaussian parametrizations in Secs. 5 and F.1. More generally, the *test distribution $\nu_{\mathbb{Q}}$ is not the optimal distribution* from which to generate training data for regression problems. This fact is known to the numerical analysis community (Adcock, 2024; Cohen & Migliorati, 2017), building off of variance reduction and importance sampling ideas (Agapiou et al., 2017) for Monte Carlo and quasi-Monte Carlo methods (Caflisch, 1998; Dick et al., 2013). It remains true for (linear) operator learning, where de Hoop et al. (2023) suggests that one should pick a training data distribution over input functions whose samples are rougher than function samples from the test distribution. This implies that the training distribution support should (strictly) contain the support of the test distribution. However, there are fundamental limits on how much improvement one can expect to obtain by adjusting the training data distribution (Grohs et al., 2025; Kovachki et al., 2024a).

Another practical takeaway from the preceding discussions is that if data from $\mathbb{Q}$ are very limited and fixed once and for all, then a finite sample approximation will not capture the mixture $\nu_{\mathbb{Q}}$ well. This could lead to a poor performing model when trained on the aggregate empirical mixture samples. On the other hand, the proposed approach delivers a distribution $\widehat{\nu}_N$ that ideally can be sampled from at will (leading to many more training samples, e.g., a generative model) and uses information about $\mathcal{G}^{\star}$ and $\mathcal{H}$ to compensate for limited information about $\mathbb{Q}$.

## D.2 DERIVATIONS FOR SUBSECTION 4.1: EXACT BILEVEL FORMULATION

**Proofs.** Recall the use of the adjoint method in real (i.e., $\mathcal{Y} = \mathbb{R}$) RKHS $\mathcal{H}$ with kernel $\kappa$ from Sec. 4.1. For a fixed $\nu \in \mathscr{P}_2(\mathcal{U})$, the data misfit $\Psi \colon \mathcal{H} \to \mathbb{R}$ for this $\nu$ is

$$\mathcal{G} \mapsto \Psi(\mathcal{G}) := \frac{1}{2} \| \iota_{\nu} \mathcal{G} - \mathcal{G}^{\star} \|_{L_{\nu}^2(\mathcal{U})}^2 = \mathbb{E}_{u \sim \nu} |\mathcal{G}(u) - \mathcal{G}^{\star}(u)|^2 . \tag{D.6}$$

This is the inner objective in (4.1). In the absence of RKHS norm penalization, typically one must assume the existence of minimizers of $\Psi$ (Rudi et al., 2015, Assump. 1, p. 3). The next lemma characterizes them.

**Lemma D.1** (inner minimization solution). *Fix a set $\mathsf{P} \subseteq \mathscr{P}_2(\mathcal{U})$ of probability measures. Suppose that for every $\nu \in \mathsf{P}$, there exists $\widehat{\mathcal{G}}^{(\nu)} \in \mathcal{H}$ such that $\Psi(\widehat{\mathcal{G}}^{(\nu)}) = \inf_{\mathcal{G} \in \mathcal{H}} \Psi(\mathcal{G})$, where $\Psi$ is as in (D.6). Then $\widehat{\mathcal{G}}^{(\nu)}$ satisfies the compact operator equation $\mathcal{K}_\nu \widehat{\mathcal{G}}^{(\nu)} = \iota_\nu^* \mathcal{G}^\star$ in $\mathcal{H}$.*

*Proof.* The result follows from convex optimization in separable Hilbert space. Since we assume minimizers of $\Psi$ exist, they are critical points. The Frechét derivative $D\Psi \colon \mathcal{H} \to \mathcal{H}^*$ is

$$\mathcal{G} \mapsto D\Psi(\mathcal{G}) = \langle \iota_\nu^*(\iota_\nu \mathcal{G} - \mathcal{G}^\star), \, \cdot \, \rangle_\kappa \,.$$

At a critical point $\mathcal{G}$, we have $D\Psi(\mathcal{G}) = 0$ in $\mathcal{H}^*$. This means

$$\langle \iota_\nu^*(\iota_\nu \mathcal{G} - \mathcal{G}^\star), h \rangle_\kappa = 0 \quad \text{for all} \quad h \in \mathcal{H} \,.$$

By choosing $h = \iota_\nu^*(\iota_\nu \mathcal{G} - \mathcal{G}^\star)$, it follows that $\iota_\nu^* \iota_\nu \mathcal{G} = \mathcal{K}_\nu \mathcal{G} = \iota_\nu^* \mathcal{G}^\star$ in $\mathcal{H}$ as asserted. $\qquad \square$

Next, recall the Lagrangian $\mathcal{L}$ from (4.2). We prove Lem. 4.2 on the adjoint state equation.

*Proof of Lem. 4.2.* The action of the Frechét derivative with respect to the second coordinate is

$$D_2 \mathcal{L}(\nu, \mathcal{G}, \lambda) = \mathbb{E}_{\nu' \sim \mathbb{Q}} \langle \iota_{\nu'}^*(\iota_{\nu'} \mathcal{G} - \mathcal{G}^\star), \, \cdot \, \rangle_\kappa + \langle \mathcal{K}_\nu \lambda, \, \cdot \, \rangle_\kappa \in \mathcal{H}^* \,.$$

Setting the preceding display equal to $0 \in \mathcal{H}^*$, we deduce that

$$\mathcal{K}_\nu \lambda = \mathbb{E}_{\nu' \sim \mathbb{Q}} \big[ \iota_{\nu'}^*(\mathcal{G}^\star - \iota_{\nu'} \mathcal{G}) \big] \quad \text{in} \quad \mathcal{H} \,.$$

Simplifying the right-hand side using the definition of $\iota_\nu^*$ completes the proof. $\qquad \square$

We are now in a position to prove Prop. 4.3, which computes the first variation of $\mathcal{J} \colon \nu \mapsto \mathcal{L}(\nu, \widehat{\mathcal{G}}^{(\nu)}, \lambda^{(\nu)}(\widehat{\mathcal{G}}^{(\nu)}))$ in the sense of Santambrogio (2015, Def. 7.12, p. 262).

*Proof of Prop. 4.3.* The first variation (Santambrogio, 2015, Def. 7.12, p. 262) of $\mathcal{L}$ with respect to the variable $\nu$ is given by the function

$$D_1 \mathcal{L}(\nu, \mathcal{G}, \lambda) = (\mathcal{G} - \mathcal{G}^\star) \lambda$$

that maps from $\mathcal{U}$ to $\mathbb{R}$, which is independent of $\nu$ given $\mathcal{G}$ and $\lambda$. Now let $\widehat{\mathcal{G}}^{(\nu)}$ be as in Lem. D.1 and $\lambda^{(\nu)}$ be as in Lem. 4.2. Due to the smoothness of $\mathcal{L}$ (4.2) with respect to both $\mathcal{G}$ and $\nu$—in particular, $\mathcal{L}$ is convex in $\mathcal{G}$ and linear in $\nu$—the envelope theorem (Afriat, 1971) ensures that

$$(D\mathcal{J})(\nu) = \big(D_1 \mathcal{L}\big)\big|_{(\nu, \mathcal{G}, \lambda) = (\nu, \widehat{\mathcal{G}}^{(\nu)}, \lambda^{(\nu)}(\widehat{\mathcal{G}}^{(\nu)}))} = \big(\widehat{\mathcal{G}}^{(\nu)} - \mathcal{G}^\star\big) \lambda^{(\nu)}(\widehat{\mathcal{G}}^{(\nu)})$$

as asserted. $\qquad \square$

***Remark D.2*** (Wasserstein gradient). As discussed in Rmk. 4.4, it is clear that

$$\nabla\big[(D\mathcal{J})(\nu)\big] = \big(\nabla \widehat{\mathcal{G}}^{(\nu)} - \nabla \mathcal{G}^\star\big) \lambda^{(\nu)}(\widehat{\mathcal{G}}^{(\nu)}) + \big(\widehat{\mathcal{G}}^{(\nu)} - \mathcal{G}^\star\big) \nabla\big(\lambda^{(\nu)}(\widehat{\mathcal{G}}^{(\nu)})\big)$$

is the Wasserstein gradient of $\mathcal{J}$ (Santambrogio, 2015, Chp. 8.2). Its computation requires the usual $\mathcal{U}$-gradient of the trained model $\widehat{\mathcal{G}}^{(\nu)}$, the true map $\mathcal{G}^\star$, and the optimal adjoint state $\lambda^{(\nu)}(\widehat{\mathcal{G}}^{(\nu)})$.

We now turn to the parametric training distribution setting. Let $\nu = \nu_\vartheta$, where $\vartheta \in \mathcal{V}$ and $(\mathcal{V}, \langle \cdot, \cdot \rangle_\mathcal{V}, \| \cdot \|_\mathcal{V})$ is a parameter Hilbert space as introduced in Sec. 4.1. The main result of Sec. 4.1 is Thm. 4.5. Before proving that theorem, we require the following lemma.

**Lemma D.3** (differentiation under the integral). *Let the density $p_\vartheta := d\nu_\vartheta / d\mu_0$ be such that $(u, \vartheta) \mapsto p_\vartheta(u)$ is sufficiently regular. Then for all bounded continuous functions $\phi \in C_b(\mathcal{U})$ and all $\vartheta \in \mathcal{V}$, it holds that*

$$\nabla_\vartheta \int_\mathcal{U} \phi(u) \nu_\vartheta(du) = \int_\mathcal{U} \phi(u) \big(\nabla_\vartheta \log p_\vartheta(u)\big) \nu_\vartheta(du) \,. \tag{D.7}$$

*Proof.* By the assumed smoothness and absolute continuity with respect to $\mu_0$,

$$\nabla_\vartheta \int_\mathcal{U} \phi(u) \nu_\vartheta(du) = \int_\mathcal{U} \phi(u) \nabla_\vartheta p_\vartheta(u) \mu_0(du) \,.$$

Shrinking the domain of integration to the support of $\nu_\vartheta$, multiplying and dividing the rightmost equality by $p_\vartheta(u)$, and using the logarithmic derivative formula delivers the assertion. $\qquad \square$

With the preceding lemma in hand, we are now able to prove Thm. 4.5.

*Proof of Thm. 4.5.* We use the chain rule for Frechét derivatives to compute $D\mathsf{J}\colon \mathcal{V} \to \mathcal{V}^*$ and hence the gradient $\nabla\mathsf{J}\colon \mathcal{V} \to \mathcal{V}$. Define $Z\colon \mathcal{V} \to \mathscr{P}_2(\mathcal{U})$ by $Z(\vartheta) = \nu_\vartheta$. Then Lem. D.3 and duality allow us to identify $DZ$ with the (possibly signed) $\mathcal{V}$-valued measure

$$\big(DZ(\vartheta)\big)(du) = \big(\nabla_\vartheta \log p_\vartheta(u)\big) Z(\vartheta)(du)\,.$$

We next apply the chain rule for Frechét derivatives to the composition $\mathsf{J} = \mathcal{J} \circ Z$. This yields

$$D\mathsf{J}(\vartheta) = (D\mathcal{J})(Z(\vartheta)) \circ (DZ)(\vartheta) \in \mathcal{V}^*$$

for any $\vartheta \in \mathcal{V}$. To interpret this expression, let $\vartheta' \in \mathcal{V}$ be arbitrary. Then formally,

$$\big(D\mathsf{J}(\vartheta)\big)(\vartheta') = \int_{\mathcal{U}} \big(\widehat{\mathcal{G}}^{(Z(\vartheta))}(u) - \mathcal{G}^\star(u)\big)\big(\lambda^{(Z(\vartheta))}(\widehat{\mathcal{G}}^{(Z(\vartheta))})\big)(u) \Big\langle \big(DZ(\vartheta)\big)(du), \vartheta' \Big\rangle_{\mathcal{V}}$$

$$= \Big\langle \int_{\mathcal{U}} \big(\widehat{\mathcal{G}}^{(\nu_\vartheta)}(u) - \mathcal{G}^\star(u)\big)\big(\lambda^{(\nu_\vartheta)}(\widehat{\mathcal{G}}^{(\nu_\vartheta)})\big)(u)\big(\nabla_\vartheta \log p_\vartheta(u)\big)\nu_\vartheta(du),\, \vartheta' \Big\rangle_{\mathcal{V}}\,.$$

The formula (4.4) for $\nabla\mathsf{J}$ follows. $\qquad\square$

**Discretized implementation.** A practical, discretized implementation of the parametric bilevel gradient descent scheme Alg. 1 further requires regularization of compact operator equations and finite data approximations. When the inner objective of (4.1) is approximated over a finite data set, we have the following regularized solution for the model update obtained from kernel ridge regression. The lemma uses the compact vector and matrix notation $U = (u_1, \ldots, u_N)^\top \in \mathcal{U}^N$, $\mathcal{G}^\star(U)_n = \mathcal{G}^\star(u_n)$ for every $n$, and $\kappa(U, U)_{ij} = \kappa(u_i, u_j)$ for every $i$ and $j$.

**Lemma D.4** (model update). *Fix a regularization strength $\sigma > 0$. Let $\{u_n\}_{n=1}^N \sim \nu^{\otimes N}$. Then*

$$\widehat{\mathcal{G}}_N^{(\nu,\sigma)} := \sum_{n=1}^N \kappa(\,\cdot\,, u_n)\beta_n \approx \widehat{\mathcal{G}}^{(\nu)}, \quad \text{where} \quad \beta = \big(\kappa(U, U) + N\sigma^2 I_N\big)^{-1}\mathcal{G}^\star(U) \in \mathbb{R}^N\,. \quad \text{(D.8)}$$

*Proof.* To numerically update $\mathcal{G}$ by solving $\mathcal{K}_\nu \mathcal{G} = \iota_\nu^* \mathcal{G}^\star$ in $\mathcal{H}$ for fixed $\nu$, we first regularize the inverse with a small nugget parameter $\sigma > 0$. Then we define

$$\widehat{\mathcal{G}}^{(\nu,\sigma)} := (\mathcal{K}_\nu + \sigma^2 \operatorname{Id}_{\mathcal{H}})^{-1}\iota_\nu^* \mathcal{G}^\star = \iota_\nu^*(\iota_\nu \iota_\nu^* + \sigma^2 \operatorname{Id}_{L^2})^{-1}\mathcal{G}^\star \approx \widehat{\mathcal{G}}^{(\nu)}\,.$$

The second equality is due to the usual Woodbury push-through identity for compact linear operators. Let $\nu^{(N)} := \frac{1}{N}\sum_{n=1}^N \delta_{u_n}$ be the empirical measure with $u_n \overset{\text{i.i.d.}}{\sim} \nu$. We identify $L^2_{\nu^{(N)}}$ with $\mathbb{R}^N$ equipped with the usual (i.e., unscaled) Euclidean norm. The action of $\mathcal{K}_{\nu^{(N)}}$ on a function $h \in \mathcal{H}$ is

$$\mathcal{K}_{\nu^{(N)}}h = \frac{1}{N}\sum_{n=1}^N \kappa(\,\cdot\,, u_n)h(u_n) =: \frac{1}{N}\kappa(\,\cdot\,, U)h(U) \qquad\qquad \text{(D.9)}$$

and similar for $\iota_{\nu^{(N)}}^*$ acting on $\mathbb{R}^N$. Define $\widehat{\mathcal{G}}_N^{(\nu,\sigma)} := \widehat{\mathcal{G}}^{(\nu^{(N)},\sigma)}$. By the preceding displays,

$$\widehat{\mathcal{G}}_N^{(\nu,\sigma)} = \sum_{n=1}^N \kappa(\,\cdot\,, u_n)\beta_n, \quad \text{where} \quad \beta = \big(\kappa(U, U) + N\sigma^2 I_N\big)^{-1}\mathcal{G}^\star(U)\,,$$

which recovers standard kernel ridge regression under square loss and Gaussian likelihood. This implies the asserted approximation $\widehat{\mathcal{G}}_N^{(\nu,\sigma)} \approx \widehat{\mathcal{G}}^{(\nu)}$. $\qquad\square$

In Lem. D.4, we expect that $\widehat{\mathcal{G}}_N^{(\nu,\sigma)} \to \widehat{\mathcal{G}}^{(\nu)}$ as $N \to \infty$ and $\sigma \to 0$. This lemma takes care of the finite data approximation of the model. Next, we tackle the adjoints.

When discretizing the integral in the gradient (4.4) with Monte Carlo sampling over the training data, we recognize that only the values of $\lambda = \lambda^{(\nu)}(\widehat{\mathcal{G}}^{(\nu)})$ at the training points are needed and not $\lambda \in \mathcal{H}$ itself. That is, we never need to query $\lambda$ away from the current training data samples. The next lemma exploits this fact to numerically implement the adjoint state update.

**Lemma D.5** (adjoint state update). *For $J \in \mathbb{N}$ and a sequence $\{M_j\} \subseteq \mathbb{N}$, define $\mathbb{Q}^{(J)} := \frac{1}{J} \sum_{j=1}^{J} \delta_{\mu_j^{(M_j)}}$, where $\mu_j^{(M_j)} := \frac{1}{M_j} \sum_{m=1}^{M_j} \delta_{v_m^j}$, $V^j := \{v_m^j\}_{m=1}^{M_j} \sim \mu_j^{\otimes M_j}$, and $\{\mu_j\}_{j=1}^{J} \sim \mathbb{Q}^{\otimes J}$. Then with $\widehat{\mathcal{G}}_N^{(\nu,\sigma)}$ and $U$ as in (D.8), the adjoint values $(\lambda^{(\nu)}(\widehat{\mathcal{G}}^{(\nu)}))(U) \in \mathbb{R}^N$ are approximated by*

$$\widehat{\lambda}_{N,J}^{(\nu,\sigma)}(U) := \left( \kappa(U,U) + N\sigma^2 I_N \right)^{-1} \frac{N}{J} \sum_{j=1}^{J} \frac{1}{M_j} \kappa(U, V^j)\left( \mathcal{G}^\star(V^j) - \widehat{\mathcal{G}}_N^{(\nu,\sigma)}(V^j) \right). \quad \text{(D.10)}$$

*Proof.* Let $\mu \sim \mathbb{Q}$, where $\mathbb{Q} \in \mathscr{P}_2(\mathscr{P}_2(\mathcal{U}))$ is fixed. For some $M$, let $\mu^{(M)} = \frac{1}{M} \sum_{m=1}^{M} \delta_{v_m}$ be the empirical measure with $V = \{v_m\}_{m=1}^{M} \sim \mu^{\otimes M}$. Consider the case $J = 1$ so that $\mathbb{Q}^{(1)} = \delta_{\mu^{(M)}}$; the general case for $J > 1$ as asserted in the lemma follows by linearity. With $\nu = \nu^{(N)}$ being the empirical data measure corresponding to $U \in \mathcal{U}^N$, $\mathcal{G} = \widehat{\mathcal{G}}_N^{(\nu,\sigma)}$ being the trained model from Lem. D.4, and $\mathbb{Q}^{(1)}$ in place of $\mathbb{Q}$, the adjoint equation (4.3) from Lem. 4.2 is

$$\mathcal{K}_{\nu^{(N)}} \lambda = \frac{1}{M} \sum_{m=1}^{M} \kappa(\,\cdot\,, v_m)\left( \mathcal{G}^\star(v_m) - \widehat{\mathcal{G}}_N^{(\nu,\sigma)}(v_m) \right).$$

By evaluating the preceding display at the $N$ data points $U = \{u_n\}_{n=1}^{N}$ and using (D.9), we obtain

$$\frac{1}{N} \kappa(U,U) \lambda(U) = \frac{1}{M} \kappa(U, V)\left( \mathcal{G}^\star(V) - \widehat{\mathcal{G}}_N^{(\nu,\sigma)}(V) \right).$$

This is a linear equation for the unknown $\lambda(U) \in \mathbb{R}^N$. Finally, we regularize the equation by replacing $\kappa(U,U)$ with $\kappa(U,U) + \sigma^2 I_N$. We denote the solution of the resulting system by $\widehat{\lambda}_{N,1}^{(\nu,\sigma)}(U)$, which is our desired approximation (D.10) of $(\lambda^{(\nu)}(\widehat{\mathcal{G}}^{(\nu)}))(U) \in \mathbb{R}^N$. $\qquad \square$

In Lem. D.5, the vector $\widehat{\lambda}_{N,J}^{(\nu,\sigma)}(U) \in \mathbb{R}^N$ depends on $\nu$ (through the empirical measure $\nu^{(N)} := \frac{1}{N} \sum_{n=1}^{N} \delta_{u_n}$), the regularization parameter $\sigma$, the current model $\widehat{\mathcal{G}}_N^{(\nu,\sigma)}$, and the random empirical measure $\mathbb{Q}^{(J)}$ (which itself depends on the sequence $\{M_j\}_{j=1}^{J}$). The formula (D.10) requires the inversion of the same regularized kernel matrix as in the model update (D.8). Thus, it is advantageous to pre-compute a factorization of this matrix so that it may be reused in the adjoint update step.

Finally, we perform the numerically implementable training data distribution update by inserting the approximations $\widehat{\mathcal{G}}_N^{(\nu,\sigma)}$ and $\widehat{\lambda}_{N,J}^{(\nu,\sigma)}(U)$ from Lems. D.4–D.5 and empirical measure $\nu_\vartheta^{(N)}$ into the exact gradient formula (4.4) for $\nabla \mathsf{J}$. This yields the approximate gradient

$$\widehat{\nabla \mathsf{J}}(\vartheta) := \int_{\mathcal{U}} \left( \widehat{\mathcal{G}}_N^{(\nu_\vartheta,\sigma)}(u) - \mathcal{G}^\star(u) \right) \widehat{\lambda}_{N,J}^{(\nu_\vartheta,\sigma)}(u) \left( \nabla_\vartheta \log p_\vartheta(u) \right) \nu_\vartheta^{(N)}(du), \quad \text{(D.11)}$$

where $\nu_\vartheta^{(N)}$ is the empirical measure of the same training data points used to obtain both the trained model and adjoint state vector, and we identify the vector $\widehat{\lambda}_{N,J}^{(\nu,\sigma)}(U) \in \mathbb{R}^N$ with a function $\widehat{\lambda}_{N,J}^{(\nu,\sigma)} \in L_{\nu_\vartheta^{(N)}}^2$. This leads to Alg. 3. In this practical algorithm, we allow for a varying number $N_k$ of training samples from $\nu_{\vartheta^{(k)}}$ and a varying regularization strength $\sigma_k$ at each iteration $k$ in Line 4. The integrals in (D.11) and Line 4 of Alg. 3 are with respect to the empirical measure; hence, they equal the equally-weighted average over the finite data samples that make up the empirical measure.

One limitation of the practical bilevel gradient descent scheme Alg. 3 as written is that it is negatively impacted by overfitting in the model update step, as discussed in the following remark.

***Remark* D.6** (overfitting and vanishing gradients). A drawback of approximating the integral in the exact gradient (4.4) with a Monte Carlo average over the i.i.d. training samples in (D.11) and Line 4 of Alg. 3 is that the resulting gradient approximation incorporates the training error residual. If the trained model overfits to the data, then this residual becomes extremely small, which slows down the convergence of gradient descent significantly. In the extreme case, the gradient vanishes if the model perfectly fits the training data. Thus, the proposed method as currently formulated would likely fail for highly overparametrized models. To address this problem, one solution is to hold out a "validation" subset of the training data pairs for the adjoint state and Monte Carlo integral calculations;

---

**Algorithm 3** Gradient Descent on a Parametrized Bilevel Objective: Discretized Scheme

---

1: **Initialize:** Parameter $\vartheta^{(0)}$, step sizes $\{t_k\}$, nuggets $\{\sigma_k\}$, sample sizes $\{N_k\}$, measure $\mathbb{Q}^{(J)}$
2: **for** $k = 0, 1, 2, \ldots$ **do**
3:     **Sample and label:** Construct the empirical measure $\nu_{\vartheta^{(k)}}^{(N_k)}$ and label vector $\mathcal{G}^\star(U) \in \mathbb{R}^{N_k}$
4:     **Gradient step:** Update the training distribution's parameters via

$$\vartheta^{(k+1)} = \vartheta^{(k)} - t_k \int_{\mathcal{U}} \left( \widehat{\mathcal{G}}_{N_k}^{(\nu_{\vartheta^{(k)}}, \sigma_k)}(u) - \mathcal{G}^\star(u) \right) \widehat{\lambda}_{N_k, J}^{(\nu_{\vartheta^{(k)}}, \sigma_k)}(u) \left( \nabla_\vartheta \log p_{\vartheta^{(k)}}(u) \right) \nu_{\vartheta^{(k)}}^{(N_k)}(du)$$

5:     **if** stopping criterion is met **then**
6:         Return $\vartheta^{(k+1)}$ then **break**

---

the remaining data should be used to train the model. Another approach is to normalize the size of the approximate gradient by its current magnitude. This adaptive normalization aims to avoid small gradients. In the numerical experiments of Sec. 5, we use kernel ridge regression with strictly positive regularization; this prevents interpolation of the data and helps avoid the vanishing gradient issue.

### D.3 DERIVATIONS FOR SUBSECTION 4.2: UPPER-BOUND MINIMIZATION

Recall the alternating minimization algorithm Alg. 2 from Sec. 4.2. We provide details about its numerical implementation. For gradient-based implementations, the first variation of the upper bound objective (4.8) is required. This is the content of Prop. 4.9, which we now prove.

*Proof of Prop. 4.9.* Since the first term $\int_{\mathcal{U}} f(u)\nu(du)$ in the objective $F$ in (4.8) is linear in the argument $\nu$, the first variation of that term is simply $f$ itself (Santambrogio, 2015, Def. 7.12, p. 262). The chain rule of Euclidean calculus and a similar first variation calculation delivers the second term in the asserted derivative (4.9). The third and final term arising from the chain rule requires the first variation of $\nu \mapsto \mathbb{E}_{\nu' \sim \mathbb{Q}} \mathsf{W}_2^2(\nu, \nu')$. By Santambrogio (2015, Prop. 7.17, p. 264, applied with cost $c(x, y) = \|x - y\|_{\mathcal{U}}^2/2$), this first variation equals the Kantorovich potential $2\phi_{\nu,\nu'}$, where the asserted characterization of $\phi_{\nu,\nu'}$ is due to Brenier's theorem; see Chewi (2024, Thm. 1.3.8, p. 27) or (Villani, 2021). Finally, we exchange expectation over $\mathbb{Q}$ and differentiation using dominated convergence due to the assumed regularity of the set P. $\square$

The Wasserstein gradient of $F$ is useful for interacting particle discretizations. We have the following corollary of Prop. 4.9.

**Corollary D.7** (Wasserstein gradient). *Instate the hypotheses of Prop. 4.9. The Wasserstein gradient $\nabla D F$ at $\nu \in \mathsf{P} \subseteq \mathscr{P}_2(\mathcal{U})$ is given by the function mapping any $u \in \mathcal{U}$ to the vector*

$$\left( (\nabla D F)(\nu) \right)(u) = \nabla f(u) + \sqrt{\frac{\mathbb{E}_{\nu' \sim \mathbb{Q}} \mathsf{W}_2^2(\nu, \nu')}{1 + \mathsf{m}_2(\nu)}} u + \sqrt{\frac{1 + \mathsf{m}_2(\nu)}{\mathbb{E}_{\nu' \sim \mathbb{Q}} \mathsf{W}_2^2(\nu, \nu')}} \left( u - \mathbb{E}_{\nu' \sim \mathbb{Q}} T_{\nu \to \nu'}(u) \right).$$

Alternatively, one can derive standard gradients when working with a parametrized family $\{\nu_\vartheta\}_{\vartheta \in \mathcal{V}}$ of candidate training distributions. The next lemma instantiates this idea in the specific setting of a finite-dimensional Gaussian family.

**Lemma D.8** (chain rule: Gaussians). *Instate the hypotheses of Prop. 4.9. Suppose that $\mathcal{U} \subseteq \mathbb{R}^d$. Let $\nu_\vartheta := \mathcal{N}(m_\vartheta, \mathcal{C}_\vartheta)$ and $\mathsf{F}(\vartheta) := F(\nu_\vartheta)$, where $F$ is given in (4.8). Write $p_\vartheta : \mathcal{U} \to \mathbb{R}_{\geq 0}$ for the density of $\nu_\vartheta$ with respect to any $\sigma$-finite dominating measure. Then for any parameter $\vartheta \in \mathcal{V}$, it*

*holds that*

$$\nabla \mathsf{F}(\vartheta) = \int_{\mathcal{U}} f(u) \big(\nabla_\vartheta \log p_\vartheta(u)\big) \nu_\vartheta(du)$$

$$+ \frac{1}{2} \sqrt{\frac{\mathbb{E}_{\nu' \sim \mathbb{Q}} \mathsf{W}_2^2(\nu_\vartheta, \nu')}{1 + \|m_\vartheta\|_{\mathcal{U}}^2 + \mathrm{tr}(\mathcal{C}_\vartheta)}} \int_{\mathcal{U}} \frac{\|u\|_{\mathcal{U}}^2}{2} \big(\nabla_\vartheta \log p_\vartheta(u)\big) \nu_\vartheta(du)$$

$$+ \sqrt{\frac{1 + \|m_\vartheta\|_{\mathcal{U}}^2 + \mathrm{tr}(\mathcal{C}_\vartheta)}{\mathbb{E}_{\nu' \sim \mathbb{Q}} \mathsf{W}_2^2(\nu_\vartheta, \nu')}} \int_{\mathcal{U}} \frac{1}{2} \big\langle u, (I_d - \mathbb{E}_{\mathbb{Q}}[A'_\vartheta])u \big\rangle_{\mathcal{U}} \big(\nabla_\vartheta \log p_\vartheta(u)\big) \nu_\vartheta(du)$$

$$+ \sqrt{\frac{1 + \|m_\vartheta\|_{\mathcal{U}}^2 + \mathrm{tr}(\mathcal{C}_\vartheta)}{\mathbb{E}_{\nu' \sim \mathbb{Q}} \mathsf{W}_2^2(\nu_\vartheta, \nu')}} \int_{\mathcal{U}} \big\langle u, \mathbb{E}_{\mathbb{Q}}[A'_\vartheta]m_\vartheta - \mathbb{E}_{\mathbb{Q}}[m'] \big\rangle_{\mathcal{U}} \big(\nabla_\vartheta \log p_\vartheta(u)\big) \nu_\vartheta(du),$$

*where $\nu' = \mathcal{N}(m', \mathcal{C}') \sim \mathbb{Q}$ is a random Gaussian and $A'_\vartheta := \mathcal{C}_\vartheta^{-1/2}(\mathcal{C}_\vartheta^{1/2}\mathcal{C}'\mathcal{C}_\vartheta^{1/2})^{1/2}\mathcal{C}_\vartheta^{-1/2}$.*

*Proof.* The $\mathsf{W}_2$-optimal transport map from $\nu = \mathcal{N}(m_1, \mathcal{C}_1)$ to $\nu' = \mathcal{N}(m_2, \mathcal{C}_2)$ is

$$u \mapsto T_{\nu \to \nu'}(u) := m_2 + A(u - m_1), \quad \text{where} \quad A := \mathcal{C}_1^{-1/2}(\mathcal{C}_1^{1/2}\mathcal{C}_2\mathcal{C}_1^{1/2})^{1/2}\mathcal{C}_1^{-1/2}.$$

Furthermore, $\phi_{\nu,\nu'} = \|\cdot\|_{\mathcal{U}}^2/2 - \varphi_{\nu,\nu'}$, where $\nabla \varphi_{\nu,\nu'} = T_{\nu \to \nu'}$ by Brenier's theorem (Chewi, 2024, Thm. 1.3.8, p. 27). Upon integration, we find that

$$u \mapsto \phi_{\nu,\nu'}(u) = \frac{\|u\|^2}{2} - \frac{1}{2}\langle u, Au \rangle - \langle u, m_2 - Am_1 \rangle.$$

This formula is unique up to a constant. Now let $\nu := \nu_\vartheta = \mathcal{N}(m_\vartheta, \mathcal{C}_\vartheta)$ and $\nu' = \mathcal{N}(m', \mathcal{C}') \sim \mathbb{Q}$ be as in the hypotheses of the lemma. By the chain rule for Frechét derivatives,

$$\nabla \mathsf{F}(\vartheta) = \int_{\mathcal{U}} \big(DF(\nu_\vartheta)\big)(u) \big(\nabla_\vartheta \log p_\vartheta(u)\big) \nu_\vartheta(du)$$

for each $\vartheta$. Applying (4.9) from Prop. 4.9, closed-form formulas for the second moments of Gaussian distributions, and the Fubini–Tonelli theorem to exchange integrals completes the proof. □

Lem. D.8 requires the gradient $\nabla_\vartheta \log p_\vartheta$ of the log density with respect to the parameter $\vartheta$. In general, this gradient has no closed form and must be computed numerically, e.g., with automatic differentiation. However, when only the mean of the Gaussian family is allowed to vary, the following remark discusses an analytical expression for the gradient.

**Remark D.9** (Gaussian mean parametrization). Let $\mathcal{U} \subseteq \mathbb{R}^d$. If $\vartheta = m$ so that $\nu_\vartheta = \nu_m = \mathcal{N}(m, \mathcal{C}_0)$ for fixed $\mathcal{C}_0$, then $\nabla_m \log p_m(u) = \mathcal{C}_0^{-1}(u - m)$, where $p_m$ is the Lebesgue density of $\nu_m$.

# E  OPERATOR LEARNING ARCHITECTURES

In this appendix, we provide detailed descriptions of the various deep operator learning architectures used in the paper. These are the DeepONet (**SM** E.1), NIO (**SM** E.2), and the newly proposed AMINO (**SM** E.3).

## E.1  DEEPONET: DEEP OPERATOR NETWORK

Deep operator networks (DeepONet) (Lu et al., 2021) provide a general framework for learning nonlinear operators $\mathcal{G}^\star : u \mapsto \mathcal{G}^\star(u)$ from data. A DeepONet $\mathcal{G}_\theta$ achieves this via two jointly trained subnetworks:

- **Branch network** B: takes as input the values of the input function $u$ at a fixed set of sensor locations $\{x^{(j)}\}_{j=1}^J$ and outputs a feature vector $b \in \mathbb{R}^p$ for some fixed latent dimension $p$.
- **Trunk network** T: takes as input a query point $z$ in the output domain and outputs feature vector $t(z) \in \mathbb{R}^p$.

The operator prediction at function $u$ and query point $z$ is then formed by the inner product of these features, yielding

$$\mathcal{G}_\theta(u)(z) = \left\langle \mathsf{B}\big(u(x^{(1)}), \ldots, u(x^{(J)})\big), \mathsf{T}(z) \right\rangle_{\mathbb{R}^p} = \sum_{k=1}^{p} \mathsf{B}_k\big(u(x^{(1)}), \ldots, u(x^{(J)})\big) \mathsf{T}_k(z) \,. \quad \text{(E.1)}$$

In practice, a constant bias vector is added to the preceding display. During ERM training as in (2.1), the weights $\theta$ of both neural networks $\mathsf{B}$ and $\mathsf{T}$ are optimized together to minimize the average discrepancy between $\mathcal{G}_\theta(u)(z)$ and ground truth values $\mathcal{G}^\star(u)(z)$ over a dataset of input-output function pairs. This branch-trunk architecture enables DeepONet to approximate a broad class of operators, including the Neumann-to-Dirichlet (NtD) map in EIT and the conductivity-to-solution map in Darcy flow considered in this work; see Sec. 5 and **SM** F.

### E.2 NIO: Neural Inverse Operator

Traditional approaches for constructing direct solvers for inverse problems, such as the D-bar method for EIT (Siltanen et al., 2000), have laid the foundation for direct solver approximations utilizing modern machine learning techniques. Among these, the Neural Inverse Operator (NIO) (Molinaro et al., 2023) framework stands out as a significant advancement. In the context of operator learning, many inverse problems can be formulated as a mapping from an operator $\Lambda_a$, dependent on a physical parameter $a$, to the parameter $a$ itself, i.e., $\Lambda_a \mapsto a$. In practical scenarios, the operator $\Lambda_a$ is not directly accessible; instead, we observe its action on a set of input-output function pairs $\{(f_n, g_n)\}_n$, where $g_n = \Lambda_a(f_n)$. This observation leads to a two-step problem: first, infer the operator $\Lambda_a$ from the data pairs, and second, deduce the parameter $a$ from the inferred operator. This process can be further generalized by considering the operator as a pushforward map $(\Lambda_a)_\#$, which transforms an input distribution $\mu$ of functions to an output distribution $\nu_a$ of functions, i.e., $(\Lambda_a)_\# \mu = \nu_a$; see Sec. 2 for the definition of the pushforward operation. Consequently, a measure-centric machine learning approach to solving the inverse problem can be reformulated as learning the mapping $(\mu, \nu_a) \mapsto a$ (Nelsen & Yang, 2025, Sec. 2.4.1).

Conditioned on a fixed input distribution $\mu$, NIO is designed to learn mappings from output measures $\nu_a$ to functions $a$, i.e., $\nu_a \mapsto a$, effectively relaxing the operator-to-function problem to a distribution-to-function problem. Since direct manipulation of measures is computationally infeasible, NIO circumvents this by leveraging finite data approximations. Specifically, it utilizes a composition of DeepONets and Fourier Neural Operators (FNO) (Li et al., 2021) to approximate the inverse mapping from an empirical measure/approximation $\widehat{\nu}_a$ to the underlying parameter function $a$.

### E.3 AMINO: A Measure-theoretic Inverse Neural Operator

Despite the innovative architecture of NIO, it exhibits certain limitations, particularly concerning its handling of input distributions. NIO primarily focuses on the output distribution $\nu_a$ during evaluation, neglecting variability and influence of the input distribution $\mu$. This oversight can lead to models that perform well on in-distribution (ID) data but lack robustness when faced with OOD scenarios. To address this shortcoming, we introduce *A Measure-theoretic Inverse Neural Operator* (AMINO), a framework that explicitly accounts for the input distribution in the inverse problem. By formulating the inverse task in a fully measure-theoretic setting and taking into account the input distribution, AMINO aims to improve OOD accuracy. Like NIO, AMINO combines a DeepONet and an FNO. The key distinction is that AMINO incorporates both input and output functions as concatenated pairs $\{(f_n, g_n)\}_n \overset{\text{i.i.d.}}{\sim} (\mathrm{Id}, \Lambda_a)_\# \mu$ which are passed to the branch network, as illustrated in Fig. 7. See (Nelsen & Yang, 2025, Sec. 4.2.3) for a more detailed description of NIO and AMINO.

## F Details for Section 5: Numerical Results

This appendix describes detailed experimental setups for all numerical tests from Sec. 1 and Sec. 5 in the main text. Regarding function approximation, **SM** F.1 covers the bilevel gradient descent experiments. The remaining appendices concern operator learning. **SM** F.2 details the AMINO experiments. **SM** F.3 and **SM** F.4 describe the results of the alternating minimization algorithm when applied to learning linear NtD maps and the nonlinear solution operator of Darcy flow, respectively.

Figure 7: Schematic of the AMINO architecture, a variant of NIO, which combines a DeepONet, averaging, and an FNO module. The model takes as input the paired function samples $\{(f_\ell, g_\ell)\}_{\ell=1}^L$ drawn from the joint distribution $(\mathrm{Id}, \Lambda_a)_{\#}\mu$ and outputs the target parameter $a$.

Last, **SM** F.5 presents results from applying nonparametric particle-based AMA to learn a parameter-to-solution map for the radiative transport equation.

### F.1 BILEVEL MINIMIZATION FOR FUNCTION APPROXIMATION

We now describe the setting of the numerical results reported in Sec. 5 for Alg. 1 and its numerical version Alg. 3. The goal of the experiments is to find the optimal Gaussian training distribution $\nu_\vartheta = \mathcal{N}(m, \mathcal{C}) \in \mathscr{P}_2(\mathbb{R}^d)$ with respect to the bilevel OOD objective (4.1), specialized to the task of function approximation. We optimize over the mean and covariance $\vartheta = (m, \mathcal{C})$. For all $x \in \mathbb{R}^d$, the target test functions $\mathcal{G}^\star = g_i \colon \mathbb{R}^d \to \mathbb{R}$ for $i \in \{1, 2, 3, 4\}$ that we consider are given by

$$g_1(x) = \prod_{j=1}^d \frac{|4x_j - 2| + (j-2)/2}{1 + (j-2)/2}, \tag{F.1a}$$

$$g_2(x) = 10 \sin(\pi x_1 x_2) + 20(x_3 - 1/2)^2 + 10x_4 + 5x_5, \tag{F.1b}$$

$$g_3(x) = \sqrt{(100x_1)^2 + \left(x_3(520\pi x_2 + 40\pi) - \frac{1}{(520\pi x_2 + 40\pi)(10x_4 + 1)}\right)^2}, \quad \text{and} \tag{F.1c}$$

$$g_4(x) = \sum_{l=1}^{1000} c_l \kappa(x, x_l). \tag{F.1d}$$

The first function comes from (Dunbar et al., 2025) and the middle two from (Potts & Schmischke, 2021; Saha et al., 2023). They are common benchmarks in the approximation theory literature; $g_1$ is called the Sobol G function, $g_2$ the Friedmann-1 function, and $g_3$ the Friedmann-2 function. Although these functions are typically defined on compact domains (e.g., $[0, 1]^d$ or $[-\pi, \pi]^d$), we instead consider approximating them on average with respect to Gaussian distributions (or mixtures thereof) on the whole of $\mathbb{R}^d$. The final function $g_4$ is a linear combination of kernel sections for some kernel function $\kappa \colon \mathbb{R}^d \times \mathbb{R}^d \to \mathbb{R}$. We sample (and then fix) the coefficients $c_l \overset{\text{i.i.d.}}{\sim} \mathrm{Unif}([-1, 1])$ and the centers $x_l \overset{\text{i.i.d.}}{\sim} \mathrm{Unif}([-4, 4])^{\otimes d}$. We allow the dimension $d$ to be arbitrary for all test functions; in particular, the functions $g_2$ and and $g_3$ are constant in certain directions for large enough $d$.

**Model architecture and optimization algorithm.** In all bilevel minimization experiments, we use the squared exponential kernel function $\kappa \colon \mathbb{R}^d \times \mathbb{R}^d \to \mathbb{R}_{>0}$ defined by

$$(x, x') \mapsto \kappa(x, x') := \exp\left(-\frac{\|x - x'\|_2^2}{\ell^2}\right), \tag{F.2}$$

where $\ell > 0$ is a scalar lengthscale hyperparameter. While not tuned for accuracy, we do adjust $\ell$ on the order of $O(1)$ to $O(10)$ depending on the target function $g_i$ to avoid blow up of the bilevel gradient descent iterations. In particular, we set $\ell = 1$ for $g_1$, $\ell = 3$ for $g_2$, $\ell = 2/1.1$ for $g_3$, and $\ell = 5$ for $g_4$. We also experimented with the less regular Laplace kernel $(x, x') \mapsto \exp(-\|x - x'\|_1/\ell)$; although giving similar accuracy as $\kappa$, the Laplace kernel led to slower bilevel optimization. Thus, we opted to report all results using the squared exponential kernel $\kappa$. This same kernel is used to define $g_4$ in (F.1), which belongs to the RKHS of $\kappa$ as a result. Given a candidate set of training samples, the resulting kernel method is trained on these samples via kernel ridge regression as in Lem. D.4. We solve all linear systems with lower triangular direct solves using Cholesky factors of

the regularized kernel matrices. No iterative optimization algorithms are required to train the kernel regressors. When fitting the kernel models to data outside of the bilevel gradient descent loop (see the center and right columns of Fig. 8, for instance), we set the ridge strength to be $\sigma^2 = 10^{-3}/N$, where $N$ is the current training sample size.

**Training and optimization procedure.** Moving on to describe the specific implementation details of Alg. 3, we initialize $\vartheta^{(0)} = (m_0, I_d)$. We take $m_0 = \frac{1}{2}(1, \ldots, 1)^\top \in \mathbb{R}^d$ in all setups except for the one concerning test function $g_1$, where we set $m_0 = (0, \ldots, 0)^\top$. We decrease the gradient descent step sizes $\{t_k\}$ in Alg. 3 according to a cosine annealing scheduler (Loshchilov & Hutter, 2017) with initial learning rate of $10^{-2}$ and final learning rate of 0. The regularization parameters $\{\sigma_k\}$ are also selected according to the cosine annealing schedule with initial value $10^{-3}$ and final value $10^{-7}$. Although the sequence $\{N_k\}$ of training sample sizes could also be scheduled, we opt to fix $N_k = 250$ for all $k$ in Alg. 3 for simplicity. No major differences were observed when choosing $\{N_k\}$ according to some cosine annealing schedule. The identification of more principled ways to select the sequences $\{\sigma_k\}$, $\{t_k\}$, and $\{N_k\}$ to optimally balance cost and accuracy is an important direction for future work.

For the probability measure $\mathbb{Q} \in \mathscr{P}_2(\mathscr{P}_2(\mathbb{R}^d))$ over test distributions, we choose it to be the empirical measure

$$\mathbb{Q} := \frac{1}{K} \sum_{k=1}^{K} \delta_{\nu_k'}. \tag{F.3}$$

The test measures $\nu_k' \in \mathscr{P}_2(\mathbb{R}^d)$ in (F.3) are deterministic and fixed. In applications, these could be the datasets for which we require good OOD accuracy. However, to synthetically generate the $\{\nu_k'\}$ in practice, we set

$$\nu_k' := \mathcal{N}(m_k', \mathcal{C}_k'), \quad \text{where} \quad m_k' \stackrel{\text{i.i.d.}}{\sim} \mathcal{N}(0, I_d) \quad \text{and} \quad \mathcal{C}_k' \stackrel{\text{i.i.d.}}{\sim} \mathsf{Wishart}_d(I_d, d+1) \tag{F.4}$$

for each $k = 1, \ldots, K$. Once the $K$ realizations of these distributions are generated, we view them as fixed once and for all. We assume that $K$ and $\mathbb{Q}$ are fixed, but we only have access to the empirical measure $\widehat{\mathbb{Q}}$ defined by (F.3) but with each $\nu_k'$ replaced by the empirical measure $\frac{1}{M} \sum_{m=1}^{M} \delta_{v_m^{(k)}}$ generated by samples $\{v_m^{(k)}\}_{m=1}^{M} \sim \nu_k'^{\otimes M}$. In the experiments, we set $M = 5000$ and either $K = 10$ or $K = 1$ (a single test distribution). To create a fair test set, we use a fixed 500 sample validation subset to update the adjoint state as in Lem. D.5 and to monitor the validation OOD accuracy during gradient descent. The remaining 4500 samples are used to evaluate the final OOD test accuracy of our trained models; see (F.5).

Based on the available samples $\{v_m^{(k)}\}_{1 \le m \le M, 1 \le k \le K}$ associated to the meta test distribution $\mathbb{Q}$, we construct six adaptive and nonadaptive baseline distributions for comparison:

1. (Normal) The Gaussian measure $\mathcal{N}(m_0, I_d)$ which is independent of $\mathbb{Q}$;

2. (Barycenter) The empirical $\mathsf{W}_2$ barycenter of $\mathbb{Q}$ (which is Gaussian), as described in Ex. B.8;

3. (Mixture) The non-Gaussian empirical mixture of $\mathbb{Q}$ given by $\frac{1}{K} \sum_{k=1}^{K} \mathcal{N}(\widehat{m}_k, \widehat{\mathcal{C}}_k)$, where $\widehat{m}_k$ and $\widehat{\mathcal{C}}_k$ are the empirical mean and covariance of the points $\{v_m^{(k)}\}_{m=1}^{M}$, respectively. Note that we enforce the mixture components to be Gaussian instead of empirical measures;

4. (Uniform) The $d$-dimensional uniform distribution $\mathsf{Unif}([0,1]^d) := \mathsf{Unif}([0,1])^{\otimes d}$, which is independent of $\mathbb{Q}$.

5. (nCoreSet) Nonadaptive coreset described in **SM** F.6 with features $\kappa(\,\cdot\,, v_m) \in \mathcal{H}_\kappa$ for $v_m$ belonging to the fixed pool of $500K$ validation points. New indices $m$ are selected according to the maxmin sequential update (F.40) with features as above and distance between features induced by the RKHS norm of $\kappa$. We use point-by-point updates and an initial random coreset of size one.

6. (aCoreSet) Adaptive greedy coreset described in **SM** F.6 with features $\{c_n \kappa(u_n, v_m)\}_{n=1}^{N} \in \mathbb{R}^N$ for $v_m$ belonging to the fixed pool of $500K$ validation points. The numbers $\{c_n\}$ are the kernel method coefficients trained on points $\{u_n\}$, which makes this an adaptive method. New indices $m$ are selected according to the maxmin sequential update (F.40) with features

as above and distance between features induced by the Euclidean norm on $\mathbb{R}^N$. We select new points from the pool in batches of size ten and used an initial random coreset of size six.

The pool-based active learning baselines are based on the work of Musekamp et al. (2025) and results for these methods are shown in Sec. 5 and in Fig. 10.

It remains to describe the optimization of the Gaussian family $\nu_\vartheta = \mathcal{N}(m, \mathcal{C})$ with parameter $\vartheta = (m, \mathcal{C})$. The closed form update for the mean follows from Rmk. D.9. For the covariance matrix, we optimize over the lower triangular Cholesky factor $L$ of $\mathcal{C} = LL^\top \in \mathbb{R}^{d \times d}$ instead of $\mathcal{C}$ itself. This removes redundant parameters due to the symmetry of $\mathcal{C}$. We enforce strict positive definiteness of $\mathcal{C}$ by replacing all nonpositive diagonal entries of $L$ with a threshold value of $10^{-7}$. Other than this constraint, each entry of $L$ is optimized over the whole of $\mathbb{R}$. Recalling the log density formula (4.5) from Ex. 4.6, we use automatic differentiation to compute $\nabla_L \log(p_{(m, LL^\top)})$; this factor appears in the definition of the approximate gradient (D.11) and in Alg. 3.

**Model evaluation.** We evaluate the performance of all kernel regressors $\widehat{\mathcal{G}}_N^{(\nu, \sigma)}$ regularized with strength $\sigma^2$ and trained with $N$ i.i.d. samples from $\nu$ with respect to the root relative average OOD squared error, which is defined as

$$\mathsf{Err} := \sqrt{\frac{\mathbb{E}_{\widehat{\nu}' \sim \widehat{\mathbb{Q}}} \mathbb{E}_{u' \sim \widehat{\nu}'} |\mathcal{G}^\star(u') - \widehat{\mathcal{G}}_N^{(\nu, \sigma)}(u')|^2}{\mathbb{E}_{\widehat{\nu}' \sim \widehat{\mathbb{Q}}} \mathbb{E}_{u' \sim \widehat{\nu}'} |\mathcal{G}^\star(u')|^2}} \approx \sqrt{\frac{\mathscr{E}_{\mathbb{Q}}(\widehat{\mathcal{G}}_N^{(\nu, \sigma)})}{\mathscr{E}_{\mathbb{Q}}(0)}}. \tag{F.5}$$

In the preceding display, we observe that $\mathsf{Err}$ employs a Monte Carlo discretization of the integrals appearing in $\mathscr{E}_{\mathbb{Q}}$ by averaging over the empirical meta measure $\widehat{\mathbb{Q}}$, which is composed of $K$ test measures. Each individual test measure is itself an empirical measure with 4500 atoms at points that are unseen during the adjoint state calculation. See the preceding paragraph for details about our choice of $\mathbb{Q}$. We report values of $\mathsf{Err}$ in Figs. 2, 8, 10, 12, 14 and in Tab. 1.

Supplementary numerical results are given in Figs. 8–15. Fig. 8 reports the same information that Fig. 2 does, except now for the remaining functions $\{g_i\}$. The conclusions from these numerical results are mostly the same. The left column shows the training history over 50 iterations; the "seen" curve computes the OOD error with the $\mathbb{Q}$ samples used in the bilevel algorithm, while the "unseen" curve computes the OOD error on a held-out set of $\mathbb{Q}$ samples. The third row shows a slight "generalization gap" for $\mathcal{G}^\star = g_2$ ($d = 8$). The optimized Gaussian outperforms alternative distributions except for the $g_3$ ($d = 5$) and $g_4$ cases, where the empirical $\mathbb{Q}$ mixture eventually overtakes it. Fig. 10 shows that this behavior is largely mitigated if 1000 iterations of gradient descent are used to find the optimal Gaussian instead of 50. We remark that in Fig. 2, the leftmost subplot showing the training history is designed to highlight the long-time behavior (i.e., 1000 iterations) of the bilevel algorithm. In practice, we only run the algorithm for $O(10)$ to $O(100)$ gradient descent iterations; this is reflected in the leftmost column of Figs. 8 and 12.

While the optimized Gaussian is shown to consistently outperform nonadaptive distributions when $\mathbb{Q}$ in (F.3) is composed of $K = 10$ atoms (Figs. 8 and 10) and $N$ is large enough, this is less true for a single test distribution ($K = 1$), as seen in Figs. 12 and 14 when $N$ is sufficiently large. In this case, the barycenter and mixture distributions are the same test distribution and lead to trained models with lower OOD error. When $N$ is small and many iterations are used to find the optimal Gaussian, the optimal Gaussian can achieve smaller OOD error than models trained on the test distribution. Understanding for what $\mathcal{G}^\star$, $\mathbb{Q}$, $N$, and gradient descent iteration count makes this desirable improvement possible is an interesting and important question for future work.

To understand what properties the optimal Gaussian absorbs from $\mathcal{G}^\star$ and $\mathbb{Q}$, Figs. 9, 11, 13, and 15 visualize the learned covariance matrices, barycenter covariances, and mixture covariances. For instance, the learned covariance puts high weight on the $x_3$ coordinate in the $g_2$ case (Fig. 11, rows 2–3, diagonal entry of matrix) even though the barycenter and mixture do not. This makes sense because $g_2$ is quadratic in $x_3$ and lower order in all other variables (recall (F.1)); thus, the learned covariance picks out the highest order term in the ground truth map. Similarly, for $g_3$ the learned covariance assigns high importance to the $x_2$ and $x_3$ coordinates which are also fast growing. In all cases (except for $g_4$), the learned covariance differs substantially from the mixture or test distribution's covariance. These covariance visualization figures highlight an additional benefit of our $\mathcal{G}^\star$-dependent estimators: they lead to interpretable training distributions tailored to the structure of the problem.

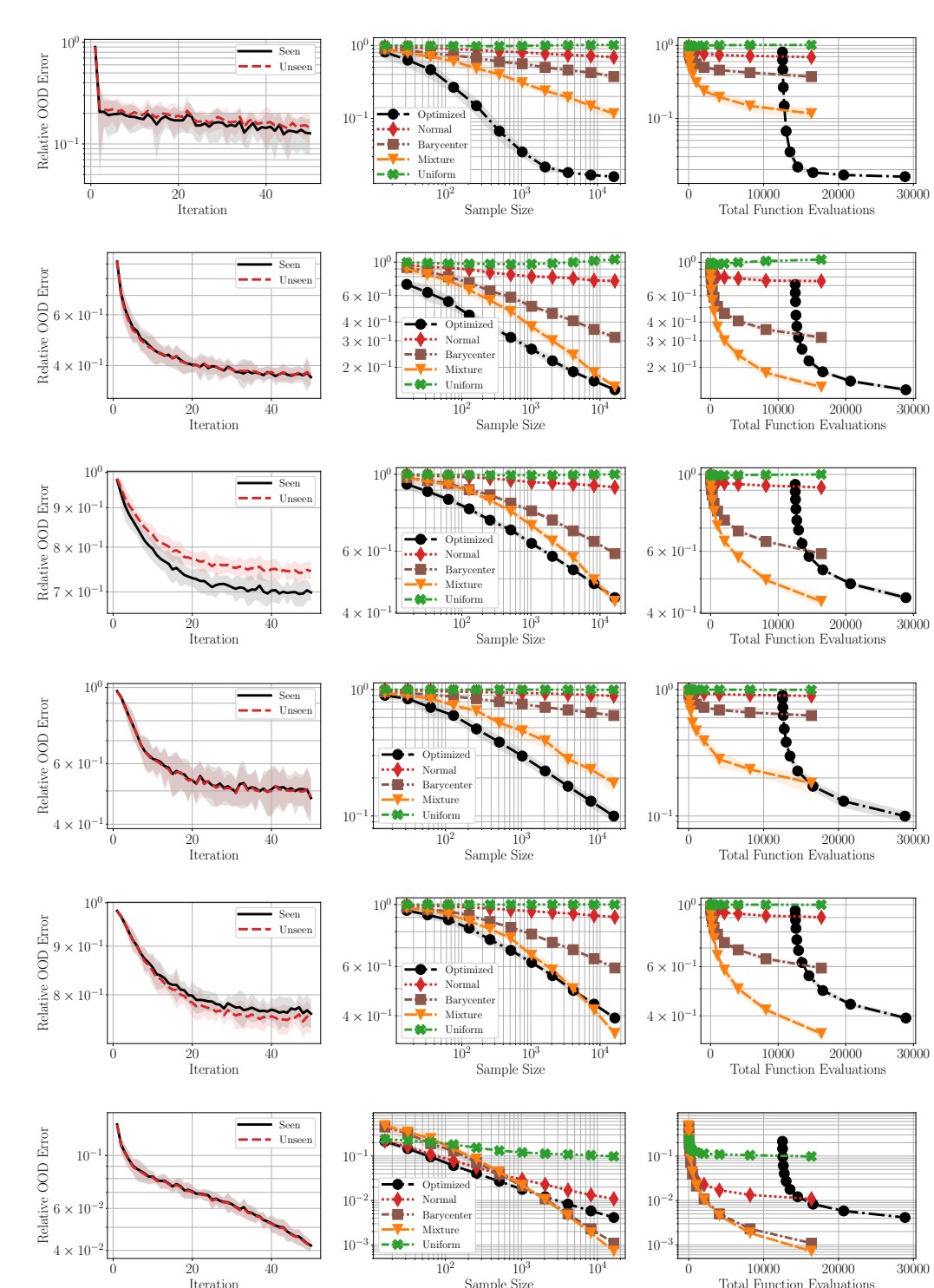

Figure 8: Bilevel Alg. 3 applied to the various ground truths $\{g_i\}$ with $K = 10$ test atoms and 50 gradient descent iterations. (Left column) Evolution of Err from (F.5) vs. iterations. (Center column) Err of model trained on $N$ samples from optimal $\nu_\vartheta$ vs. nonadaptive distributions. (Right column) Same as center, except incorporating the additional function evaluation cost incurred from Alg. 1. From top to bottom, each row represents target functions $\{g_i\}$ corresponding to those listed in each column of Tab. 1 (from left to right). Shading represents two standard deviations away from the mean Err over 10 independent runs.

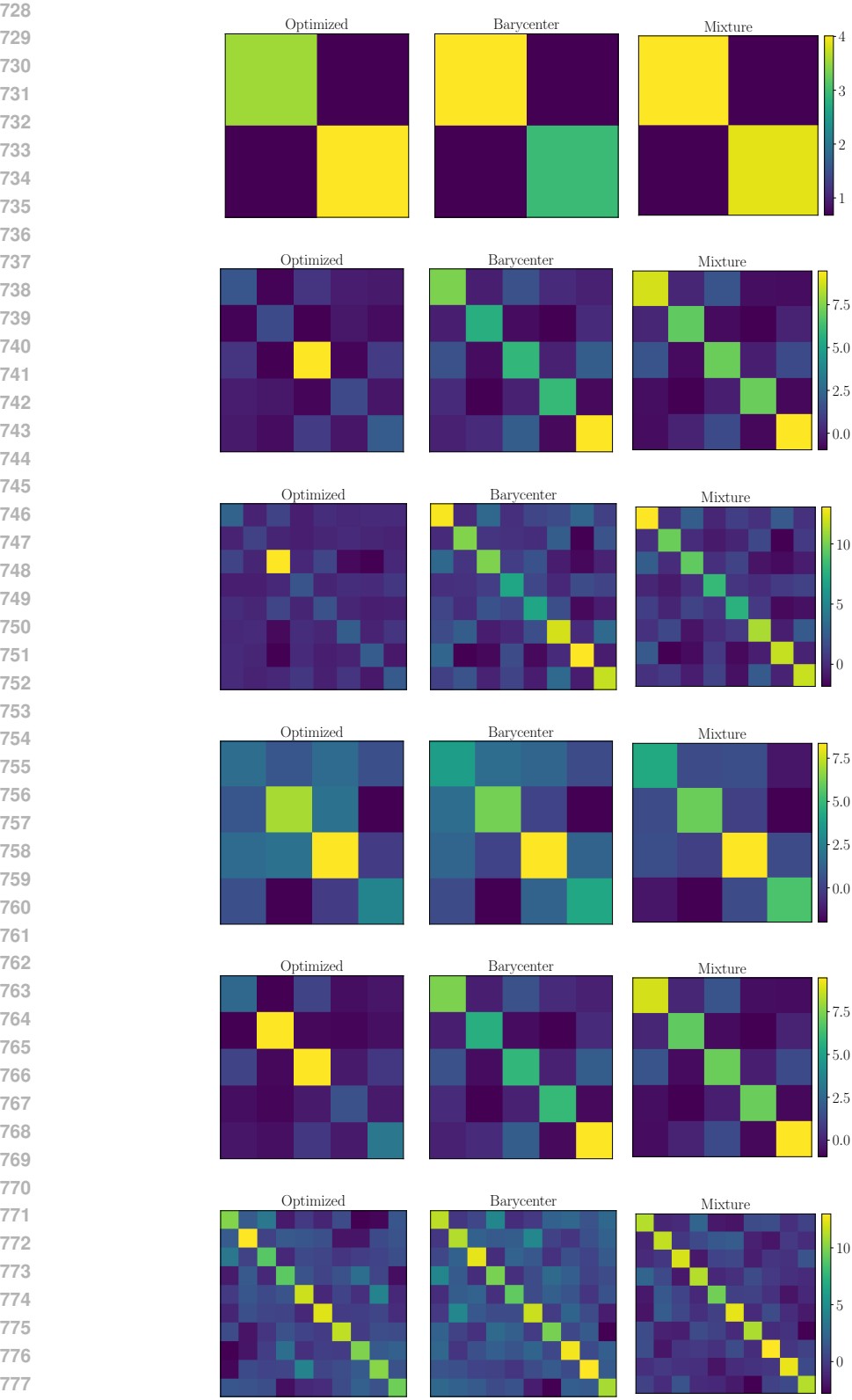

Figure 9: Bilevel Alg. 3 applied with $K = 10$ test atoms and 50 gradient descent iterations. Each column visualizes the covariance matrix of the optimized (left), empirical $\mathbb{Q}$ $W_2$ barycenter (center), and empirical $\mathbb{Q}$ mixture distributions (right). From top to bottom, each row represents target functions $\{g_i\}$ corresponding to those listed in each column of Tab. 1 (from left to right).

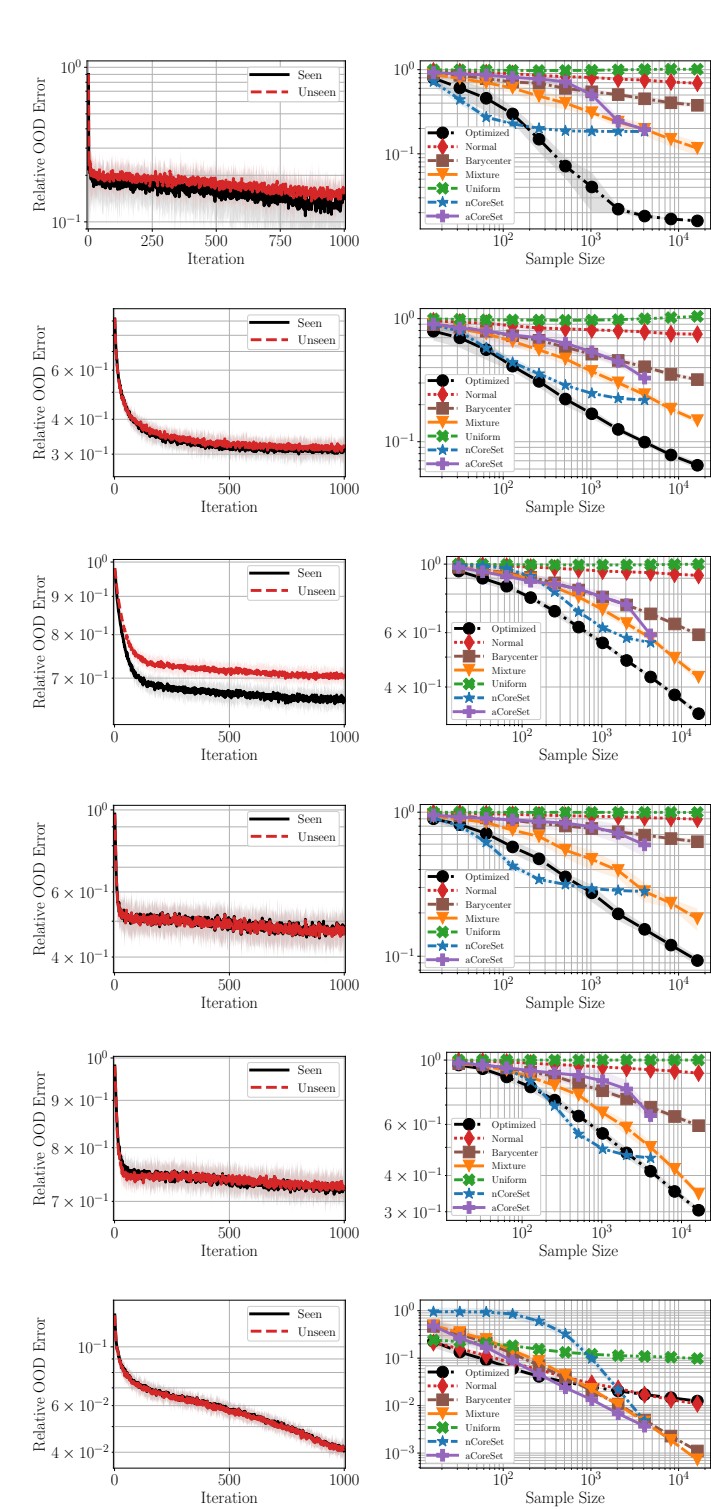

Figure 10: Bilevel Alg. 3 applied to the various ground truths $\{g_i\}$ with $K = 10$ test atoms and 1000 gradient descent iterations. (Left column) Evolution of Err from (F.5) vs. iterations. (Right column) Err of model trained on $N$ samples from optimal $\nu_\vartheta$ vs. baseline distributions. From top to bottom, each row represents target functions $\{g_i\}$ corresponding to those listed in each column of Tab. 1 (from left to right). Shading represents two standard deviations away from the mean Err over 10 independent runs.

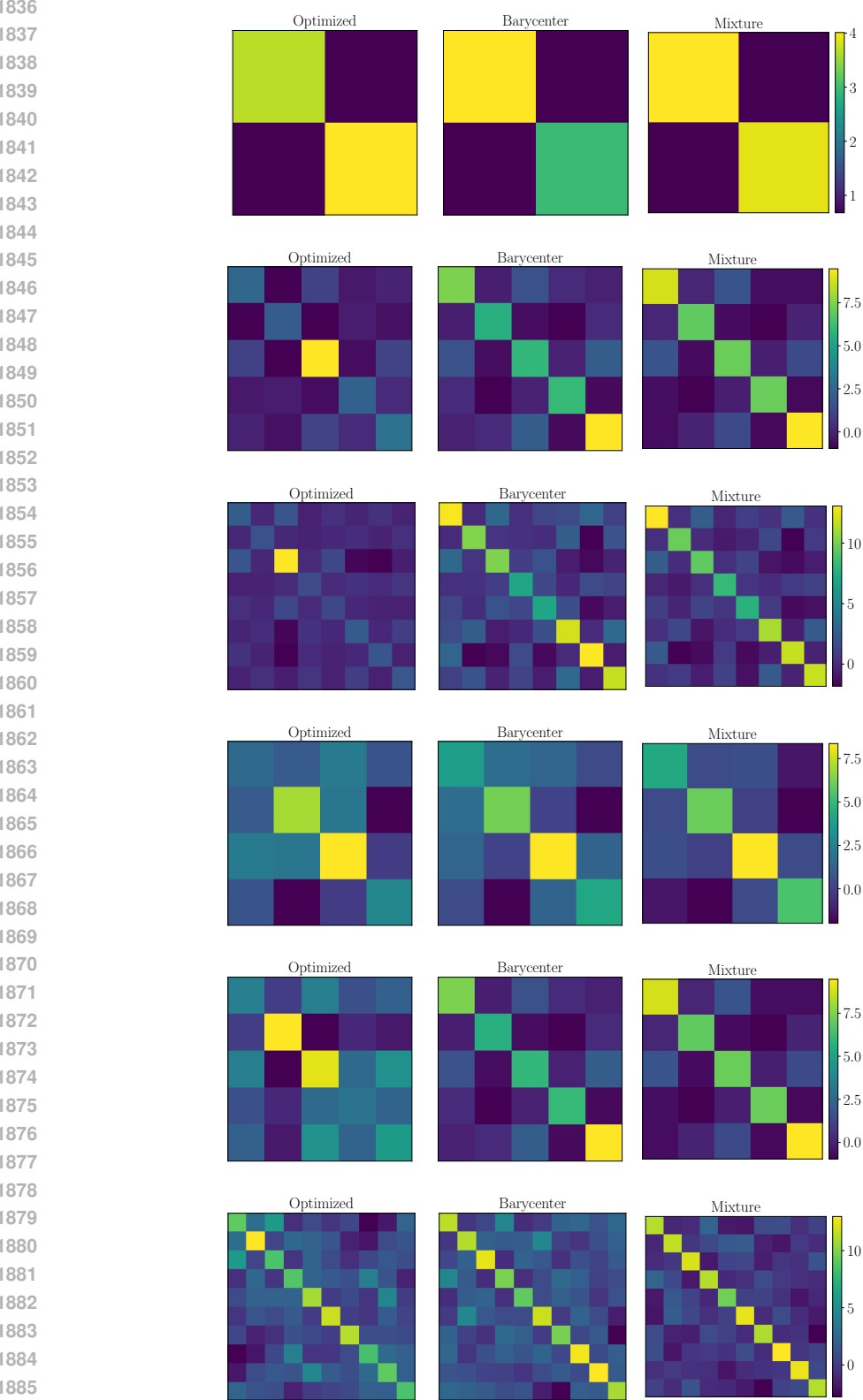

Figure 11: Bilevel Alg. 3 applied with $K = 10$ test atoms and 1000 gradient descent iterations. Each column visualizes the covariance matrix of the optimized (left), empirical $\mathbb{Q}$ $\mathsf{W}_2$ barycenter (center), and empirical $\mathbb{Q}$ mixture distributions (right). From top to bottom, each row represents target functions $\{g_i\}$ corresponding to those listed in each column of Tab. 1 (from left to right).

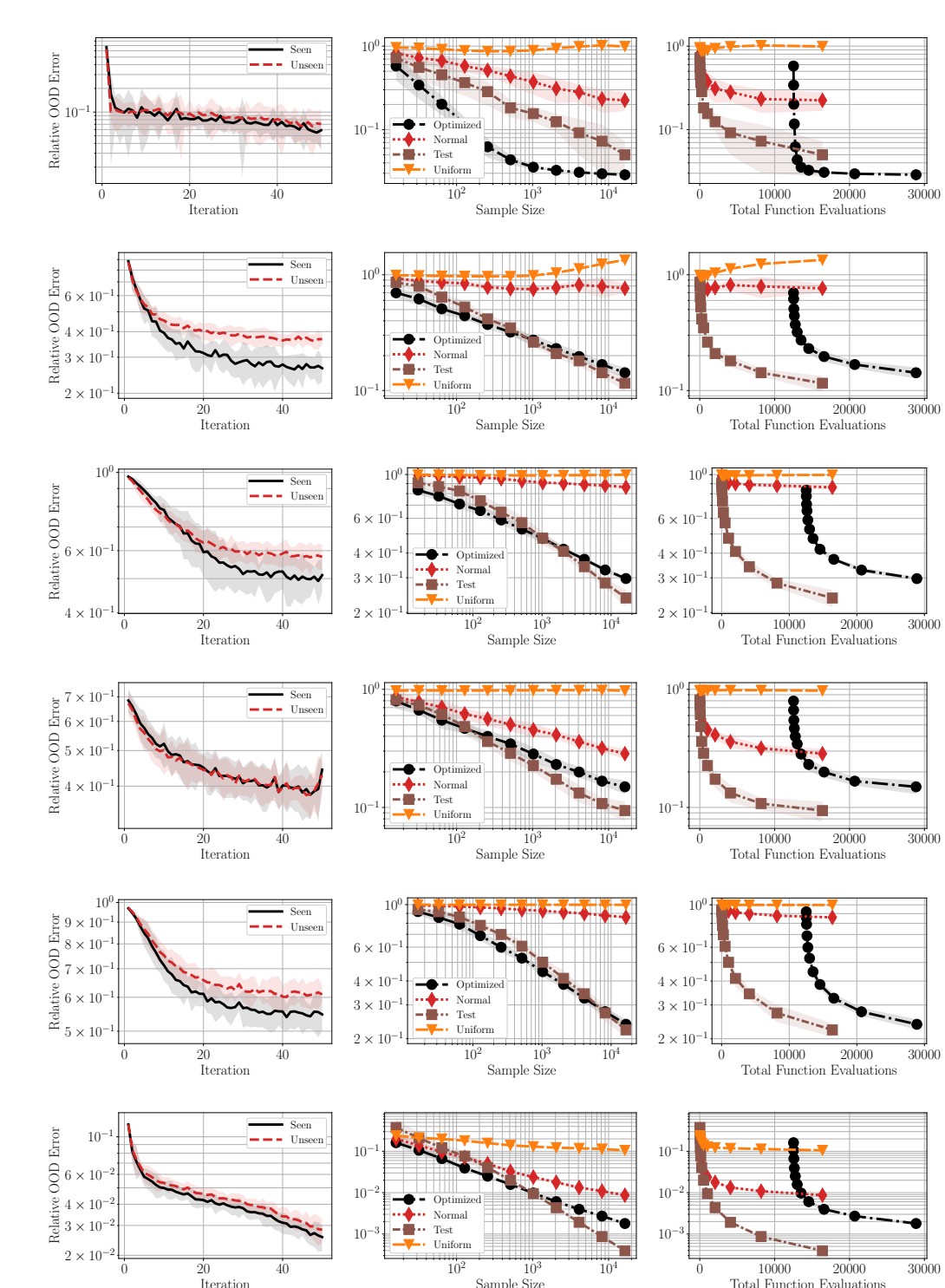

Figure 12: Bilevel Alg. 3 applied to the various ground truths $\{g_i\}$ with $K = 1$ test atoms and 50 gradient descent iterations. (Left column) Evolution of Err from (F.5) vs. iterations. (Center column) Err of model trained on $N$ samples from optimal $\nu_\vartheta$ vs. nonadaptive distributions. (Right column) Same as center, except incorporating the additional function evaluation cost incurred from Alg. 1. From top to bottom, each row represents target functions $\{g_i\}$ corresponding to those listed in each column of Tab. 1 (from left to right). Shading represents two standard deviations away from the mean Err over 10 independent runs.

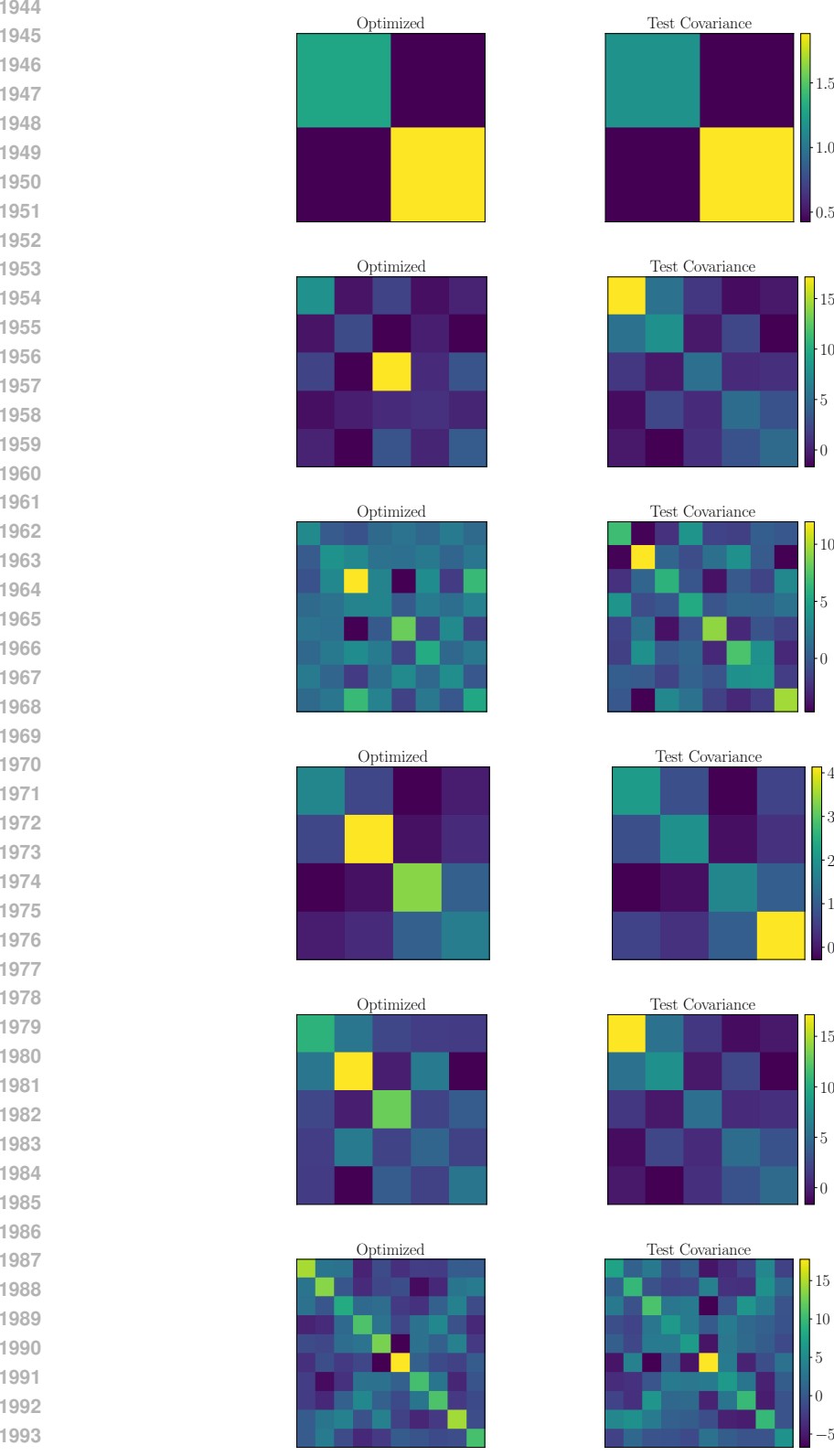

Figure 13: Bilevel Alg. 3 applied with $K = 1$ test atoms and 50 gradient descent iterations. Each column visualizes the covariance matrix of the optimized (left) and empirical test distributions (right), respectively. From top to bottom, each row represents target functions $\{g_i\}$ corresponding to those listed in each column of Tab. 1 (from left to right).

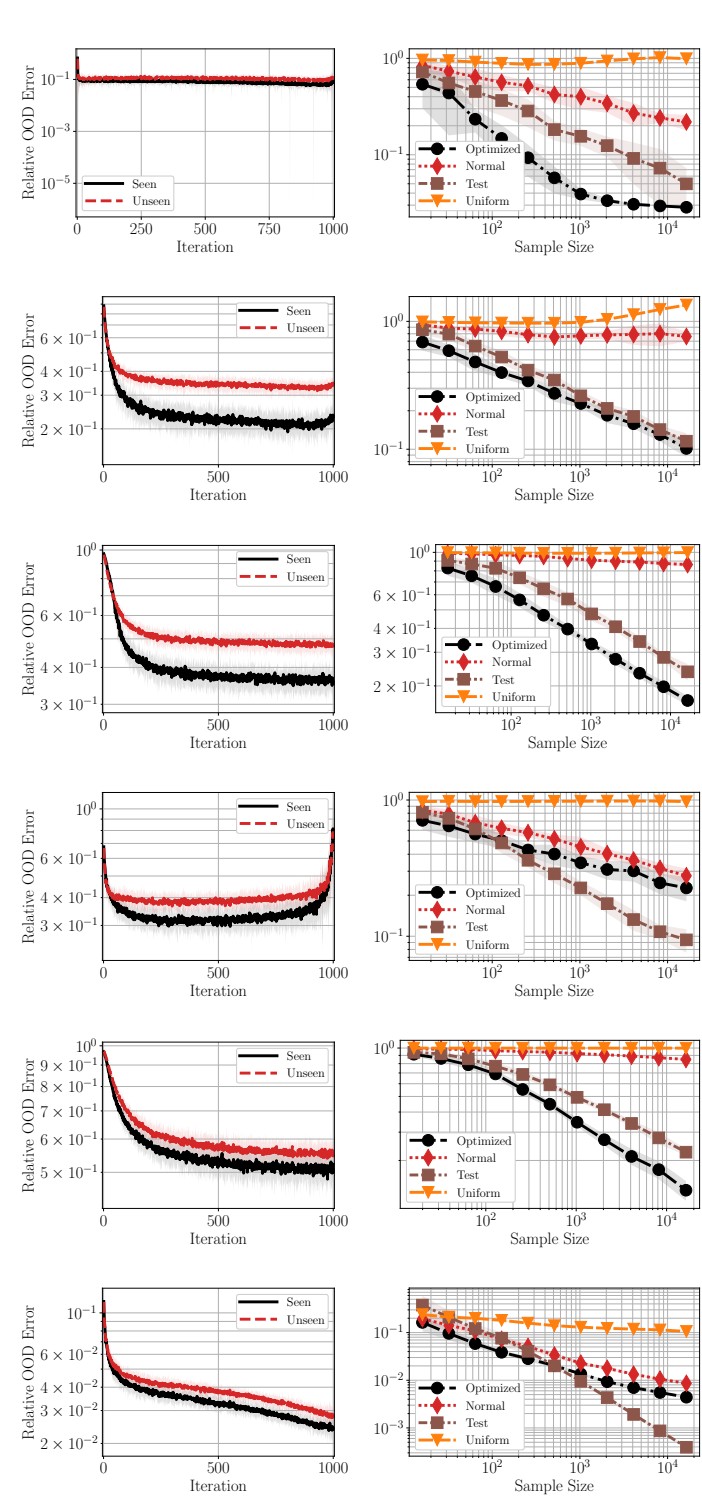

Figure 14: Bilevel Alg. 3 applied to the various ground truths $\{g_i\}$ with $K = 1$ test atoms and 1000 gradient descent iterations. (Left column) Evolution of Err from (F.5) vs. iterations. (Right column) Err of model trained on $N$ samples from optimal $\nu_\vartheta$ vs. nonadaptive distributions. From top to bottom, each row represents target functions $\{g_i\}$ corresponding to those listed in each column of Tab. 1 (from left to right). Shading represents two standard deviations away from the mean Err over 10 independent runs.

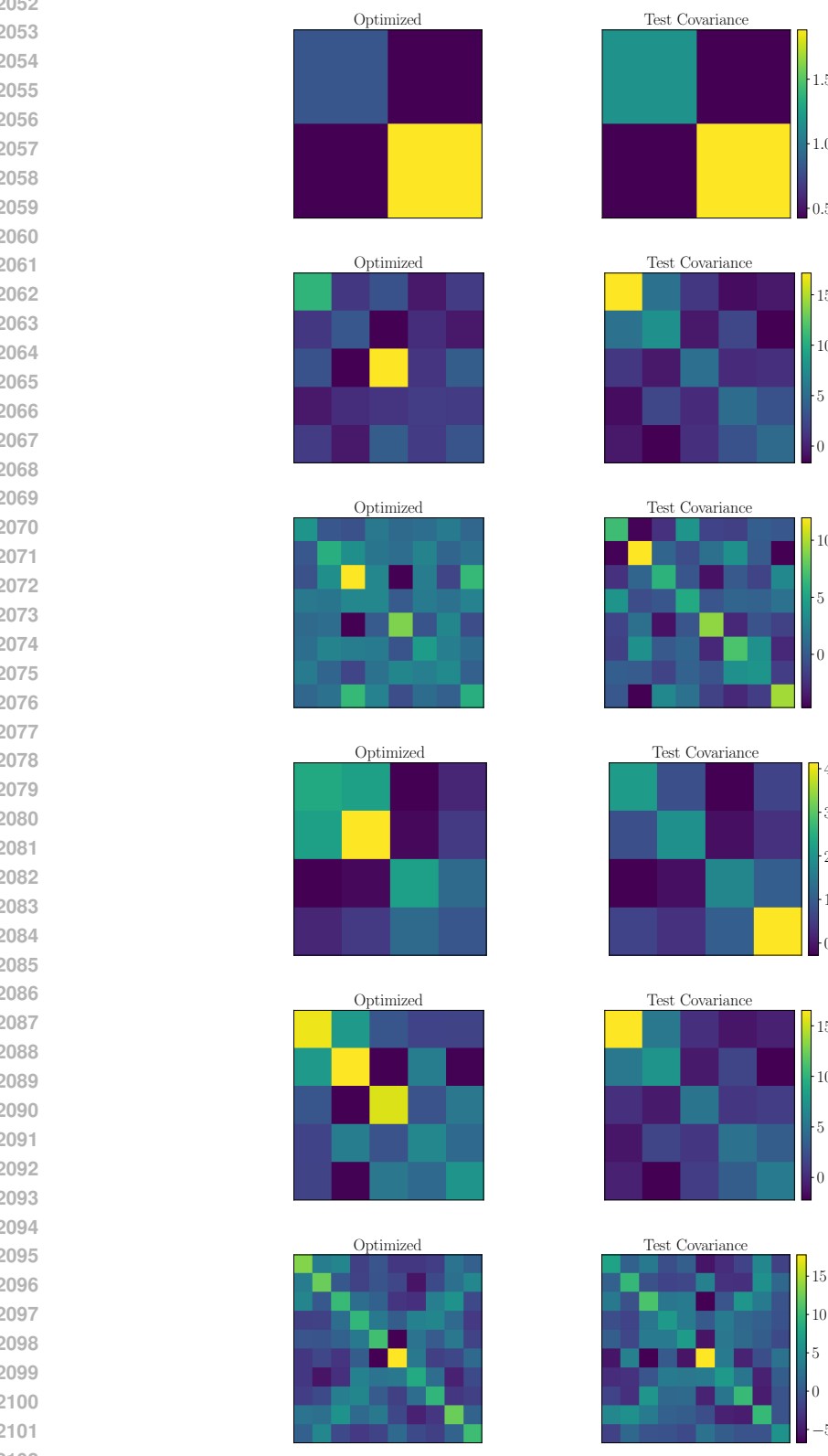

Figure 15: Bilevel Alg. 3 applied with $K = 1$ test atoms and 1000 gradient descent iterations. Each column visualizes the covariance matrix of the optimized (left) and empirical test distributions (right), respectively. From top to bottom, each row represents target functions $\{g_i\}$ corresponding to those listed in each column of Tab. 1 (from left to right).

**Computational resources.** All bilevel minimization experiments are performed in Python using the PyTorch framework in `float32` single precision on a machine equipped with an NVIDIA GeForce RTX 4090 GPU (24 GB VRAM) running CUDA version 12.4, Intel i7-10700K CPU (8 cores/16 threads) running at 4.9 GHz, and 64 GB of DDR4-3200 CL16 RAM. The experiments required less than one total CPU and GPU hour to complete. An additional five CPU and GPU hours were expended to tune hyperparameters and set up the experiments. We ensure reproducibility of our numerical results by appropriately setting PyTorch backends to be deterministic and using fixed seeds for the relevant random number generators.

## F.2 AMINO FOR LEARNING ELECTRICAL IMPEDANCE TOMOGRAPHY

In all machine learning tasks, the choice of training distribution plays a critical role. In practice, testing distributions can span a wide range, making training on the entire space infeasible. Instead, we must identify an optimal training distribution that enables good generalization across diverse testing scenarios.

To illustrate this, we consider the EIT problem with the AMINO solver; see **SM** E.3. We start with the following forward problem setup taken from Molinaro et al. (2023) for EIT on the domain $\Omega := (0,1)^2 \subset \mathbb{R}^2$. The model is the two-dimensional elliptic PDE boundary value problem

$$\begin{cases} \nabla \cdot (a\nabla u) = 0 & \text{in} \quad \Omega, \\ u = f & \text{on} \quad \partial\Omega. \end{cases} \tag{F.6}$$

For one conductivity realization $a$ defined by

$$x \mapsto a(x) = \exp\left(\sum_{k=1}^{m} c_k \sin(k\pi x_1)\sin(k\pi x_2)\right), \tag{F.7}$$

where $m = \lfloor \overline{m} \rfloor$, $\overline{m} \sim \mathsf{Unif}([1,5])$, and $\{c_k\} \sim \mathsf{Unif}([-1,1]^m)$, we take $\mathsf{L} \in \mathbb{N}$ different Dirichlet boundary conditions $\{f_\ell\}_{\ell=1}^{\mathsf{L}}$ for $f$ in (F.6). These are given by

$$x \mapsto f_\ell(x) = \cos\left(\omega\big(x_1\cos(\theta_\ell) + x_2\sin(\theta_\ell)\big)\right) \tag{F.8}$$

with $\omega \sim \mathsf{Unif}([\pi, 3\pi])$ and $\theta_l := 2\pi\ell/\mathsf{L}$ for $\ell = 1, \ldots, \mathsf{L}$. We denote by $\Lambda_a$ the DtN operator that maps the Dirichlet data of the PDE, $f_\ell$, to the Neumann boundary data, $a\frac{\partial u}{\partial \mathbf{n}}|_{\partial\Omega}$, where $u$ is the PDE solution and $\mathbf{n}$ is the outward normal unit vector. Here, the Dirichlet boundary conditions are functions drawn from a probability distribution $\mu$. AMINO learns the map from the joint Dirichlet–Neumann distribution $(\mathrm{Id}, \Lambda_a)_{\#}\mu$ to the conductivity $a$.

Suppose the general distribution of interest for the Dirichlet boundary condition is generated according to the law of $f_l$ in (F.8) with $\omega \sim \mathsf{Unif}([\frac{\pi}{2}, 7\pi])$. A reasonable distribution to train upon is this very same distribution. However, suppose we were constrained to focus and train on distributions corresponding to $\mathrm{Law}(\omega)$ supported only on a subset of the domain $[\frac{\pi}{2}, 7\pi]$. In that case, the choice of the distribution will greatly affect the OOD error when testing with $\omega \sim \mathsf{Unif}([\frac{\pi}{2}, 7\pi])$. To be more concrete, we will focus on four training distributions for $\omega$, which directly determines the input Dirichlet boundary condition as in **SM** E.3: $\mathsf{Unif}([\frac{\pi}{2}, \pi])$, $\mathsf{Unif}([\frac{7\pi}{2}, 4\pi])$, $\mathsf{Unif}([5\pi, \frac{11\pi}{2}])$, and $\mathsf{Unif}([\frac{13\pi}{2}, 7\pi])$. After obtaining the well-trained neural operators (in the sense that the ID generalization error is small enough) corresponding to each training distribution, we study the OOD error of the trained neural operators for Dirichlet data obtained by drawing $\omega \sim \mathsf{Unif}([\frac{\pi}{2}, 7\pi])$ in (F.8).

**Model architecture and optimization algorithm.** We train four distinct AMINO models; see **SM** E.3 to recall AMINO and **SM** E.1 for its DeepONet component. Each model comprises three jointly trained components:

- **Branch encoder**: Eight sequential convolutional blocks with LeakyReLU activations:
  - ConvBlock($1 \rightarrow 64$, kernel $= (1,7)$, stride $= (1,2)$, pad $= (0,3)$)
  - ConvBlock($64 \rightarrow 128$, kernel $= (1,3)$, stride $= (1,2)$, pad $= (0,1)$)
  - ConvBlock($128 \rightarrow 128$, kernel $= (1,3)$, pad $= (0,1)$)
  - ConvBlock($128 \rightarrow 256$, kernel $= (1,3)$, stride $= (1,2)$, pad $= (0,1)$)

- ConvBlock($256 \rightarrow 256$, kernel $= (1, 3)$, pad $= (0, 1)$)
- ConvBlock($256 \rightarrow 512$, kernel $= (1, 3)$, stride $= (1, 2)$, pad $= (0, 1)$)
- ConvBlock($512 \rightarrow 512$, kernel $= (1, 3)$, pad $= (0, 1)$)
- ConvBlock($512 \rightarrow 512$, kernel $= (4, 5)$, pad $= 0$).

The feature map is then flattened and passed through a linear layer to produce the $p$-dimensional branch output.

- **Trunk network**: An 8–layer multilayer perceptron (MLP) with 512 neurons per hidden layer, LeakyReLU activations, zero dropout, and a final output dimension of $p = 100$.

- **Fourier Neural Operator (FNO)**: A single spectral convolution layer with 128 channels and 32 Fourier modes.

All parameters are learned using the AdamW optimizer with a learning rate of $10^{-3}$.

**Training and optimization procedure.**   The four neural operator models "Model 1", "Model 2", "Model 3", and "Model 4" are trained using four different input parameter distributions for $\text{Law}(\omega)$, respectively: $\text{Unif}([\frac{\pi}{2}, \pi])$, $\text{Unif}([\frac{7\pi}{2}, 4\pi])$, $\text{Unif}([5\pi, \frac{11\pi}{2}])$, and $\text{Unif}([\frac{13\pi}{2}, 7\pi])$; see (F.8) to recall how $\omega$ is reflected in the actual input distributions on function space. For each distribution, 4096 training samples are generated. Each sample consists of $\mathsf{L} = 100$ Dirichlet–Neumann boundary condition pairs with the underlying conductivity discretized on a $70 \times 70$ grid for the domain $\Omega = (0, 1)^2$. After 500 training epochs, the trained models are evaluated on 1024 test samples of the distribution for the Dirichlet boundary condition (F.8) determined by $\omega \sim \text{Unif}([\frac{\pi}{2}, 7\pi])$.

**Model evaluation.**   For both ID and OOD settings, we report the average relative $L^1$ error given as

$$\text{Expected Relative } L^1 \text{ Error} := \mathbb{E}_{a \sim \text{Law}(a)}\left[\frac{\left\|a - \mathcal{G}_\theta\big((\text{Id}, \Lambda_a)_{\#}\mu\big)\right\|_{L^1(\Omega)}}{\|a\|_{L^1(\Omega)}}\right], \qquad \text{(F.9)}$$

where $\text{Law}(a)$ is the distribution of $a$ from (F.7) and $\mathcal{G}_\theta$ is the AMINO model. Using the same setup as in E.3, we approximate (F.9) with a Monte Carlo average over $N' = 1024$ test conductivity samples $a_n \overset{\text{i.i.d.}}{\sim} \text{Law}(a)$ and each with $\mathsf{L} = 100$ pairs of corresponding Dirichlet and Neumann boundary data. That is, the expected relative $L^1$ error is approximately given by the average

$$\frac{1}{N'} \sum_{n=1}^{N'} \frac{\left\|a_n - \mathcal{G}_\theta\big(\frac{1}{\mathsf{L}} \sum_{\ell=1}^{\mathsf{L}} \delta_{(f_\ell^{(n)}, g_\ell^{(a_n)})}\big)\right\|_{L^1}}{\|a_n\|_{L^1}}, \text{ where } \big\{(f_\ell^{(n)}, g_\ell^{(a_n)})\big\}_{\ell=1}^{\mathsf{L}} \sim \big((\text{Id}, \Lambda_{a_n})_{\#}\mu\big)^{\otimes \mathsf{L}}.$$

$$\text{(F.10)}$$

The $L^1(\Omega)$ norms in the preceding display are approximated with Riemann sum quadrature over the equally-spaced $70 \times 70$ grid.

Tab. 2 illustrates how OOD performance varies with the choice of training distribution. Model 1 achieves strong ID performance but performs poorly in the OOD setting. Model 2 exhibits consistent yet mediocre performance across both ID and OOD evaluations. Model 4 performs well in the ID setting but shows a noticeable drop in OOD performance. Model 3 offers the best balance, achieving low error in both ID and OOD scenarios. Notably, despite being trained on a Dirichlet data distribution with a small support regarding the parameter $\omega$, Model 3 generalizes significantly better to the larger test domain $[\frac{\pi}{2}, 7\pi]$ for the parameter $\omega$.

These results are further illustrated in Fig. 1, which presents two representative conductivity samples. The top row displays the predicted conductivities from each of the four models when evaluated on Dirichlet–Neumann data pairs sampled from the same distribution used during training. In this ID setting, all four models accurately recover the true conductivity. In contrast, the bottom row shows predictions when the Dirichlet data is drawn from the OOD setting, i.e., $\omega \sim \text{Unif}([\frac{\pi}{2}, 7\pi])$, and the joint Dirichlet–Neumann data distribution also changes accordingly. Under this distributional shift, Models 1, 2, and 4 exhibit noticeable degradation in performance, while Model 3 maintains a reasonable level of accuracy. We use this test to demonstrate the importance of carefully selecting the training data distribution to achieve robust OOD performance.

Table 2: Average relative $L^1$ error of AMINO models under ID and OOD settings for the EIT inverse problem solver. Model 1 is trained on $\omega_1 \sim \mathsf{Unif}([\frac{\pi}{2}, \pi])$, Model 2 on $\omega_2 \sim \mathsf{Unif}([\frac{7\pi}{2}, 4\pi])$, Model 3 on $\omega_3 \sim \mathsf{Unif}([5\pi, \frac{11\pi}{2}])$, and Model 4 on $\omega_4 \sim \mathsf{Unif}([\frac{13\pi}{2}, 7\pi])$. In the OOD setting, all models are evaluated on Dirichlet data from $\omega \sim \mathsf{Unif}([\frac{\pi}{2}, 7\pi])$. One standard deviation is reported. See Fig. 1 for a visual comparison on two representative samples.

|  | Model 1 | Model 2 | Model 3 | Model 4 |
|---|---|---|---|---|
| In-Distribution (ID) | $0.034 \pm 0.019$ | $0.081 \pm 0.028$ | $0.021 \pm 0.010$ | $0.017 \pm 0.009$ |
| Out-of-Distribution (OOD) | $0.796 \pm 0.251$ | $0.078 \pm 0.047$ | $\mathbf{0.066} \pm 0.038$ | $0.132 \pm 0.047$ |

**Computational resources.** The AMINO experiment was conducted on a machine equipped with an AMD Ryzen 9 5950X CPU (16 cores / 32 threads), 128 GB of RAM, and an NVIDIA RTX 3080Ti GPU (10 GB VRAM) running with CUDA version 12.1. Twelve CPU and GPU hours total were needed for the training and testing of all four models.

### F.3 NTD LINEAR OPERATOR LEARNING

In this example from Sec. 5, we begin by defining a conductivity field $a\colon \Omega \subset \mathbb{R}^2 \to \mathbb{R}$ given by

$$x \mapsto a(x) = \exp\left(\sum_{j=1}^{M+1}\sum_{k=1}^{M+1} \frac{2\sigma}{\left((j\pi)^2 + (k\pi)^2 + \tau^2\right)^{\alpha/2}} z_{jk}\sin(j\pi x_1)\sin(k\pi x_2)\right) \qquad \text{(F.11)}$$

with fixed parameters $\alpha = 2$, $\tau = 3$, $\sigma = 3$ and random coefficients $z_{jk} \overset{\text{i.i.d.}}{\sim} \mathcal{N}(0,1)$. Once these random coefficients are drawn, they are fixed throughout the example. Thus, in this experiment, our realization of the conductivity $a$ remains fixed; see the left panel of Fig. 3 for a visualization of this $a$. Our goal is to develop neural operator approximations of the NtD linear operator $\mathcal{G}^\star := \Lambda_a^{-1}$ that maps the Neumann boundary condition to the Dirichlet boundary condition according to the PDE (F.6) (with Neumann data $g$ given instead of Dirichlet data $f$). Using the alternating minimization algorithm (AMA, Alg. 2) from Sec. 4.2, we want to optimize the training distribution over the Neumann boundary conditions to achieve small OOD error (3.3).

To evaluate the OOD error, we need to assign a random measure $\mathbb{Q}$. Let $\mathbb{Q} = \frac{1}{K}\sum_{k=1}^{K}\delta_{\nu'_k}$, where $\{\nu'_k\}_{k=1}^{K}$ are distributions over functions. For a fixed $k$, the function $g \sim \nu'_k$ is a Neumann boundary condition. It takes the form $g\colon \partial\Omega \to \mathbb{R}$, where

$$t \mapsto g\big(r(t)\big) = 10\sin\big(\omega_k\big(t - \tfrac{1}{2}\big)\big) + \sum_{j=1}^{N_{\text{basis}}} \frac{\sigma_k}{\left((j\pi)^2 + \tau_k^2\right)^{\alpha_k/2}} z_j\sqrt{2}\sin(j\pi t)\,. \qquad \text{(F.12)}$$

We view $r\colon [0,1] \to \partial\Omega$ as a 1D parametrization of the boundary $\partial\Omega \subset \mathbb{R}^2$ for the domain $\Omega := (0,1)^2$. The entire domain is discretized on an $M \times M$ grid, where $M$ is as in (F.11). Here $z_j \overset{\text{i.i.d.}}{\sim} \mathcal{N}(0,1)$, $\alpha_k = 1.5$ for all $k$, and $N_{\text{basis}} = 4M - 4$. We set the grid parameter to be $M = 128$ and sample $K = 64$ test distributions in this example. The parameters $\omega_k \overset{\text{i.i.d.}}{\sim} \rho([-2\pi, 2\pi])$ and $\tau_k \overset{\text{i.i.d.}}{\sim} \rho([0, 50])$ are sampled from a custom distribution $\rho = \rho([x_{\min}, x_{\max}])$ and held fixed throughout the algorithm. The custom distribution $\rho$ has the probability density function

$$p(x) = C\sin\big(x - \tfrac{\pi}{2}\big) + 1 \qquad \text{(F.13)}$$

for $x \in [x_{\min}, x_{\max}] \subset \mathbb{R}$, where $C$ is a normalizing constant ensuring that $p$ integrates to 1 over the domain $[x_{\min}, x_{\max}]$. Samples are generated by using inverse transform sampling. Any time a Neumann boundary condition $g$ is sampled according to (F.12) and (F.13), we subtract the spatial mean of $g$ from itself to ensure that the resulting function integrates to zero. This is necessary to make the Neumann PDE problem—as well as the NtD operator itself—well-posed.

The training distribution $\nu$ is assumed to generate functions of the form

$$t \mapsto \widetilde{g}(t) = 10\sin\big(\omega\big(t - \tfrac{1}{2}\big)\big) + \sum_{j=1}^{N_{\text{basis}}} \frac{\sigma}{\left((j\pi)^2 + \tau^2\right)^{\alpha/2}} \widetilde{z}_j\sqrt{2}\sin(j\pi t)\,, \qquad \text{(F.14)}$$

where $\widetilde{z}_j \overset{\text{i.i.d.}}{\sim} \mathcal{N}(0,1)$, $\alpha = 2$, $\tau = 3$, and $\sigma = \tau^{3/2}$. That is, $\nu = \text{Law}(\widetilde{g})$ with $\widetilde{g}$ as in (F.14). The frequency parameter $\omega$ appearing in the mean term of (F.14) is optimized using Alg. 2 to identify a training distribution that minimizes average OOD error with respect to $\mathbb{Q}$.

Note that $g$ and $\widetilde{g}$ are finite-term truncations of the KLE of Gaussian measures with Matérn-like covariance operators; recall Lem. B.2. In particular, for $\widetilde{g}$, and similarly for $g$, the covariance operator of the Gaussian measure has eigenpairs

$$\lambda_j = \frac{\sigma^2}{\left((j\pi)^2 + \tau^2\right)^\alpha} \quad \text{and} \quad t \mapsto \phi_j(t) = \sqrt{2}\,\sin(j\pi t) \quad \text{for} \quad j = 1, 2, \ldots, N_{\text{basis}}. \tag{F.15}$$

Using (F.15) and (B.1), the squared 2-Wasserstein distance in Prop. 3.1 reduces to

$$\mathsf{W}_2^2(\nu, \nu_k') = \int_0^1 \left(10\sin\!\left(\omega\!\left(t - \tfrac{1}{2}\right)\right) - 10\sin\!\left(\omega_k\!\left(t - \tfrac{1}{2}\right)\right)\right)^2 dt + \sum_{j=1}^{N_{\text{basis}}} \left(\sqrt{\lambda_j} - \sqrt{\lambda_{j,k}}\right)^2, \tag{F.16}$$

where $\lambda_{j,k} = \sigma_k^2/((j\pi)^2 + \tau_k^2)^{\alpha_k}$. We approximate the integral with standard quadrature rules. Moreover, the second moment of $\nu$, and similarly for $\nu_k'$, is

$$\mathsf{m}_2(\nu) = \int_0^1 \left(10\sin\!\left(\omega\!\left(t - \tfrac{1}{2}\right)\right)\right)^2 dt + \sum_{j=1}^{N_{\text{basis}}} \lambda_j. \tag{F.17}$$

These closed-form reductions follow directly from the finite-dimensional KLE and Ex. B.1.

Furthermore, we estimate the Lipschitz constants of both the true NtD operator $\mathcal{G}^\star$ and the learned model $\mathcal{G}_\theta$—which appear in the multiplicative factor $c$ from Prop. 3.1—by sampling 250 realizations of Neumann input function pairs $(g_1, g_2)$ from 11 candidate distributions $\nu$. We denote the finite set of these pairs of realizations by $\mathsf{S}$. We then use the quantity

$$\max_{(g_1, g_2) \in \mathsf{S}} \frac{\|\mathcal{G}(g_1) - \mathcal{G}(g_2)\|_{L^2(\partial\Omega)}}{\|g_1 - g_2\|_{L^2(\partial\Omega)}} \tag{F.18}$$

to estimate the Lipschitz constant of map $\mathcal{G} \in \{\mathcal{G}^\star, \mathcal{G}_\theta\}$. These Lipschitz constants are then fixed throughout the algorithm.

**Model architecture and optimization algorithm.** The model architecture is a DeepONet comprising a branch network with five fully connected layers of sizes $(4M - 4)$, 512, 512, 512, and 200, and a trunk network with five fully connected layers of sizes 2, 128, 128, 128, and 200, respectively; see **SM** E.1 for details. All layers use ReLU activation functions. Weights are initialized using the Glorot normal initializer. The model is trained with the Adam optimizer with a learning rate of $10^{-3}$. After the model-training half step in the alternating minimization scheme Alg. 2, the parameter $\omega$ in (F.14) is optimized using SciPy's default implementation of differential evolution with a tolerance of $10^{-7}$. Due to the presence of many local minima in the optimization landscape, this global optimization algorithm is necessary.

**Training and optimization procedure.** The initial value for $\omega$ is set to $8\pi$. For every model-training step, 500 pairs of Neumann and Dirichlet data are generated using the current value of $\omega$. For practical reasons, the model is initially trained for 2000 iterations of Adam. If the AMA loss has not improved relative to the previous iteration, training continues for an additional 2000 iterations. This process is repeated up to 10 times, after which the algorithm proceeds to the parameter optimization step for the training distribution $\nu$. During distribution parameter optimization, the AMA loss is evaluated using 500 newly generated data pairs, used specifically to compute the first term of the objective (4.7). To ensure deterministic behavior, the 500 sets of random coefficients $\{z_j\}_{j=1}^{N_{\text{basis}}}$ are fixed separately for training and optimization. This results in 1000 total sets of coefficients fixed for the entire algorithm. This process is repeated across 80 independent runs using the same conductivity but different realizations of the random coefficients $\{z_j\}$. The relative AMA loss and relative OOD error is computed for each run; see (F.19).

**Model evaluation.** The relative AMA loss is defined as the algorithm's objective value (4.7) normalized by its initial value. The middle panel of Fig. 3 illustrates the rapid decay of this normalized loss over subsequent iterations. The relative OOD loss is defined as

$$\text{Relative OOD Error} := \sqrt{\frac{\mathbb{E}_{\nu' \sim \mathbb{Q}} \mathbb{E}_{g \sim \nu'} \|\mathcal{G}^\star g - \mathcal{G}_\theta(g)\|_{L^2}^2}{\mathbb{E}_{\nu' \sim \mathbb{Q}} \mathbb{E}_{g \sim \nu'} \|\mathcal{G}^\star g\|_{L^2}^2}}, \tag{F.19}$$

which measures the squared prediction error of the model $\mathcal{G}_\theta$ relative to the ground truth NtD operator $\mathcal{G}^\star$ on average with respect to the given random measure $\mathbb{Q}$.

The relative OOD error is estimated using Monte Carlo sampling with $K = 64$ and $N' = 500$ as

$$\text{Relative OOD Error} \approx \sqrt{\frac{\frac{1}{K} \sum_{k=1}^{K} \frac{1}{N} \sum_{n=1}^{N'} \|\mathcal{G}^\star g_{k,n} - \mathcal{G}_\theta(g_{k,n})\|_{L^2}^2}{\frac{1}{K} \sum_{k=1}^{K} \frac{1}{N'} \sum_{n=1}^{N'} \|\mathcal{G}^\star g_{k,n}\|_{L^2}^2}}, \tag{F.20}$$

where $g_{k,n}$ is the $n^{th}$ i.i.d. sample from $\nu'_k$. Further numerical quadrature is used to discretize the $L^2(\partial\Omega)$ norms in the preceding display.

After 80 independent runs of Alg. 2, a 95% confidence interval for the true relative OOD error was computed using a Student's $t$-distribution as seen in the middle panel of Fig. 3.

**Computational resources.** The computational resources here are the same as those reported in **SM** F.2, except three total CPU and GPU hours were needed for all independent runs.

### F.4    DARCY FLOW FORWARD OPERATOR LEARNING

In this example from Sec. 5, we are interested in learning another operator that is relevant and related to the EIT problem. In contrast with the previous example in **SM** F.3 where we fix the conductivity, the Darcy flow forward operator that we consider is nonlinear and maps the log-conductivity function $\log(a)$ to the PDE solution $u$ of

$$\begin{cases} -\nabla \cdot (a\nabla u) = 1 & \text{in} \quad \Omega, \\ \qquad\quad u = 0 & \text{on} \quad \partial\Omega. \end{cases} \tag{F.21}$$

That is, $\mathcal{G}^\star \colon \log(a) \mapsto u$. Here, $\Omega := (0,1)^2$.

Again, we first assign a random measure $\mathbb{Q}$ to evaluate the OOD performance. We take $\mathbb{Q} = \frac{1}{K} \sum_{k=1}^{K} \delta_{\nu'_k}$, where each test distribution $\nu'_k$ is defined by parameters sampled i.i.d. from custom distributions (F.13): $m_k \sim \rho([0,1])$, $\alpha_k \sim \rho([2,3])$, and $\tau_k \sim \rho([1,4])$. The scaling parameter is set as $\sigma_k = \tau_k^{\alpha_k - 1}$. We set $K = 20$ and the grid parameter $M = 64$. Given these parameters, each random log-conductivity function $\log(a')$ from test distribution $\nu'_k$ has the form

$$x \mapsto \log(a'(x)) = m_k + \sum_{i=1}^{M+1} \sum_{j=1}^{M+1} \frac{2\sigma_k}{\left((i\pi)^2 + (j\pi)^2 + \tau_k^2\right)^{\alpha_k/2}} z'_{ij} \sin(i\pi x_1) \sin(j\pi x_2), \tag{F.22}$$

where $z'_{ij} \sim \mathcal{N}(0,1)$ independently. Thus, $a$ is lognormal. Sample log-conductivities $\log(a)$ from training distribution $\nu$ are assumed to follow the same form

$$x \mapsto \log(a(x)) = m + \sum_{i=1}^{M+1} \sum_{j=1}^{M+1} \frac{2\sigma}{\left((i\pi)^2 + (j\pi)^2 + \tau^2\right)^{\alpha/2}} z_{ij} \sin(i\pi x_1) \sin(j\pi x_2), \tag{F.23}$$

with fixed parameters $\alpha = 2$, $\tau = 3$, and $\sigma = 3$ and $z_{ij} \overset{\text{i.i.d.}}{\sim} \mathcal{N}(0,1)$. Alg. 2 is used to optimize the mean shift parameter $m \in \mathbb{R}$ in order to reduce the OOD error. As in **SM** F.3, $\log(a)$ is modeled via a finite KLE of a Gaussian measure with Matérn-like covariance operator. Thus, by working with $\log(a)$ directly, we can exploit (B.1) to write the squared 2-Wasserstein distance factor in objective (4.7) in closed-form.

**Model architecture and optimization algorithm.** The same DeepONet architecture and training setup from **SM** F.3 are used here, except the branch network has layer sizes $M^2$, 256, 256, 128, and 200. After the model-training half step in Alg. 2, the scalar parameter $m$ is optimized using PyTorch's Adam optimizer with a learning rate of $10^{-3}$.

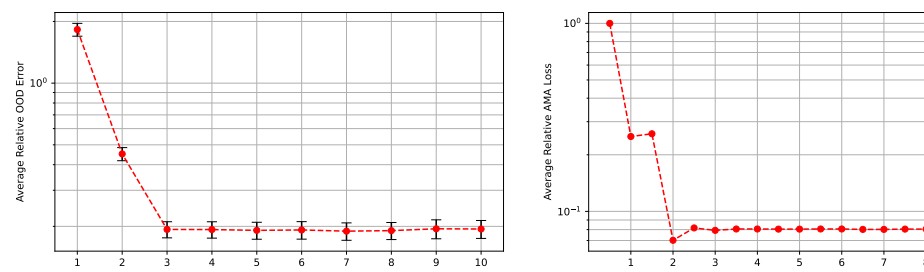

Figure 16: Average relative OOD error and average relative loss of Alg. 2 on the Darcy flow forward problem. (Left) After 13 independent runs, decay of the average relative OOD error of the model when trained on the optimal distribution identified at each iteration; a 95% confidence interval of the true relative OOD error is provided at each iteration. (Right) Decay of average AMA loss defined in (4.7) vs. iteration relative to the same loss at initialization.

**Training and optimization procedure.** The initial value of $m$ is set to zero. The training and optimization procedure follows the same structure as in **SM** F.3, but with the following differences: 200 conductivity-solution pairs are generated for training using the current value of $m$. The model is initially trained for 5000 iterations. If the AMA loss has not improved relative to the previous iteration, training continues for an additional 5000 iterations. This process is repeated up to three times, after which the algorithm proceeds to the distribution parameter optimization step. During distribution parameter optimization, 500 steps are taken. Then the AMA loss is evaluated using three newly generated data pairs. If the AMA loss has not improved relative to the previous iteration, another 500 steps are taken, up to 1500 total steps. After this, the procedure returns to model training again. To ensure deterministic behavior, 203 sets of random coefficients $\{z_{ij}\}$ are fixed across training and optimization. The preceding process is repeated across 13 independent runs, each using different realizations of the coefficients. The average relative OOD error and average relative AMA loss are computed for each run.

**Model evaluation.** After 13 independent runs, the average relative OOD error and relative AMA loss are computed and are shown in Fig. 16. In Fig. 4, we visualize the pointwise absolute error for iterations 1, 2, 3, and 8. At these iterations, the DeepONet model is trained using the optimal training distribution produced by AMA. The model is then evaluated on the same fixed test conductivity. The progressive reduction in the spatial error across these panels of Fig. 4 corroborates the quantitative decrease in the relative OOD error, which converges to approximately $4\%$ in our final model.

**Computational resources.** The computational resources here are the same as those reported in **SM** F.3, except three total CPU and GPU hours were needed for all independent runs.

### F.5 RADIATIVE TRANSPORT OPERATOR LEARNING

In this example from Sec. 5, we consider the hyperbolic PDE for radiative transport governing the particle density $u(z, v)$ at position $z \in \Omega \subset \mathbb{R}^d$ and velocity $v \in V \subset \mathbb{R}^d$:

$$\begin{cases} v \cdot \nabla_z u(z, v) = \frac{1}{\varepsilon} a(z) Q[u](z, v), & (z, v) \in \Omega \times V, \\ u(z, v) = \phi(z, v), & (z, v) \in \Gamma_-, \end{cases} \quad \text{(F.24)}$$

where $a$ is the scattering coefficient, $\varepsilon$ is the Knudsen number, and the collision operator $u \mapsto Q[u]$ is given by

$$Q[u](z, v) := \int_V \mathcal{K}(v, v') u(z, v') \, dv' - u(z, v). \quad \text{(F.25)}$$

In this example, we use the constant kernel $\mathcal{K}(v, v') = \frac{1}{|V|}$. The inflow boundary is defined as

$$\Gamma_- := \{(z, v) \in \partial\Omega \times V \mid -\boldsymbol{n}_z \cdot v \geq 0\}, \quad \text{(F.26)}$$

where $\boldsymbol{n}_z$ denotes the outward normal at $z \in \partial\Omega$.

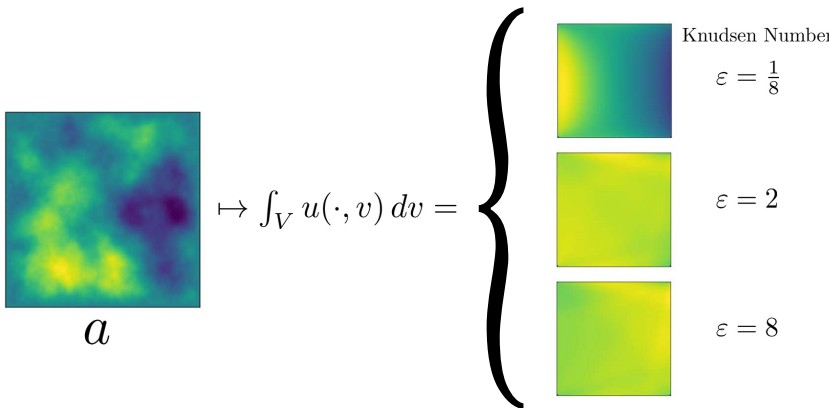

Figure 17: Mapping from scattering coefficient to spatial density in the radiative transport equation for three Knudsen numbers $\varepsilon$. Larger $\varepsilon$ values yield a more hyperbolic regime.

On three separate experiments with error results given in Fig. 5, we set the Knudsen number $\varepsilon$ equal to 1/8, 2, and 8. Changing $\varepsilon$ puts us in different physical regimes as seen in Fig. 17. The inflow boundary condition is given by

$$\phi(z, v) = \begin{cases} \big(1 + 0.5\sin(\pi y/L)\big)\max(v \cdot \boldsymbol{n}_z, 0), & \text{if } z = 0\,, \\ 0, & \text{elsewhere on } \Gamma_-\,. \end{cases} \tag{F.27}$$

We seek the optimal input training distribution $\nu$ for learning the nonlinear operator that maps the scattering coefficient function $a(\cdot)$ to the spatial domain density $\int_V u(\cdot, v)\, dv$.

In this example, $\Omega := (0,1)^2$ is the unit square and $V := \mathbb{S}^1 = \{(\cos\theta, \sin\theta)\,|\,\theta \in [-\pi, \pi)\}$ is the unit circle. Thus, we parametrize $v$ by $\theta$, i.e., $v(\theta) = (\cos\theta, \sin\theta)$ for $\theta \in [-\pi, \pi)$, $\int_V u(\cdot, v)\, dv = \int_{-\pi}^{\pi} u(\cdot, v(\theta))\, d\theta$, and $v \cdot \nabla_z = (\cos\theta)\partial_{x_1} + (\sin\theta)\partial_{x_2}$. We first generate samples from $\mathbb{Q}$ to then evaluate the OOD performance of the model trained on $\nu$. We set $K = 3$. Each distribution $\nu'_k \sim \mathbb{Q}$ is set to be $(\exp)_{\#}\gamma_k$, where $\gamma_k$ is a Gaussian process parametrized by $\sigma_k$, $\alpha$, and $\tau$. Thus, samples from $\nu'_k$ have the form

$$a'(x) = \exp\left( f_k(x) + \sum_{i=1}^{\infty} \sum_{j=1}^{\infty} \frac{2\sigma_k}{\big((i\pi)^2 + (j\pi)^2 + \tau^2\big)^{\alpha/2}} z_{ij}\sin(i\pi x_1)\sin(j\pi x_2) \right), \tag{F.28}$$

where $z_{ij} \overset{\text{i.i.d.}}{\sim} \mathcal{N}(0,1)$. The mean $f_k$ of each distribution $\gamma_k$ is selected as follows. For a fixed centered Gaussian process $\gamma$, draw $f_k \overset{\text{i.i.d.}}{\sim} \gamma$ for each $k$. Then these $f_k$ functions are then fixed once and for all at the beginning of the experiment. By supplying distinct nonzero means to each of the $K$ test distributions, the problem of finding the optimal training distribution becomes harder. Indeed, the test distributions now exhibit not only covariance shifts, but also mean shifts. We set $\tau = 1$ and $\alpha = 2$ for all $K = 3$ test distributions, and set $\sigma_1 = 1$, $\sigma_2 = 2$, and $\sigma_3 = 3$. Note that the parameters of $\gamma$ are set to $\tau = 1$, $\alpha = 2$, and $\sigma = 1$.

**Model architecture and optimization algorithm.** Following Alg. 2, we represent $\nu$ as an empirical measure of $N$ equally weighted particles: $\nu = \frac{1}{N}\sum_{q=1}^{N}\delta_{a_q}$. Each particle $a_q$ is a function of the form

$$a_q(x) = \exp\left( \sum_{i=1}^{\infty} \sum_{j=1}^{\infty} c_{qij} \cdot 2\sin(i\pi x_1)\sin(j\pi x_2) \right), \tag{F.29}$$

where the coefficients $c_q := \{c_{qij}\}_{i,j\geq 1}$ belong to $\ell^2$. Rather than optimizing the particles $a_q$ directly, we optimize their representations in the coefficient space through the transformation

$$a_q = \exp(Lc_q)\,, \tag{F.30}$$

where $L\colon \ell^2 \to L^2(\Omega)$ is the linear operator mapping coefficients to the series in (F.29) and $\exp$ is applied pointwise. The distribution $\nu$ is thus obtained as the pushforward $\nu = (\exp \circ L)_{\#}\mu$, where $\mu$ is the distribution over coefficient sequences. We can then write the loss as $\widetilde{F}(\mu) := F((\exp \circ L)_{\#}\mu) = F(\nu)$, where $F$ is defined as the upper bound in Prop. 3.1, i.e., $\widetilde{F} = F \circ (\exp \circ L)_{\#}$. For each step $k$, the discretized Wasserstein gradient flow update rule with respect to $\{c_q\}_{q=1}^N$ is then

$$c_q^{(k+1)} = c_q^{(k)} - \eta^{(k)} \nabla_{\ell^2} \left[ D\widetilde{F}\big(\mu_N^{(k)}\big) \right] \big(c_q^{(k)}\big), \quad \text{where} \quad \mu_N^{(k)} := \frac{1}{N} \sum_{q'=1}^N \delta_{c_{q'}^{(k)}} \tag{F.31}$$

and $\eta^{(k)}$ is the step size. By the chain rule and linearity of pushforward,

$$D\widetilde{F}(\mu) = DF\big((\exp \circ L)_{\#}\mu\big) \circ (\exp \circ L) \tag{F.32}$$

for each $\mu$, where $DF$ denotes the derivative from Prop. 4.9. Thus, we can write the update rule (F.31) in terms of the known derivative $DF$.

The $\ell^2$ gradient with respect to $c_q$ is computed by a combination of PyTorch's autograd and the adjoint state method. After updating $\{c_q\}_{q=1}^N$, we train a model with discretized versions of particles $\{a_q\}_{q=1}^N$ denoted as $\{\hat{a}_q\}_{q=1}^N$. Each particle $a_q$ is discretized as an $m \times m$ matrix. Similar to the setups from **SM** F.3 and **SM** F.4, the model architecture is a DeepONet. However, the branch network now has 6 fully connected layers of sizes $m^2$, 100, 100, 100, 64, and 32, and the trunk network has 6 fully connected layers of sizes 2, 100, 100, 100, 64 and 32.

**Training and optimization procedure.** During optimization, we work with $N$ samples from each distribution $\{\nu_k'\}_{k=1}^K$ rather than the distributions themselves. For computational simplicity, we set the number of samples from each $\nu_k'$ equal to the number of particles in the training distribution.

The particles $\{a_q\}_{q=1}^N$ are discretized by truncating the number of terms in its expansion, i.e.,

$$a_q(x) = \exp\left( \sum_{i=1}^{N_{\text{truncate}}} \sum_{j=1}^{N_{\text{truncate}}} c_{qij} \cdot 2 \sin(i\pi x_1) \sin(j\pi x_2) \right). \tag{F.33}$$

The coefficient sequences $\{c_q\}_{q=1}^N$ are truncated to $N_{\text{truncate}} = 20$ for both indices $i$ and $j$, yielding sequences of 100 coefficients each. These are initialized by sampling from $\mathcal{N}(0, 10^{-6})$.

We discretize $\Omega$ on a $51 \times 51$ grid ($m = 51$) and $V$ with 12 different directions, i.e. $\Delta\theta = \pi/6$, so we approximate the collision term by

$$Q[u_q] = \frac{1}{|V|} \int_V u_q(z, v') \, dv' - u_q(z, v)$$

$$\approx \frac{1}{|V|} \sum_{i=1}^{12} u_q(\cdot, \theta_i) \, \Delta\theta - u_q$$

$$= \frac{\pi/6}{2\pi} \sum_{i=1}^{12} u_q(\cdot, \theta_i) - u_q$$

$$= \frac{1}{12} \sum_{i=1}^{12} u_q(\cdot, \theta_i) - u_q$$

and the spatial-domain density by

$$\tilde{u}_q := \int_V u_q(\cdot, v) \, dv \approx \sum_{i=1}^{12} u_q(\cdot, \theta_i) \, \Delta\theta. \tag{F.34}$$

Initialize the step size as $\eta = 1 \times 10^{-6}$. At each iteration of Alg. 2:

Table 3: Radiative transport operator learning with particle-based Alg. 2. Reported are mean relative OOD errors with 95% confidence intervals computed over 10 independent runs for three separate models with varying Knudsen number $\varepsilon$. All models have the same DeepONet architecture and are trained with $N = 120$ samples for 5,000 epochs. What makes each model different is their training distribution. The initial model was trained with random particles $c_{qij} \sim \mathcal{N}(0, 10^{-6})$, the optimized model was trained with optimized particles from particle-based AMA, and the benchmark model is a model trained with samples from the mixture of test distributions, i.e., $\nu_{\mathbb{Q}} = \frac{1}{K} \sum_{k=1}^{K} \nu'_k$.

| $\varepsilon$ | Initial | Optimized | Benchmark |
|---|---|---|---|
| 1/8 | $8.65 \times 10^{-2} \pm 8.18 \times 10^{-3}$ | $1.55 \times 10^{-2} \pm 9.72 \times 10^{-4}$ | $1.62 \times 10^{-2} \pm 5.34 \times 10^{-3}$ |
| 2 | $8.76 \times 10^{-2} \pm 8.63 \times 10^{-3}$ | $1.29 \times 10^{-2} \pm 1.20 \times 10^{-3}$ | $1.24 \times 10^{-2} \pm 2.69 \times 10^{-3}$ |
| 8 | $9.09 \times 10^{-2} \pm 4.39 \times 10^{-3}$ | $1.13 \times 10^{-2} \pm 1.33 \times 10^{-3}$ | $1.16 \times 10^{-2} \pm 2.26 \times 10^{-3}$ |

1. Compute the true outputs $\{\tilde{u}_q\}_{q=1}^{N}$ as $m \times m$ matrices using the current $\{c_q\}_{q=1}^{N}$.

2. Train the DeepONet for 5,000 epochs with an 80/20 train-test split.

3. Update $\{c_q\}_{q=1}^{N}$ according to Eqn. (F.31).

4. If the loss $F(\nu)$ increases, halve the step size and recompute the update. The decreased step size carries over to the next iteration.

The stopping condition for the algorithm is if $\|\text{step}^{(k)}\|_2 < \texttt{tol}$, where $\text{step}^{(k)} = \eta^{(k)} \nabla_{\ell^2} \left[ D\widetilde{F}\left(\mu_N^{(k)}\right) \right]$ and $\texttt{tol} = 1 \times 10^{-6}$. Note that in this example, convergence usually occurs in about seven iterations.

For the loss computation in Step 4 above, the second moment of a distribution $\nu$ is estimated with $N$ samples by

$$\mathsf{m}_2(\nu) = \mathbb{E}_{a \sim \nu} \|a\|_{L^2(\Omega)}^2 \approx \frac{1}{N} \sum_{i=1}^{N} \|a_i\|_{L^2(\Omega)}^2, \tag{F.35}$$

where $\|a_i\|_{L^2(\Omega)} = \int_\Omega a_i(z)^2 dz$ is estimated with the trapezoidal rule. Note that for our empirical training distribution, F.35 is not an approximation, but instead an equality. We estimate the Wasserstein distance $\mathsf{W}_2(\nu, \nu')$ between distributions $\nu$ and $\nu'$ using $N$ samples from each, with the ground cost function taken to be $c(a, a') = \|a - a'\|_{L^2(\Omega)}^2$.

**Model evaluation.** We compare models trained on the optimized distribution at each iteration of particle-based AMA with a benchmark model trained on $N$ samples from the mixture of test distributions $\nu_{\mathbb{Q}} = \frac{1}{K} \sum_{k=1}^{K} \nu'_k$ (Eqn. D.1), using the same DeepONet architecture and 5,000 training epochs for all models. Performance is measured by the relative OOD error (Eqn. F.19), approximated with the original $N$ samples from each $\nu'_k$. As shown in Fig. 5, the optimized model improves substantially after particle-based AMA, achieving performance comparable to the benchmark and even outperforming the benchmark in small sample-size regimes. Given that AMA minimizes an upper bound, we are not directly minimizing the OOD error which can account for why the optimized distribution does not outperform the benchmark distribution for most sample sizes. Tab. 3 reports 95% confidence intervals for relative OOD errors over 10 independent trials at $N = 120$, demonstrating that the optimized model reduces its relative OOD error by 82% from its original value on average for $\varepsilon = \frac{1}{8}$, 85% for $\varepsilon = 2$, and 88% for $\varepsilon = 8$. For this particular sample size, the runtime, shown in Fig. 18, averages 20.6 minutes for $\varepsilon = \frac{1}{8}$, 26.7 minutes for $\varepsilon = 2$, and 26.1 minutes for $\varepsilon = 8$.

**Computational resources.** All experiments were conducted on AWS using Galaxy's Deep Learning AMI GPU, PyTorch 2.0.1, and Ubuntu 20.04-prod-ada4rwvf26mda with an EC2 instance type of g5.12xlarge. The experiments for the three different $\varepsilon$ values were run in parallel, with the $\varepsilon = 2$ case taking the longest at 27 hours. Thus, a total of approximately 27 GPU-hours (and corresponding CPU usage) were required for all independent runs with three GPUs running in parallel.

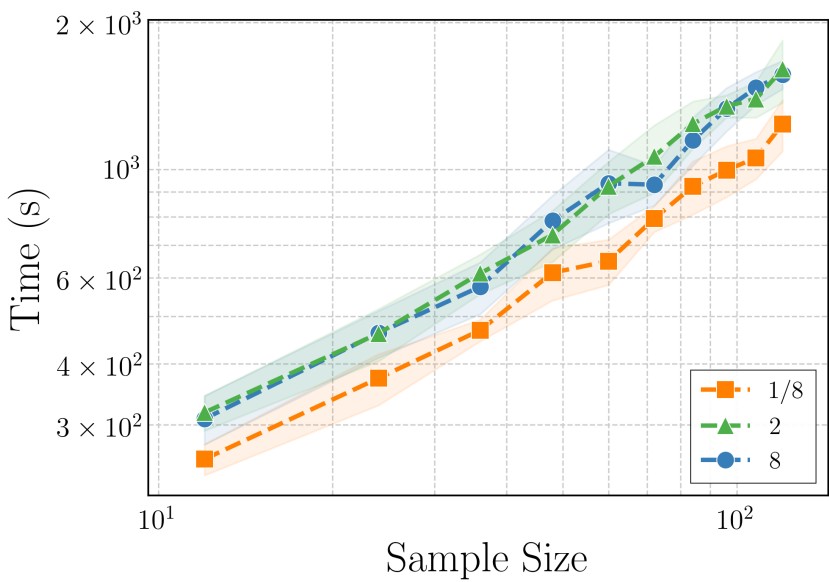

Figure 18: Runtime of particle-based AMA for the radiative transport equation with sample size $N$ at three Knudsen numbers $\varepsilon \in \{1/8, 2, 8\}$. Error bars indicate 95% confidence intervals over 10 trials.

### F.6 BURGERS' EQUATION OPERATOR LEARNING

In the final example from Sec. 5, we consider the viscous Burgers' equation

$$\frac{\partial u}{\partial t} + u\frac{\partial u}{\partial x} = \kappa\frac{\partial^2 u}{\partial x^2} \tag{F.36}$$

with viscosity $\kappa = 0.1$ on the periodic domain $x \in [0, 2\pi]_{\mathrm{per}}$. Our goal is to learn the operator that maps the initial condition $a := u(\,\cdot\,, 0)$ to the solution $u := u(\,\cdot\,, 1)$. This benchmark problem is taken from Nelsen & Stuart (2024); Li et al. (2021); Kovachki et al. (2023).

Following Alg. 2, we model the training data distribution $\nu$ as a Gaussian process with a Matérn covariance kernel. Samples from $\nu$ are generated according to

$$a(x) = \frac{1}{\sqrt{\pi}}\sum_{j=1}^{n/2}\frac{\sigma}{(j^2 + \tau^2)^{\alpha/2}}\left(z_j^{(s)}\sin(jx) + z_j^{(c)}\cos(jx)\right), \tag{F.37}$$

where $z_j^{(s)}$ and $z_j^{(c)}$ are i.i.d. samples from $\mathcal{N}(0,1)$ and $n = 8192$ is the discretization size. Only $n/2$ modes are retained due to the Nyquist limit.

**Model architecture and optimization algorithm.** We adopt the DeepONet architecture described in **SM** F.3. The branch network has layer sizes $n$, 1024, 512, 256, 128, $p$ and the trunk network has layer sizes 1, 128, 128, 128, 128, $p$ where $p = 128$. Training uses Adam with initial learning rate $10^{-3}$ and inverse-time decay (decay steps = 2000, decay rate = 0.9). We add $\ell_2$ regularization via weight decay $10^{-3}$ and apply early stopping on the test loss with patience 1000 and min_delta $= 10^{-4}$. The maximum number of training iterations is 10,000. After the model-training half-step in Alg. 2, we optimize the Matérn covariance parameters $(\alpha, \tau, \sigma)$ by minimizing the upper bound in Prop. 3.1 using `scipy.optimize.minimize`, with gradients from Lem. D.8.

**Training and optimization procedure.** We initialize the parameters at $\alpha = 3$, $\tau = 1.4$, and $\sigma = 5$. All experiments use the fixed publicly available dataset `burgers_data_R10.mat` from Kovachki et al. (2023), which contains 2048 input-output sample pairs. We reserve 10% of the data (205 samples) as a held-out set for estimating the relative OOD error and treat the remaining 1843 samples as i.i.d. draws from a test distribution which we assume to be Gaussian. To optimize the training distribution $\nu$, we generate $N = 1843$ samples from the current parameter values, train the

DeepONet, and then update the parameters using this model and test samples. We repeat this process until the improvement in the objective function falls below $3 \times 10^{-3}$. This occurs at iteration 10, yielding optimal parameters $\alpha = 2.906$, $\tau = 1.456$, $\sigma = 5.005$.

**Model evaluation.** After identifying the optimal training distribution, we compare the model trained on $N$ samples from this distribution with models trained using two pool-based active learning strategies, Query-by-Committee (QbC) and CoreSet. We treat the 1843 test samples as the pool, i.e., $N_{\text{pool}} = 1843$. Each method begins with 10 randomly selected samples, after which we fully train the model, select an additional $n_{\text{select}} = 40$ samples according to the method-specific criterion, and retrain the model from scratch. For subsequent iterations we use $n_{\text{select}} = 50$ until each method reaches a total of 1000 training samples. We run 10 independent trials of this experiment. The selection criteria for each strategy are summarized below.

**Selection criteria.** The pool contains $N_{\text{pool}}$ candidate inputs. Let $n_{\text{select}}$ denote the number of points selected at each acquisition step and let $\Delta x = 2\pi/n$ be the spatial grid spacing.

- **Query-by-Committee (QbC).** Given an ensemble of $M = 4$ models that produce predictions $u_i^{(m)} \in \mathbb{R}^n$ for pool sample $i$ and model $m$, define the ensemble mean $\bar{u}_i = \frac{1}{M} \sum_{m=1}^{M} u_i^{(m)}$. The uncertainty score is the average squared $L^2$ deviation from the mean:

$$\text{uncertainty}(i) := \frac{1}{M} \sum_{m=1}^{M} \left\| u_i^{(m)} - \bar{u}_i \right\|_{L^2}^2 \quad \text{and} \quad \|v\|_{L^2} := \sqrt{\Delta x \sum_{k=1}^{n} v_k^2}. \quad \text{(F.38)}$$

  Select the $n_{\text{select}}$ pool points with largest $\text{uncertainty}(i)$.

- **CoreSet (greedy $k$-center on last-layer features).** Let $k := n_{\text{select}}$. For each pool input, compute a last-layer feature vector $f_i \in \mathbb{R}^p$ from the DeepONet evaluated at point $a_i$ from the pool, apply a Gaussian sketch with $d = 32$, i.e.,

$$U \sim \mathcal{N}(0,1)^{d \times p}, \qquad s_i = \frac{f_i U^\top}{\sqrt{d}} \in \mathbb{R}^d, \quad \text{(F.39)}$$

  and normalize $\tilde{s}_i = s_i / \|s_i\|_2$. Initialize the selected set $S_0$ with one random index and iteratively add

$$i_{t+1} = \arg\max_{i \in \text{pool} \setminus S_t} \min_{j \in S_t} \|\tilde{s}_i - \tilde{s}_j\|_2 \quad \text{(F.40)}$$

  until $|S_T| = k$. Return the indices in $S_T$.

For each model trained on $N$ samples, performance is measured by the relative OOD error (Eqn. F.19), approximated with the 205 held-out samples.

Fig. 6 compares the model trained on $N$ samples from the optimal distribution with the two active-learning-based models. The results show that the model trained on the optimal distribution performs comparably to the active learning methods. A key advantage of the AMA approach is that it allows sampling beyond the size of the fixed data pool, whereas pool-based active learning methods are restricted to selecting at most the available pool samples. We remark that generally within the field of active learning and adaptive sampling, one cannot expect to always outperform nonadaptive/i.i.d. sampling, especially in high dimensions. The target mapping and choice of deployment regime $\mathbb{Q}$ will determine the gains, if any, of active sampling (Adcock & Brugiapaglia, 2022).

**Computational resources.** All experiments were conducted on AWS using Galaxy's Deep Learning AMI GPU, PyTorch 2.0.1, and Ubuntu 20.04-prod-ada4rwvf26mda with an EC2 instance type of g5.12xlarge. The experiments for AMA and the two different active learning methods were run in parallel, with the QbC method taking the longest at 11 hours. Thus, a total of approximately 11 GPU-hours (and corresponding CPU usage) were required for all independent runs with four GPUs running in parallel.

