# OpenReview forum: "Learning Where to Learn: Training Data Distribution Optimization for Scientific Machine Learning"
_ICLR.cc/2026/Conference — Submitted to ICLR 2026_

### Official Review · Reviewer_MHkh · 2025-10-31

**Soundness:** 2
**Presentation:** 2
**Contribution:** 3
**Rating:** 4
**Confidence:** 4

**Summary:**

This submission discusses training algorithms for operator learning: for some operator $G^\star$, find $\hat G$ such that
$$
\hat G = \arg\min_{G} \mathbb{E}_{x \sim \mu} \left[\|G(x)- G^\star(x)\|^2\right]
$$
The quality of predictions with $\hat G$ on test points depends on the distributions of training points $x \sim \mu$.
The submission now discusses two algorithms for optimising those locations via optimising the average-case accuracy (in the sense that the distribution $\mu$ itself is a random variable), which leads to a bilevel optimisation problem:
- One algorithm solves the bilevel problem under the assumption that the hypothesis functions are elements in a reproducing kernel Hilbert space;
- The other algorithm is based on an upper bound of the average case accuracy that decomposes into the sum of in-distribution error and average deviation from the hypothesis measures, which can be optimised alternatingly (like in expectation maximisation).

The resulting algorithm is a theoretically founded way to optimise the training data distribution in operator learning, and experimental results show how this approach outperforms not optimising training data distributions.



**Summary of my recommendation:** Overall, I find the paper has distinct strengths and weaknesses, which makes it difficult to give a definitive recommendation.
On the one hand, I find the infinite-dimensional analysis compelling, but on the other hand, there are gaps in the experimental evaluation.
And while I appreciate the technical thoroughness, the paper is significantly longer than the usual ICLR paper (47 pages), which hinders readability. Altogether, I recommend rejection for now, but I am giving a borderline score because I do appreciate the analysis. I am looking forward to reading what the other reviewers think.

**Strengths:**

**Infinite-dimensional analysis:** the algorithm applies to the infinite-dimensional setting (except for some implementation details, see Lines 343f, but the implementation needing discretisation is typical and not a weakness).
When first reading the manuscript, I was surprised that optimising training distributions with average-case losses was new. I expected it to be known in statistical learning theory, but it turns out that it is only standard knowledge in finite-dimensional settings, and the paper discusses operator learning (which is infinite-dimensional).
As function-space perspectives have grown in popularity in both scientific machine learning and generative modelling recently, the generality of the method fits well into this growing body of literature.


**Correctness and clarity:** I have not found any issues with clarity (or correctness), which, since the derivation of the algorithm is relatively technical, should be considered a major strength of this submission. The solution of the bilevel optimisation problem in Section 4.1 is sound, and the approximation in Section 4.2 is theoretically well motivated.

**Related work:** Finally, I like how the paper discusses related work thoroughly (eg how the derivative in Proposition 4.9 relates to Wasserstein gradients). This is especially convincing since the fields of adaptive sampling and active learning are broad, and many different communities have to be acknowledged.

**Weaknesses:**

**Experimental evaluation:**
The biggest weakness of the submission, in my opinion, is the experimental evaluation:
- The bilevel optimisation problem from Section 4.1 is evaluated only on three finite-dimensional, textbook-style problems, which do not relate to operator learning. Since I read the main contribution of this submission as two algorithms that also work in an infinite-dimensional setup, both of them should be evaluated as such. I recommend benchmarking the bilevel gradient descent algorithm in an operator-learning framework. It would be okay to choose a simple problem, but the benchmark needs to include a function-space scenario.
- The related work section lists numerous algorithms for training data distribution in scientific ML, and I am surprised that none of these algorithms are used as baselines in the experiments. To embed this work better into the scientific ML literature, I recommend including at least one or two such baselines.
- The experiments use very simple PDEs (Poisson problem and transport equation, both in 2d). To make this work comparable to contemporary algorithms, I would recommend including other PDEs that are (at least slightly) more complicated, e.g. Burger's equation or the shallow-water problem; see the PDEBench paper for inspiration: https://arxiv.org/abs/2210.07182.

As is, these experimental weaknesses are the main reason why I lean towards rejection.


**Average-case vs alternatives:**
The paper focuses on average-case expected accuracies, and mentions that other measures would be possible, too (eg worst case; Line 174). Since this choice of average vs worst-case matters for performance, I would recommend justifying this choice somewhere close to Equation 3.3.




**Length:** The submission is 47 pages long. While I understand that it is common practice to defer proofs and additional experiments to appendices, it is problematic if the main paper relies too strongly on these additional results. And I think with the present submission, this is the case:
- The related work section is essentially outsourced to the appendix. The main paper discusses approaches from numerical analysis and scientific machine learning, but active and meta learning, domain generalisation, and experimental design are entirely outsourced to Appendix A. All of these are important connections, and in my opinion, necessary to appreciate the main results of the submissions.
- Many preliminaries are in Appendix B, not in Section 2.
- Crucial examples are only listed in the appendix (eg the one for Corollary 3.2)
- Appendix D contains so much more detail than Section 3 that it is almost required reading.


In total, about 80% of the results are in appendices (in terms of volume), and I find this a bit excessive. It's not something that can be changed easily,  but I wonder whether a longer-form paper would have been a more appropriate medium for presenting the results.

**Questions:**

The following points do not really affect my score, but I find them worth mentioning regardless:

- There is a similar paper by Subedi and Tewari (https://arxiv.org/abs/2504.03503). Since the preprint by Subedi and Tewari was uploaded to arXiv earlier this year, I think it might have been developed in parallel with the submission, so its existence shouldn't affect the acceptance/rejection of the present work. Still, a footnote or so in the submitted manuscript might help the reader connect the articles.
- Line 411f: In the explanation of the numerical examples, a diagram mapping out the Neumann-to-Dirichlet and the parameter-to-solution setups (as in, what is the input and what is the output) would make the descriptions easier to parse. If space permits, I would find that helpful. (If necessary, drop Algorithm 1 from the main paper, because I would expect that gradient descent can be assumed as known at a venue like ICLR.)

---

> ### Author Response · Authors · 2025-11-24
>
> We start by thanking the reviewer for their appreciation of the merits of our paper. Below, we address the questions of the reviewer and thank the reviewer in advance for their patience in reading our detailed reply.
>
> Replies to Weaknesses:
>
> 1. Experimental evaluation:
>
>    * It is difficult to fully satisfy the reviewer’s request while keeping the paper to a reasonable length. Applying Alg. 1 to operator learning would require re-deriving the entire analysis in the setting of infinite-dimensional output spaces, which we estimate would add roughly ten pages. This also introduces nontrivial technical issues, such as defining and controlling function-valued random variables. As a compromise, we will include a heuristic version of this extension in the revision to illustrate how the framework would generalize. The test problem will likely be the NtD operator. Any additional guidance from the reviewer on the specific level of detail they would like to see would be very helpful as we proceed.
>
>    * We certainly agree that such baselines would better contextualize our contribution in SciML. We attempted to add an RPCholesky sampling method for the kernel bilevel method Alg. 1, but for our kernels the algorithm fails at generating more than a few hundred samples due to low numerical rank of the kernel matrices. We will try alternatives from active learning instead. For operator learning, we will include a pool-based active learning baseline for a new Burgers' equation PDE example with Alg. 2 using DeepONets.
>
>    * We chose our PDE examples to illustrate a deliberate progression in complexity. We begin with the NtD map, which is based on a homogeneous elliptic PDE and corresponds to an underlying linear operator. We then move to the Darcy flow example, which uses the same PDE but involves a genuinely nonlinear operator. Finally, we consider the radiative transfer equation (RTE), a substantially more challenging test case: it is a four-dimensional (two spatial, two velocity) hyperbolic, nonlocal integro-differential equation whose behavior can be tuned by the Knudsen number, with larger values producing increasingly transport-dominated dynamics. To further address the reviewer's concern, we will add an additional Burgers' equation example in the revision.
>
> 2. Average-case vs alternatives: We refer to the worst-case metric in Line 174 only to note that it is another possible way to quantify OOD error. We are not suggesting that the remainder of the paper can be developed in this worst-case setting, as doing so would require re-deriving the analysis and formulating entirely new algorithms. In contrast, using expected accuracy is the natural choice when learning from finite samples, and our framework relies on the linearity of the expected OOD error in $\mathbb{Q}$, which is a key advantage over the supremum formulation. We will clarify this point in the revision.
>
> 3. Length: We thank the reviewer for these helpful comments. While the appendix is indeed lengthy, much of the supplementary material is included to meet ICLR’s reproducibility standards by providing full proofs, derivations, and experimental details. Indeed, they are supplementary. Many of the numerical results in the SM provide further background and context, but can be safely ignored without detriment because the main text includes the most crucial findings and figures. Regarding the one page of related work in SM A, we placed the non-PDE/non-SciML literature in the appendix to maintain a coherent narrative in the main text, while ensuring that all essential connections appear in the paper itself. The additional preliminaries and examples (e.g., for Corollary 3.2) are included in the appendix to avoid interrupting the flow, but are not required to follow the main arguments. Similarly, Appendix D offers full technical detail, whereas Section 3 contains all conceptual components needed to understand the core of the method. We appreciate the reviewer’s perspective and will refine the organization to further improve clarity in the revision.

---

> ### Author Response · Authors · 2025-11-24
>
> Replies to Questions:
>
> 1. The preprint linked by the reviewer is a review paper on operator learning and not a contribution on data selection. Nonetheless, we cite it in the revision. Perhaps the reviewer meant to link the paper: Unique Subedi, Ambuj Tewari, On the Benefits of Active Data Collection in Operator Learning, ICML 2025? That paper is a theoretical contribution on active data acquisition which we already cite in our paper on line 142--143. Here, they only study linear target operators (we have nonlinear ones) and do not give a practical algorithm. Their method requires knowing $\mathbb Q$ and chooses input samples deterministically as the eigenfunctions of the covariance of $\mathbb Q$.
>
> 2. We have changed the first image of Figure 3 to be the NtD matrix approximation. Applying this matrix to a given Neumann boundary condition will give us the approximate Dirichlet boundary condition. Fig 4 already shows the input to output fields as requested for Darcy flow. In SM F.5, Fig 17 shows parameter input and solution output for the RTE example as requested.
>
> We sincerely hope that we have addressed all your concerns and kindly request the reviewer to update their assessment accordingly.

---

> > ### Comment · Reviewer_MHkh · 2025-11-26
> >
> > Thank you for the reply.
> >
> > Regarding the bilevel problem, I seem to have misunderstood. I had the impression that both algorithms target operator learning (which to me is an infinite-dimensional setup), which is why I criticised that the bilevel algorithm was only evaluated on finite-dimensional problems. That said, I am wondering whether this difference in scope between the exact and the approximate algorithms (finite- vs infinite-dimensional problems), in combination with the total volume of the submission, might warrant splitting the submission into two in order to explain both setups more succinctly. However, I consider the first point under "1" resolved. Regarding the other two experiment revisions: it would be nice to see the results in the PDF. I think they are valuable additions, and I hope the authors agree with me.
> >
> > Regarding average-case vs alternatives, I did not mean to imply that the paper can be straightforwardly developed in the worst-case setting, and I hope the review did not read as such. My review only mentions the weakness that the choice of developing the theory in average-case (as opposed to alternatives) has not been explained. If there were concrete reasons for choosing average-case over, say, worst-case, I recommend mentioning those in the paper because different targets lead to different behaviour. I did not ask to redo the theory for worst-case metrics.
> >
> > In summary, under the assumption that the changes promised in the rebuttal make it into the final PDF, I will increase my score.

---

> > > ### Author Response · Authors · 2025-11-26
> > >
> > > We thank the reviewer for their clarifying reply and we will take it into account as we continue revising the manuscript. We remain at your disposal during the discussion period if you have any further questions. Until then, we aim to upload the revised paper before the Dec 3rd deadline. In particular, with the valuable experimental revisions suggested by the reviewer (which we agree with) and with the reasons for choosing average-case over worst-case clearly stated.

---

### Official Review · Reviewer_4Yxe · 2025-10-31

**Soundness:** 2
**Presentation:** 2
**Contribution:** 2
**Rating:** 2
**Confidence:** 3

**Summary:**

This paper studies OOD generalization motivated by engineering application. This is done by trying to find an optimal training distribution $\nu$ that minimizes the error across a family of deployment distributions. The paper describes the framework in a rather dense manner. What I took away from it was that you first need an upper bound on the average OOD error, then link it to the 2-Wasserstein distance between train and deployment distributions. This bound will give one important component that will be used. After this, a bilevel algorithm is described. Extensive analysis/results are include but then demonstrated for kernel methods. A separate simpler algorithm minimizes the upper bound and used for the experimental results.

**Strengths:**

1. The paper at a high level is tackling a real problem. Distribution shift is a genuine challenge and the motivating example shown does show that performance can be improved. So the formulation of optimizing training distribution to minimize average error across deployment distributions is practically relevant and makes sense.

2. For a theoretically focused paper, the paper does test the approach across a few different settings. This includes EIT inverse etc and parametric/nonparametric. I feel that this is more than adequate. Also the alternating minimization algorithm is fine. I don't know if something similar exists in the OOD literature (not very familiar with that body of work) but it gives monotone decrease guarantees and can be used. A gradient flow option is also provided.

**Weaknesses:**

There are a number of issues I have with the paper and the presentation.

1. I feel that the technical novelty is quite misrepresented. The paper uses a full basket of classical results (KR duality, adjoint state, envelope theorem) and does minimal effort to clarify which results/propositions should be viewed as contributions unique to this paper. Wouldn't (3.1) follow from existing results for Wasserstein stability or KR duality? The paper never clearly states: "This known results we will use" versus "Here is our novel contribution and this is why its interesting" This makes it exhausting to read the paper because you're wondering what's new. I will acknowledge that these techniques are being applied to operator learning setting.

 2. The theory needs Lipschitz continuous maps with known Lipschitz constants. In practice, one uses neural networks and estimate Lipschitz constants via sampling. Won't these constants be enormous for deep networks making the bounds a bit meaningless. Alg 2 minimizes an upper bound but is this bound tight or how useful it is. Does minimizing the bound correlate with actual OOD error? The entire theoretical framing rests on Lipschitz assumptions that are probably unrealistic. I think this is fine to some extent but the paper really should identify one or more clear insights from the theory and acknowledge what is going on.

3. The optimized distributions sometimes help, but often only marginally. To my reading, the optimized Gaussian often performs similarly to simple baselines (barycenter, mixture)? Sometimes, a single test distribution with sufficient samples and training on the distribution itself is good. So this is confusing because there is no clear guidance on when optimization is worth the cost versus using simple heuristics.

4. It is also not clear whether the extensive and carefully developed RKHS theory apply to any of the paper's main experiments. The alternating algorithm works more generally, but then why develop the inapplicable RKHS machinery at such length? This disconnect between theory and experiments is never explained.

5. Finally, the presentation is excessively dense and also not well structured. The paper throws everything at us. There is no incremental buildup. A simple 2D function approximation example showing the core idea would be far more valuable than pages and pages of abstract machinery with no central theorem highlighted and other results feeding into it. The results appear as Proposition 3.1 etc with no narrative  explaining which are setup, which are contributions, and how they will connect.

6.  The bilevel gradient descent needs computing adjoint states etc, then running iterative optimization. How many function evaluations of the ground truth are needed? What is total runtime analysis? How should we understand comparison of compute cost versus accuracy trade-offs? For expensive PDE solvers, is this optimization worth it or should one focus on more training samples from a simple distribution?

**Questions:**

please see weaknesses above. Answer as many as you can.

---

> ### Author Response · Authors · 2025-11-24
>
> We start by thanking the reviewer for their appreciation of the merits of our paper. Below, we address the questions of the reviewer and thank the reviewer in advance for their patience in reading our detailed reply.
>
> Replies to Weaknesses:
>
> 1. We agree with the reviewer that our theoretical contributions build upon established tools and may seem uninteresting to experts. We will remark on this in the revision. While these results may be known to experts and are easily derived, we are unaware of results in the literature in the exact form needed by us so we include them here. For example, Proposition 3.1 in the Wasserstein-2 case, while straightforward, differs from existing bounds in the literature, which predominantly focus on Wasserstein-1 and rely on Kantorovich-Rubinstein duality. In contrast, we employ a coupling argument to overcome the limitations of this approach. Additionally, for Theorem 4.4, we integrate the adjoint-state method, a technique commonly used in PDE-constrained optimization, to address bilevel optimization where the approximation model is chosen from a vector space of functions (including the RKHS family considered in this work). To the best of our knowledge, this specific combination and its application to optimization over the space of probability measures have not been established as results within the kernel methods community. If there are related works that we may have overlooked, we would greatly appreciate it if the reviewer could bring them to our attention.
>
> 2. We agree that for standard trained deep neural networks, their Lipschitz constants will be enormous. These constants appear in our OOD upper bound. We believe that the bound Alg. 2 minimizes is tight because only sharp inequalities (Cauchy--Schwarz, triangle) and couplings are used. However, we recall that this bound is used to motivate Alg. 2; Alg. 2 need not follow the bound exactly. For instance, the bound exposes two terms, a training error term and a Wasserstein $W_2$ distance term weighted by the Lipschitz constants. While we estimate these Lipschitz constants in the paper for simplicity, one could just as easily treat the factor multiplying $W_2$ as a regularization penalty term. And then determine this penalty using cross validation on a held out validation set. Regardless, our experimental results demonstrate a strong correlation between minimizing the upper bound and minimizing the true OOD error. Beyond Lipschitz assumptions, we also remark in the paper (line 166) that generalizations to Holder continuity also hold. Moreover, we could handle locally Lipschitz maps at the expense of controlling higher moments of the measures.  Finally, we acknowledge the reviewer's broader point: Lipschitz control in deep networks is challenging. Still, there has been substantial recent progress on architectures with explicit Lipschitz constraints (spectral normalization, orthogonal/Householder layers, 1-Lipschitz activations), showing that expressive models with controlled Lipschitz behavior are practical. We will clarify these points and highlight the specific insights from the theory in the revision.
>
> 3. The reviewer raises an important point. Regarding the Gaussian example, Fig. 2 shows an almost order of magnitude improvement over the mixture and barycenter baselines. This is not marginal. For other test problems in Table 1, we observe consistently 15 to 50 percent improvements. The $g_4$ function improvement is marginal because that is an easy test problem, so training on any data distribution does well. However, we totally agree that understanding when to expect order of magnitude gains versus marginal ones is important. It is challenging to give guidance as this is an open question here and in the field of active/adaptive sampling in general; see [r1] and references therein. To truly answer this question would require developing theory to help guide the user to optimally balance additional function evaluation cost with the desired accuracy. This is highly non-trivial due to the need to analyze the infinite-dimensional bilevel optimization problem.
>
> [r1] Adcock, Ben and Brugiapaglia, Simone. Monte Carlo is a good sampling strategy for polynomial approximation in high dimensions. arXiv:2208.09045, 2022

---

> ### Author Response · Authors · 2025-11-24
>
> Replies to Weaknesses (Cont.):
>
> 4. Section 5 (lines 371-373) state that Alg. 1 is applied with RKHS methods directly following our theory. This applies to the substantial number of main bilevel experiments presented in Table 1, Fig 2, and the ten pages of RKHS bilevel experiments in SM F.1, pp 28-38. In particular, the RKHS machinery in conjunction with Alg. 1 showcases the core bilevel framing of the paper without any approximations. Thus, we disagree that the RKHS machinery is inapplicable.
>
> 5. This is very helpful feedback. We agree with the reviewer that, due to this being a theory and algorithm focused paper, some amount of denseness is to be expected. That said, we intend to revise the paper transitions between abstract results to better guide the reader through Sections 3 and 4.
>
> 6. Based on our answer to the reviewers previous question and our answers to other reviewers, a mathematical understanding of the required number of function evaluations and computational complexity analysis are hard theoretical questions that deserve future study in a dedicated paper to those issues. This is a significant challenge that plagues most other complex adaptive target dependent sampling algorithms. The answers will be highly dependent on the problem solved and the meta-distribution $\mathbb{Q}$. What we can do is perform numerical experiments to probe the pareto frontier of the method, and Fig 2 (right) attempts this as a starting point. For PDE problems in which any expensive offline cost is acceptable as long as online performance is best as possible in many-query settings, we expect the training distribution optimization to be worth it.
>
> We sincerely hope that we have addressed all your concerns and kindly request the reviewer to update their assessment accordingly.

---

### Official Review · Reviewer_sZTA · 2025-11-01

**Soundness:** 3
**Presentation:** 3
**Contribution:** 3
**Rating:** 6
**Confidence:** 2

**Summary:**

The paper studies the choice of training data distribution; to determines where a model learns from which is usually fixed and heuristic. The authors propose to optimize the training distribution itself, so that the resulting model generalizes better across anticipated deployment regimes that may differ from the training distribution. The paper proposes Lipschitz-based generalization bounds linking OOD error to training error and the Wasserstein distance between training and test distributions. Two adaptive practical algorithms are proposed: a bilevel procedure that directly minimizes OOD error and a computationally efficient alternating scheme that minimizes an upper bound on the OOD error. Empirical results on involving function approximation and forward and backward operator learning show that the proposed algorithms reduce OOD error, and often outperform nonadaptive baselines.

**Strengths:**

1. The paper has a good introduction to preliminaries.
2. The related work is discussed covering a broader perspective.
3. I like how the paper unifies adaptive sampling in PINNs, domain generalization, and optimal experimental design within a coherent mathematical framework.
4. The bounds relating OOD error, Lipschitz continuity, and Wasserstein distance are sound and help in interpreting the generalization guarantees.
5. The proposed algorithms are demonstrated both using parametric distributions as well as using non-parametric particle based gradient flows.

**Weaknesses:**

1. My main concern is regarding, the experimental evaluation which includes only a small number of baseline settings and lacks systematic ablations across models and problem types. As a result, it is difficult to assess how broadly the proposed training distribution optimization outperforms standard or heuristic designs.
   a) How do author(s) decide the choice of baseline problems?
   b) Can author(s) include comparison with some adaptive sampling method(s)? For example: the several methods that iteratively refine sampling locations (e.g., residual-based PINN adaptation, active learning). Some of these are mentioned in the related work(s).
   c) Can other example(s) be included to help assess the robustness and general applicability of the proposed framework?

2. The computational complexity of the proposed algorithm is not discussed extensively in the numerical experiments. Usually, the incorporation of bilevel optimization increases computation complexity. Can author(s) comment on the increase in wall-clock time of the proposed framework over heuristic (fixed) and adaptive baseline choice(s)?

3. The gradient estimation used for the bilevel optimization associated with both algorithm 1 and 2 is not very clear. Can the author(s) make this clear in the main paper? It will be good to have a discussion on what are the different choices that are available and how this can impact the convergence properties and performance in the numerical experiments.

4. How sensitive is the proposed optimization framework to inaccuracies or misspecification in the deployment prior Q?

**Questions:**

Please see weakness section and associated question.

---

> ### Author Response · Authors · 2025-11-24
>
> We start by thanking the reviewer for their appreciation of the merits of our paper. Below, we address the questions of the reviewer and thank the reviewer in advance for their patience in reading our detailed reply.
>
> Replies to Weaknesses:
> 1.
>
> a.) We would like to clarify that our experiments do span a diverse set of problem types, including both function-approximation tasks with varying behaviors and PDEs of differing complexity. For function approximation, we choose standard approximation theory benchmark functions. These include high-dimensional functions with inactive dimensions or dimensions with varying importance. For operators, we begin with a homogeneous elliptic PDE, where the NtD operator is linear. This provides a baseline level of difficulty. We then increase complexity in the Darcy Flow example, in which the solution operator of interest is nonlinear. Finally, to move beyond elliptic PDEs, we include the RTE example, which is a hyperbolic nonlocal integro-differential equation whose behavior can be tuned via the Knudsen number. It also has 4 independent variables, which goes well beyond existing 2 and 3D PDE problems. Larger Knudsen numbers yield more transport-dominated dynamics, in contrast to the diffusion-driven behavior of the earlier examples.
>
> b.) Thank you for the nice suggestion. We intend to include comparisons with adaptive sampling from randomized numerical linear algebra and pool-based active learning (for DeepONets). Residual-based sampling is primarily found in the unsupervised PINN setting where the PDE is known. In contrast, our paper considers supervised operator learning where we only have data to train. An interesting research question beyond the scope of our paper would be to extend residual sampling to the supervised setting in an efficient way.
>
> c.) Yes, we intend to include a Burgers' equation PDE example to further showcase applicable problem-types (hyperbolic, time dependent), as suggested by another reviewer. Last, we remark that generally within the field of active and adaptive sampling, one cannot expect adaptive sampling to always outperform nonadaptive/i.i.d. sampling, especially in high dimensions. The target mapping and choice of deployment regime will determine the gains, if any, of active sampling. [r1] Adcock, Ben and Brugiapaglia, Simone. Monte Carlo is a good sampling strategy for polynomial approximation in high dimensions. arXiv:2208.09045, 2022.

---

> ### Author Response · Authors · 2025-11-24
>
> Replies to Weaknesses (Cont.):
>
> 2. We agree with the reviewer that bilevel optimization increases computational complexity. However, wall-clock time alone is not the main concern, since acquisition of the sampling points occurs offline. In our setting, the main price is reflected in the additional offline ground truth function or operator evaluations, which may be expensive. Fig. 2 was included to be transparent about this tradeoff between complexity and accuracy. In the rightmost plot, the crossover point in the curves occurs at around 12000 evaluations, where bilevel error continues to decay while non-adaptive error stagnates. While the present manuscript does not focus on balancing cost versus accuracy, a crucial line of future work is to develop theory to help guide the user to optimally balance additional function evaluation cost with the desired accuracy. This is highly non-trivial due to the need to analyze the infinite-dimensional bilevel optimization problem. Practically, the idea is that such cost can be amortized if the final model trained on the optimized data distribution will be used in many-query problems such as PDE-constrained optimization or uncertainty quantification, where the required number of PDE solves with a true solver is intractable.
>
> 3. In the revised paper, we will clarify our references to Remark D.2, Corollary D.7, and Lemma D.8, which provide the exact closed-form gradient expressions. With these exact expressions for the gradients, our method does not require any gradient estimation techniques to compute them other than discretization of integrals or statistical divergences. These discretization details are provided in SM D already. If the reviewer was referring to a different aspect of gradient estimation, then please let us know in more detail.
>
> 4. Like other forms of distribution shift, changes in the deployment meta-distribution Q may degrade performance, especially as the distance between the Q used in Algorithm 1 or 2 and the misspecified Q increases. However, due to the least squares objective being linear in Q (when viewed as an element of signed measures over measures), we can expect Lipschitz stability estimates to hold which suggests that the proposed framework is quite robust to misspecification. However, a mathematical proof would require detailed arguments from the calculus of variations to control distance between minimizers of the objective, which is beyond our scope.
>
> We sincerely hope that we have addressed all your concerns and kindly request the reviewer to update their assessment accordingly.

---

### Official Review · Reviewer_PWdK · 2025-11-02

**Soundness:** 3
**Presentation:** 3
**Contribution:** 3
**Rating:** 6
**Confidence:** 2

**Summary:**

This paper presents a new and principled framework for optimizing training data distributions to enhance the out-of-distribution (OOD) generalization of models in scientific ML. The authors offer a theoretical analysis that links OOD error to the Wasserstein distance between training and test distributions. Building on this theory, they develop two algorithms: a bilevel optimization scheme and an alternating minimization scheme aimed at identifying optimal training distributions. The effectiveness of the proposed methods is demonstrated through extensive experiments.

**Strengths:**

1. The paper addresses the critical challenge of distribution shift in operator learning.

2. The techniques are motivated through fundamental theoretical analysis

3. The authors conduct a thorough empirical study

**Weaknesses:**

## Appropriate Baselines
I do not see a comparison against any baselines[1-5]. I am not sure how to evaluate the proposed approach.

## Scaling the Approach
The experiments use architectures that, while suitable for demonstrating the theory, are not the current state-of-the-art (Fourier Neural Operator or transformer-based neural operators).




1. MetaPhysiCa: OOD Robustness in Physics-informed Machine Learning
2. Open‑Sampling: Exploring Out‑of‑Distribution data for Re‑balancing Long‑tailed datasets
3. Active learning for neural PDE solvers
4. Variational Bayesian Optimal Experimental Design
5. Learning to Sample for Discovering Governing Equations

**Questions:**

N/A

---

> ### Author Response · Authors · 2025-11-24
>
> We start by thanking the reviewer for their appreciation of the merits of our paper. Below, we address the questions of the reviewer and thank the reviewer in advance for their patience in reading our detailed reply.
>
> Replies to Weaknesses:
> 1. We thank the reviewer for noting the lack of explicit baseline comparisons. While our work studies a new problem formulation, namely optimizing the training distribution over function/operator inputs to improve OOD error, we agree that numerical benchmarks will strengthen the paper, and we will add comparisons to pool-based active learning (random, uncertainty, and core-set sampling) to our revision in an operator learning set up. Under a fixed pool $\mathcal{P}$ and budget $N$, our method will produce a distribution $\mu$ and draw $N$ training samples, while each AL baseline will select $N$ points from $\mathcal{P}$. All methods will train the same model with identical hyperparameters and be evaluated on the same held-out sets.
>
> 2. We agree that the vanilla DeepONets used in the paper are by now not likely state-of-the-art (SOTA). However, our framework applies equally well to SOTA architectures such as FNOs or transformer-based neural operators. Indeed, Algorithm 2 is agnostic to the choice of architecture, which is precisely one of its advantages. Our goal in the paper is not to compare across different SOTA architectures, but instead to study the performance of each where-to-learn method for a fixed architecture, in our case, kernel methods and DeepONets. The simplicity of these architectures allows the paper to better isolate the merits of the proposed data distribution design algorithms themselves.
>
> We sincerely hope that we have addressed all your concerns and kindly request the reviewer to update their assessment accordingly.

---

### Official Review · Reviewer_xw3A · 2025-11-03

**Soundness:** 3
**Presentation:** 2
**Contribution:** 2
**Rating:** 4
**Confidence:** 3

**Summary:**

This paper propose to design an algorithm that identifies a training data distribution that generalizes well to test distributions, possibly in out-of-distribution settings. The authors frame this as a "learning-where-to-learn" problem, derive theoretical bounds for OOD error, and propose two practical algorithms that solve a bilevel optimization problem (for the kernel setup) and an alternating minimization problem (for the neural net setup).

Experiments are conducted over SciML applications (e.g. PDE operator learning problems such as EIT, Darcy Flow, and radiative transport), in which the authors show that their proposed approach can improve OOD generalization.

**Strengths:**

- The presentation of mathematical formalisms and algorithms are clear.
- The paper aims to address a significant problem in SciML, namely improving performance of ML models under the the presence of train-test distribution mismatch. The authors' focus on operator learning is especially timely.
- I'm not particularly well-read on the neural operator literature, but I find the authors' approach to solving the bilevel optimization formulation is appropriate for the context, and is different from prior efforts in improving generalization performance in unseen boundary conditions (e.g. [1])
- On a related note, I wasn't able to check the proofs in detail due to limited reviewing time, but I'm optimistic that they appear to be correct.
- The authors have gone great lengths in detailing their design choices and motivations in the supplementary material.


[1] https://arxiv.org/abs/2504.19496

**Weaknesses:**

- The organization / presentation of key contributions of the paper could be improved. In my view, the core contribution of the paper is a tractable learning-where to learn algorithm, but the authors simultaneously introduce a variant of the popular NIO architecture (AMINO) in the appendix and show it demonstrate better OOD performance. Then, the key results are conducted over a combination of the key algorithms and AMINO, which can be confusing and in my opinion, diminish the contribution of the algorithm.
- The improved performance of the algorithm comes at the cost of more function evaluations (e.g. Figure 2, right), which can be expensive to obtain in certain scientific tasks.
- A minor point: while the theoretical analysis of the exact bi-level optimization problem (4.1) is nice, it is only implemented for simpler models such as kernel ridge regression. As the authors stated in D.6, their approach would suffer from vanishing gradient if the inner-loop model overfits (i.e. in the interpolation regime), which is often the case with neural nets. This limitation makes Algorithm 1 feels more like a warm up to Algorithm 2 rather than a standalone contribution.

**Questions:**

- My core concern is the unclear attribution of performance improvements to the proposed algorithms and AMINO. Would it be possible for the authors to ablate OOD performances on:
    - AMINO only with standard training
    - Algorithm 2 with NIO

This would greatly help me understand your contributions. I'm happy to read other reviews and go over additional experiments to adjust my score.

---

> ### Author Response · Authors · 2025-11-24
>
> We start by thanking the reviewer for their appreciation of the merits of our paper. Below, we address the questions of the reviewer and thank the reviewer in advance for their patience in reading our detailed reply.
>
> Replies to Weaknesses:
> 1. We thank the reviewer for the helpful comment and are happy to clarify the role of AMINO in our work. AMINO appears only in Figure 1, where it is used to motivate the where-to-learn question and to illustrate how the choice of training distribution affects OOD performance. It is not used together with Algs. 1 or 2 in the paper. Although we included a brief comparison between AMINO and NIO in the SM, this comparison is not central to our contributions, as our proposed method is model-agnostic and can be applied to any NN architecture. All operator-learning experiments (NtD, Darcy Flow, and RTE) are performed exclusively with DeepONet, not AMINO. For clarity, we will remove the AMINO-NIO comparison from the SM in the revision. We hope this resolves any remaining confusion regarding the scope of our contribution.
>
> 2. We agree that the additional function evaluations can be expensive, and this is the price to pay for an adaptive method. Fig. 2 was included to be transparent about this limitation. While the present manuscript does not focus on balancing cost versus accuracy, a crucial line of future work is to develop theory to help guide the user to optimally balance additional function evaluation cost with the desired accuracy. This is highly non-trivial due to the need to analyze the infinite-dimensional bilevel optimization problem. Practically, the idea is that such cost can be amortized if the final model trained on the optimized data distribution will be used in many-query problems such as PDE-constrained optimization or uncertainty quantification, where the required number of PDE solves with a true solver is intractable.
>
> 3. We emphasize in Line 223-225 that the exact bilevel Algorithm 1 can be applied to other general linear approximation classes, e.g., polynomial chaos expansions, local-averaging estimators, Fourier or Wavelet series, etc. This would just require re-doing the analysis to compute the Fréchet derivative for the chosen class. Our point in Rem. D.6 only refers to a particular discretization choice (Eqn D.11) and is not inherent to Alg. 1 itself. Indeed, the solution we recommend is to simply holdout a validation set of samples to compute Eqn D.11. This immediately removes the vanishing gradient problem even for interpolatory models, and our codebase in the revision has this option.
>
> Replies to Questions:
> 1. As discussed in the response to your first weakness comment, AMINO was never used in conjunction with Algorithm 2 and was only used to motivate the problem. While our work can be used for any model, the operator learning problems (NtD, Darcy Flow, RTE) were implemented with DeepONet. We also emphasize that AMINO and NIO map operators to functions, a formulation well-suited for inverse problems such as Calderón’s problem, but distinct from the operator-learning settings considered in our numerical examples in Sec. 5.
>
> We sincerely hope that we have addressed all your concerns and kindly request the reviewer to update their assessment accordingly.

---

### Author Response · Authors · 2025-12-03
**Summary of Contributions and Rebuttal**

**Summary of Contributions and Rebuttal.** We thank the reviewers for their constructive feedback and the AC for their careful evaluation of our work. We appreciate the opportunity to clarify our contributions and address the concerns raised, and have revised our work accordingly. The revised manuscript has been uploaded and changes are marked in blue text. Below is a concise summary of the concrete changes made and of the clarifications to our contributions.

**Core Contributions:**
-  **Lipschitz-based distribution shift bounds.** We derive quantitative bounds linking the OOD error of the model to the in-distribution training error, the Lipschitz constants of the model and target operator, and the Wasserstein distance between the training and test distributions.

- **Algorithms for training distribution optimization.** Guided by the above bounds, we develop two adaptive algorithms for optimizing the training distribution: (1) a bilevel procedure that directly minimizes the OOD error, and (2) an efficient alternating scheme that minimizes an upper bound. We implement both methods in the space of probability measures using (a) parametric families (e.g., Gaussian processes) and (b) nonparametric particle representations updated via Wasserstein gradient flows.

-  **Empirical performance in SciML tasks.** We evaluate our approach on function approximation and on forward and inverse operator learning across elliptic, parabolic, and hyperbolic PDEs (EIT, Darcy flow, viscous Burgers’, and radiative transport). The results confirm that the training distribution significantly affects OOD performance. Our adaptive algorithms effectively optimize this distribution, substantially reduce OOD error, and often outperform adaptive and nonadaptive baselines.

**Concrete changes and clarifications to address reviewer concerns:**
- **Improved clarity of contributions and structure.** Reviewers requested clearer attribution of novelty and a more guided narrative. We added explicit statements identifying which results are classical and which are new, clarified the conceptual flow in Sections 3 and 4, and clarified the distinct roles of Algorithms 1 and 2.

- **Removal of confusing AMINO comparisons.** Reviewer xw3A noted that AMINO–NIO comparisons in the supplementary material distracted from the main contribution. We removed this comparison entirely and clarified that AMINO appears only as motivation in Figure 1 and was never used in our algorithms.

- **Expanded baseline comparisons.** Reviewers requested comparisons with adaptive sampling and active learning. We added pool-based active learning baselines (uncertainty, core-set) for operator learning with Algorithm 2, and adaptive sampling baselines for the RKHS experiments where feasible (adaptive coreset, and new Table 1 and Fig 2). We also added a new Burgers' equation experiment with these baselines.

- **Additional PDE experiments.** To broaden the applicability beyond elliptic and simple transport PDEs, we added a Burgers' equation example and enhanced the discussion of the radiative transfer equation and the NtD/Darcy setups.

- **Clarification of theory–practice connections.** We clarified the role of Lipschitz constants in trained networks, the tightness of the Wasserstein-based upper bound, and the interpretation of the constant in Algorithm 2 as a regularization weight. We also clarified that Algorithm 1 fully applies to all RKHS-based experiments already in the paper. Finally, we explained why average-case OOD error, not worst-case error, is the natural formulation for learning from finite samples.

- **Discussion of computational cost.** We expanded the discussion of cost-accuracy tradeoffs, clarified that additional ground-truth evaluations occur offline, and explained scenarios where amortization makes the approach advantageous. Figure 2’s interpretation was clarified accordingly.

**Overall improvements.** Across the revision, we added baselines, added new PDE experiments, provided new theoretical clarifications, removed distracting material, and significantly improved the clarity and structure of the exposition. These changes directly address the reviewers' core concerns and substantially strengthen both the theoretical and empirical parts of the paper. In particular, before the reset, Reviewer MHkh increased our paper's score from 4 to 6 and indicated that we resolved their concerns.

---

### Meta-Review · Area_Chair_TdRa · 2026-01-15

**Summary:**

**Reviewer xw3A:** Presentation has a room for improvement. Cost of function evaluation is not considered.

**Reviewer PWdK:** Comparison with baselines missing. Architectures used in experiments are not state-of-the-art.

**Reviewer sZTA:** Experiments include only a few baselines. Computational complexity is not discussed well. Unclear description of gradient estimation. Sensitivity of the proposal to mismatched deployment prior is not discussed.

**Reviewer 4Yxe:** Novelty is not properly represented. Practical significance of Lipschitz assumption is not clear. No clear guidance on when to optimize distributions is given. Dense and less-structured presentation. Missing comparison of function evaluation / computation vs. accuracy.

**Reviewer MHkh:** Bilevel optimization is evaluated only on a few finite-dimensional problems. Comparison with baselines missing. Experiments on more complex PDE should also be included. Justification of choosing average-case accuracy is missing. Strong reliance on appendices.

**Reviewer Concerns:**

**Cost of function evaluation / computation (xw3A, sZTA, 4Yxe):** Although the cost of function evaluation has been demonstrated in Figure 2 and similar figures in Appendices, there is no explicit discussion on this issue in the manuscript.

**Missing baselines (PWdK, sZTA, MHkh):** Experimental results in the revised manuscript include additional baselines, which however would not necesarily support the strength of the proposal.

**More complex PDE (MHkh):** Results on viscous Bergers' equation have been added (Figure 6).

**Guidance (4Yxe):** One can admit that there certainly are cases where optimization helps (e.g., Figure 2), and that giving guidance on when to optimize should be challenging, one would still need some guidance, especially in view of the results presented in the manuscript, as one can find several instances where the proposal does not provide gain in terms of the cost of function evaluation.

**Reviewer Scores:**

The initial evaluations of Reviewers sZTA and PWdK were on the positive side. Reviewer MHkh explicitly stated that he/she will raise the score. The initial evaluations of the two remaining reviewers were negative, and in view of the above concerns, I feel it unlikely that they would have both changed their evaluations to the positive side. I would thus like to encourage the authors to thoroughly reorganize/revise the manuscript reflecting the reviewers' concerns, expecting further improvement in presentation clarity (xw3A, 4Yze, MHkh).

---

### Decision · Program_Chairs · 2026-01-26

Reject